# Attention Mechanism, Max-Affine Partition, and Universal Approximation

**Hude Liu**[*]   **Jerry Yao-Chieh Hu**[*†]   **Zhao Song**[‡]   **Han Liu**[†§]

[†]Center for Foundation Models and Generative AI & Department of Computer Science,
[‡]Simons Institute for the Theory of Computing, UC Berkeley, Berkeley, CA 94720, USA
[§]Department of Statistics and Data Science, Northwestern University, Evanston, IL 60208, USA

hudeliu0208@gmail.com, jhu@u.northwestern.edu,
magic.linuxkde@gmail.com, hanliu@northwestern.edu

## Abstract

We establish the universal approximation capability of single-layer, single-head self- and cross-attention mechanisms with minimal attached structures. Our key insight is to interpret single-head attention as an input domain-partition mechanism that assigns distinct values to subregions. This allows us to engineer the attention weights such that this assignment imitates the target function. Building on this, we prove that a single self-attention layer, preceded by sum-of-linear transformations, is capable of approximating any continuous function on a compact domain under the $L_\infty$-norm. Furthermore, we extend this construction to approximate any Lebesgue integrable function under $L_p$-norm for $1 \le p < \infty$. Lastly, we also extend our techniques and show that, for the first time, single-head cross-attention achieves the same universal approximation guarantees.

## 1   Introduction

We establish the universal approximation capability of single-layer, single-head self- and cross-attention mechanisms. Departed from prior studies, our results demonstrate that the expressive power of transformers arises from *only* the (softmax) attention module and an attached linear layer, without additional components such as positional encodings or feed-forward networks (FFNs). More importantly, our proofs show that sequence-to-sequence universal approximation requires only a minimalist configuration: single-layer, single-head attention with linear transformations.

In this era, the power of transformers [Vaswani et al., 2017] is undeniable, given their dominance in modern machine learning. They drive models such as BERT [Devlin, 2018], ChatGPT [Brown et al., 2020, Achiam et al., 2023], and LLaMA [Touvron et al., 2023a,b, Dubey et al., 2024] for language; ViT [Dosovitskiy et al., 2021] and DiT [Peebles and Xie, 2023] for image and video; DNABERT [Ji et al., 2021, Zhou et al., 2023] for genomics; and Moirai [Woo et al., 2024, Liu et al., 2024] for time series, among many others. Central to these successes is the *attention mechanism*. While numerous variants and implementations exist [Tay et al., 2022], the *softmax-based* vanilla attention

---

[*]Equal contribution. Version: December 15, 2025. Future updates are on https://arxiv.org/abs/2504.19901.

[Vaswani et al., 2017] remains a mainstay in both research and industry communities (e.g., ChatGPT and Llama).

However, despite its practical importance, theoretical insights into why softmax attention is so powerful remain incomplete. Moreover, the extent to which softmax attention alone drives performance is unclear. Empirical [Tay et al., 2022] and theoretical [Keles et al., 2023, Deng et al., 2023, Alman and Yu, 2024] evidence suggests that deviating from softmax attention (e.g., via sub-quadratic approximations) often degrades performance, indicating that softmax attention may be a central engine in Transformer architectures. At the same time, a growing body of work explores its memory capacity [Mahdavi et al., 2023, Kim et al., 2023, Kajitsuka and Sato, 2024], universal approximation properties [Yun et al., 2019, Kajitsuka and Sato, 2023, Jiang and Li, 2023], representation learning [Sanford et al., 2024b, Chen and Li, 2024], and task-specific theoretical performance [Gurevych et al., 2022, Edelman et al., 2022]. However, these studies often rely on additional components, such as feed-forward networks (FFNs) or multi-head setups or customized assumptions, as they target the entire Transformer architecture rather than isolating the role of attention module.

To this end, this work presents attention-only expressiveness results: *softmax-based attention alone* already suffices for universal approximation of sequence-to-sequence functions. We operate under three key premises for investigating the expressiveness of attention:

1. We focus on *softmax*-based attention,
2. We seek a *minimalist* design (a single layer of single-head attention plus a linear transformation),
3. We impose *minimal assumptions* on the data distribution or network architecture (no positional encodings, no multi-head expansions, no FFNs).

We provide new proofs that a *single* self-attention layer approximates any continuous sequence-to-sequence function on a compact domain, in both the $L_\infty$ and $L_p$ norms. Furthermore, we show, for the first time, a parallel result for *cross-attention*, revealing its universal approximation capability under the same minimalist setting.

**Contributions.** Our contributions are as follows:

- **Interpreting Attention as a Max-Affine Partition.** We show that single-head softmax attention, combined with a linear layer, implicitly partitions the input domain using a max-affine construction. This partitioning allows attention to assign distinct outputs to each partition cell. This perspective clarifies how softmax-based attention enables a powerful piecewise-linear approximation scheme.
- **Single-Layer, Single-Head Self-Attention Universality.** We prove that a single self-attention layer is a universal approximator for continuous sequence-to-sequence functions on compact domains. Our results cover both $L_p$- and $L_\infty$-norms guarantees and require minimal assumptions on data and architecture, highlighting the inherent expressive power of attention alone.
- **Single-Head Cross-Attention Universality.** We establish, for the first time, that the same approach also endows a single-layer, single-head *cross*-attention with universal approximation capabilities. This result further underscores that much of a Transformer's expressiveness can reside solely in its attention block, even when the queries and keys come from distinct input sequences.

**Organization.** Section 2 presents the ideas we built on. Section 3 shows our interpretation of Attention as a Max-Affine Partition in a simplified setting. Section 4 presents our universal approximation results for single-layer, single-head self- and cross-attentions.

## 1.1 Related Work

**Universal Approximation.** Early works of universal approximation theorems focuses on the expressiveness of feed-forward networks (FFN) [Cybenko, 1989, Hornik, 1991, Carroll and Dickinson,

1989]. Since Vaswani et al. [2017] propose the transformer architecture and the scaled dot-product attention module, there is a series of research aiming to explain the expressiveness of transformer. Yun et al. [2019], Kajitsuka and Sato [2023] offer explanation from the perspective of contextual mapping. Among them, Yun et al. [2019] are the first to prove the universal approximation capability of transformer. Yet since the network in [Yun et al., 2019] requires excessive layers ($\mathcal{O}(n(1/\delta)^{dn}/n!)$), Kajitsuka and Sato [2023] make more careful estimation upon the numerical results of contextual mapping and proves that with skip connections, a one-layer transformer is capable of approximating any permutation equivariant continuous function. Takakura and Suzuki [2023] add positional encoding to lift the restriction of permutation equivariance, and demonstrate a one-layer transformer approximates shift-equivariant $\alpha$-smoothness function with an error independent of input and output dimension. Jiang and Li [2023] give a non-constructive proof using Kolmogorov representation theorem on the Jackson-type approximation rate of a two-layer transformer. While prior works have achieves diverse and extensive result regarding the expressive capability of transformer, their results require the feed-forward network (FFN) to add expressiveness to the attention module in order to achieve universal approximation, which differs from our results derived from attention-only network. Concurrently, Hu et al. [2025a] give an interpolation-based proof that softmax attention alone (no FFN) is a universal approximator for continuous sequence-to-sequence maps on compact domains.

**Provable Capabilities of Transformer.** Recent theoretical studies also shed light on the practical behavior of attention mechanism. Olsson et al. [2022] show that induction heads help models learn patterns in context. Sanford et al. [2024a] prove that Transformers can do complex computations with few layers because they work in parallel. In contrast, Luo et al. [2022] find that some Transformer designs lose expressivity when using relative positional encodings. Kim and Suzuki [2024], Chen et al. [2025] provide Transformer's hardness results on learning constrained boolean functions. Building on [Hu et al., 2025a], Hu et al. [2025b] show that a fixed two-attention-layer softmax Transformer is prompt-programmable: it emulates any algorithm implementable by a single attention layer (cf. [Bai et al., 2023]), providing a constructive account of one-model-many-tasks behavior with softmax (not ReLU) Transformers. To add on these ideas, we prove that a single-layer, single-head softmax attention with a simple linear layer can approximate any continuous function on a compact domain. This shows that attention alone can learn arbitrary sequence-to-sequence mappings.

## 2 Preliminaries

We now present some ideas we built on.

**Notation.** For a vector $v$, we denote its $i$-th entry by $v_i$ and its subvector from the $i_1$-th to the $i_2$-th entry (inclusive) by $v_{i_1:i_2}$ with $i_1 < i_2$. For a matrix $M$, we use $M_{i,j}$ for the entry in the $i$-th row and $j$-th column, $M_{i,:}$ for the $i$-th row, and $M_{:,j}$ for the $j$-th column. The submatrix spanning rows $i_1$ through $i_2$ and columns $j_1$ through $j_2$ is denoted by $M_{i_1:i_2,,j_1:j_2}$ with $i_1 < i_2,, j_1 < j_2$. We define $c_{a \times b}$ as an $a \times b$ matrix with constant entries $c$, and abbreviate $c_{a \times 1}$ as $c_a$. For norms, we define $\|\cdot\|_\infty$ as the maximum absolute element in a vector or matrix. The $p$-norms are given by $\|v\|_p = (\sum_i |v_i|^p)^{1/p}$ for a vector $v$ and $\|M\|_p = (\sum_{i,j} |M_{i,j}|^p)^{1/p}$ for a matrix $M$. For function norms, we define the $L_\infty$ norm as $\|f\|_{L_\infty} := \sup_{x \in X_f} \|f(x)\|_\infty$, where $X_f$ is the input domain of $f$, and the $L_p$ norm as $\|f\|_{L_p} := (\int_{x \in X_f} |f(x)|_p^p, dx)^{1/p}$ for $1 \le p < \infty$. For functions, when a function $f : \mathbb{R} \to \mathbb{R}$ is applied on a vector or a matrix, it means to apply $f$ on every entry of the vector/matrix (i.e.,$\exp([a_1, a_2]) := [\exp(a_1), \exp(a_2)]$).

**Self-Attention and Cross-Attention Layers.** For a self-attention $\text{Attn}_s : \mathbb{R}^{D \times N} \to \mathbb{R}^{D \times N_{\text{out}}}$, and any input $Z \in \mathbb{R}^{D \times N}$, we define its output as:

$$\text{Attn}_s(Z) = W_V Z \, \text{Softmax}((W_K Z)^\top W_Q Z) W_O,$$

where $W_K, W_Q \in \mathbb{R}^{d_{\text{Attn}} \times D}, W_V \in \mathbb{R}^{D \times D}, W_O \in \mathbb{R}^{N \times N_{\text{out}}}$. Here $d_{\text{Attn}}$ stands for the hidden size of the attention block. $N_{\text{out}}$ stands for the output sequence length.

For a cross-attention $\text{Attn}_c : \mathbb{R}^{D \times N} \times \mathbb{R}^{D \times N} \to \mathbb{R}^{D \times N_{\text{out}}}$ and any input $Z_K, Z_Q \in \mathbb{R}^{D \times N}$, we define its output as:

$$\text{Attn}_c(Z_K, Z_Q) = W_V Z_K \, \text{Softmax}((W_K Z_K)^\top W_Q Z_Q) W_O.$$

Here $W_K, W_Q, W_V, W_O$ are defined as those in self-attention.

Since we provide separate discussions for self-attention and cross-attention in this work, we omit the subscript and denote them as $\mathrm{Attn}$ when this causes no ambiguity.

**Layer of Sum of Linear Transformations.** We use $\mathrm{Linear} : \mathbb{R}^{D_1 \times N_1} \to \mathbb{R}^{D_2 \times N_2}$ to denote a layer of sum of linear transformations. For any input $Z \in \mathbb{R}^{D_1 \times N_1}$, we define its output as follows:

$$\mathrm{Linear}(Z) := \sum_{i=1}^{H} P_i Z Q_i + R,$$

where $P_i \in \mathbb{R}^{D_2 \times D_1}$, $Q_i \in \mathbb{R}^{N_1 \times N_2}$ for $i \in [H]$, $R \in \mathbb{R}^{D_2 \times N_2}$. Here $H$ is a positive integer which denotes the number of linear transformations to sum.

## 3 Attention as Max-Affine Value Reassignment

In this section, we introduce a new interpretation of attention as a value reassignment to a max affine function. Essentially, we show that attention prepended with a $\mathrm{Linear}$ layer is able to reassign values to a partition generated by a max-affine function. We start with the below definition.

**Definition 3.1** (Max-Affine Function). Let $\mathcal{X} \subset \mathbb{R}^{d_x}$ be a domain, and fix a positive integer $N_{\mathrm{ma}}$. For each $i \in [N_{\mathrm{ma}}]$, define an affine function $y_i : \mathcal{X} \to \mathbb{R}$ for all $x \in \mathcal{X}$:

$$y_i(x) = a_i^\top x + b_i, \quad \text{where } a_i \in \mathbb{R}^{d_x} \text{ and } b_i \in \mathbb{R}.$$

The *max-affine function* $\mathrm{MaxAff} : \mathcal{X} \to \mathbb{R}$ corresponding to affine functions $\{y_i(\cdot)\}_{i=[N_{\mathrm{ma}}]}$ is defined as

$$\mathrm{MaxAff}(x) = \max_{i \in [N_{\mathrm{ma}}]} \{a_i^\top x + b_i\}.$$

Intuitively, a max-affine function selects, at each point $x \in \mathcal{X}$, the largest output among $N_{\mathrm{ma}}$ affine functions. Geometrically, each affine function $y_i(x) = a_i^\top x + b_i$ defines a hyperplane in $\mathbb{R}^{d_x+1}$. Thus, $\mathrm{MaxAff}$ follows the highest hyperplane at each $x$, forming a piecewise linear, convex surface — the upper envelope of the given affine hyperplanes.

**Remark 3.1** (Technical Assumption). For simplicity of presenting our interpretation, we make the following technical assumption for all results in this section:

**Assumption 3.1.** For any max-affine function $\mathrm{MaxAff}$, we exclude situations where the difference between its largest and second-largest affine components is smaller than a specified threshold. (Please see proofs for explicit definition.)

We do not apply this assumption in other sections.

### 3.1 Max-Affine Partition

We now show that a max-affine function $\mathrm{MaxAff}(\cdot)$ induces a partition of its input domain $\mathcal{X}$. Specifically, the input domain $\mathcal{X}$ is divided up according to which affine function is the maximum at each point $x$. To be concrete, we define this partition as follows:

**Proposition 3.1** (Max-Affine Partition). Following Definition 3.1, consider a max-affine function $\mathrm{MaxAff}(x) = \max_{i \in [N_{\mathrm{ma}}]} \{a_i^\top x + b_i\}$, and let $\mathcal{X} \subset \mathbb{R}^{d_x}$ be its input domain. Then $\mathrm{MaxAff}$ generates a partition on $\mathcal{X}$:

$$P_{\mathrm{ma}} := \{U_i \mid i \in [N_{\mathrm{ma}}]\}, \quad U_i := \{x \in \mathcal{X} \mid \mathrm{MaxAff}(x) = a_i^\top x + b_i\}, \quad i \in [N_{\mathrm{ma}}].$$

We call the partition $P_{\mathrm{ma}}$ the *max-affine partition* of $\mathcal{X}$ induced by $\mathrm{MaxAff}$.

Intuitively, $U_i$ is the set of all point $x$ for which the $i$-th affine function $a_i^\top x + b_i$ achieves the same value as the max-affine output. Since $\mathrm{MaxAff}(\cdot)$ is the *maximum* of all the affine components, the $i$-th component is (one of) the highest among all components. Hence, the input domain $\mathcal{X}$ becomes partitioned "regions" $\{U_i\}_{i=[N_{\mathrm{ma}}]}$. That is, if a point $x$ belongs to a region $U_i$, the corresponding affine function $a_i^\top x + b_i$ is (tied for) the largest. Please see Appendix D.1 for a detailed proof.

**Set Overlaps and Boundaries.** By construction, every $x \in \mathcal{X}$ lies in at least one of the sets $\{U_i\}$, but it may belong to multiple sets if several affine components attain the same maximal value. Hence, the collection $\{U_i\}$ is generally a "partition" in an informal sense: while each $U_i$ is typically associated with a distinct region, their pairwise intersections are non-empty on boundary hyperplanes. We address these overlaps in detail within our theorems, where boundary regions do not affect the main approximation arguments but require careful handling to ensure mathematical rigor.

**Indicator Encoding of the Partition.** For certain analytical and algorithmic tasks, it is helpful to embed the notion of "which affine part is active" into a vector-valued indicator. Formally, we define the indicators for max-affine partitions.

**Definition 3.2** (Indicator of Max-Affine Partition). Following the notations in Proposition 3.1, for a max-affine partition $\{U_i | i \in [N_{\mathrm{ma}}]\}$, we define $i_x := \mathrm{argmax}_{i \in N_{\mathrm{ma}}}(y_i(x))$ to be the label of the maximal affine component. Then, we define the indicator $E : \mathbb{R}^{d_x} \to \mathbb{R}^{N_{\mathrm{ma}}}$ as:

$$E(x) = e_{i_x}^{(N_{\mathrm{ma}})},$$

which is a one-hot vector whose only non-zero entry is the $i_x$-th one.

Namely, each component of $E(x)$ is zero unless it corresponds to an index achieving the maximum, in which it has the value of 1. In Figure 1a, we show an example of the max-affine partition.

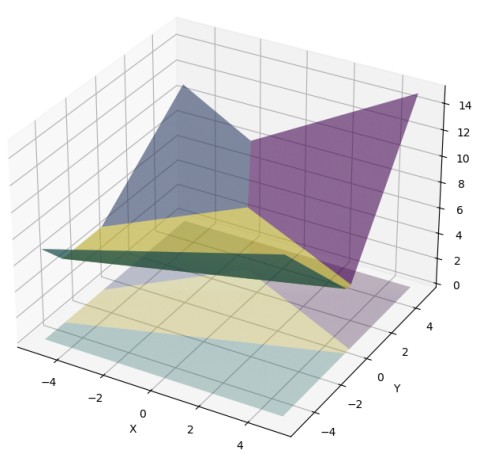
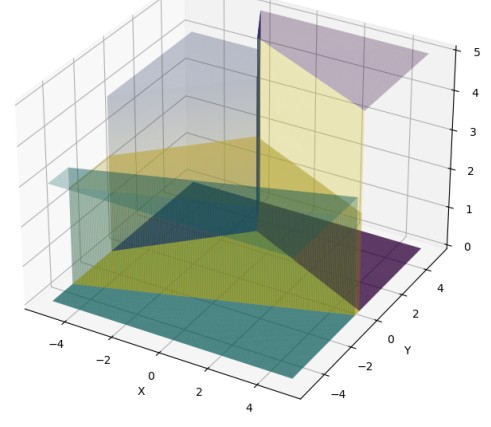

(a) **Max-Affine Partition on a 2-D Domain.** Colored regions show where each affine component is active.

(b) **Value Reassignment of Figure 1a.** Each region is reassigned a different affine function.

## 3.2 Attention Scores Encode Indicators for Max-Affine Partition

We now discuss the connection between self-attention and a max-affine partition. We show that self-attention with a Linear layer attached before it can generate a max-affine partition. Further, for every input token, the attention score matrix approximately indicates which part of the partition it belongs to. We state this result as follows:

**Proposition 3.2** (Attention Approximates Indicator of Max-Affine Partition). Let $X = [X_1, X_2, \cdots, X_n] \in \mathbb{R}^{d \times n}$ denote any input sequence. We use $\mathcal{X}$ to denote the domain of all $X_i, i \in [n]$. Let MaxAff be any max-affine function on $\mathcal{X}$ with $N_{\mathrm{ma}}$ components, and let $\epsilon > 0$ be any positive real number. We define $P_{\mathrm{ma}} = \{U_i | i \in [N_{\mathrm{ma}}]\}$ as the max-affine partition generated by MaxAff as in Proposition 3.1. Then, there exists a Linear layer and a self-attention Attn whose attention matrix satisfies:

$$\| \mathrm{Softmax}((W_K \mathrm{Linear}(X))^\top W_Q \mathrm{Linear}(X)) W_O - [E(X_1), E(X_2), \cdots, E(X_n)] \|_\infty \leq \epsilon,$$

with exception of a region of arbitrarily small Lebesgue measure in $\mathbb{R}^n$. Here $W_K, W_Q$ are the attention weights within Attn. $W_O$ only truncates the irrelevant part of the attention score matrix.

Proposition 3.2 shows that the attention matrix is able to approximate a vector denoting the position of the input token, by indicating which part of the max-affine partition contains the input token.

### 3.3 Attention Reassign Value to Each Part of the Max-Affine Partition

In the work of [Kim and Kim, 2022], they prove that max-affine functions are universal approximators for convex functions. In order to turn them into universal approximators, a possible solution is to reassign value to each part of the max-affine partition generated by the original max-affine function. In the following theorem, we show that a single-head self-attention is capable of completing this task.

**Proposition 3.3** (Attention Reassigns Value to Max-Affine Partition). Following the notation in Proposition 3.2, Let $F : \mathbb{R}^d \to \mathbb{R}^d_{\text{out}}$ be a piece-wise constant function which is separately constant on each $U_i, i \in [N_{\text{ma}}]$. We show that for any $\epsilon > 0$, there exists an self-attention $\text{Attn}$ such that

$$\|\text{Attn}(X) - [F(X_1), F(X_2), \cdots, F(X_n)]\|_\infty \leq \epsilon,$$

for every $X$ in $\mathcal{X}$ with exception of a region of arbitrarily small Lebesgue measure in $\mathbb{R}^n$.

Proposition 3.3 shows that attention is able to output different values according to the indicator generated in Proposition 3.2.

We conclude this section with two remarks.

**Remark 3.2** (Extension to Function on All Tokens). In this section, for the conciseness in demonstration of method, we adopted a token-wise function $F$ as the example function. Yet since affine functions on all tokens can be easily obtained by adding token-wise affine functions, this simplified version of our method generalizes well on functions taking all tokens as input and leads us to results shown in Section 4.

**Remark 3.3.** Lastly, we emphasize that here the approximation excludes a small area for overall simplicity in this demonstration of our method. We address this issue in the proofs of the universal approximation theorems in the next section.

Figure 1b provides us an example of Proposition 3.3.

## 4 Single-Layer, Single-Head Attention Achieves Universal Sequence-to-Sequence Approximation

In this section, we present our main results:

- A single layer of *single-head self-attention* preceded by one linear layer is a sequence-to-sequence universal approximator for continuous functions on any compact domain.

- A single layer of *single-head cross-attention* preceded by one linear layer is likewise a sequence-to-sequence universal approximator for continuous functions on any compact domain.

Importantly, we achieve attention-only universal approximation for both the $L_p$-norm and $L_\infty$-norm, whereas most existing results apply only to the $L_p$-norm and require additional auxiliary components in the transformer block (e.g., multiple attention or feed-forward layers). Moreover, our universality result for cross-attention is the first of its kind. Specifically, we present our results for self-attention in Section 4.1 and for cross-attention in Section 4.2.

### 4.1 Single-Head Self-Attention as a Universal Seq-to-Seq Approximator

We now present our main result: a single-layer, single-head self-attention module, combined with a linear transformation, is sufficient to approximate any continuous map $f : \mathbb{R}^{d \times n} \to \mathbb{R}^{d \times n}$ on a compact domain $U \subseteq [-D, D]^{d \times n}$. We present the result first in terms of the $L_\infty$ norm for continuous $f$ and then extend it to $L_p$ integrable functions.

**Theorem 4.1** ($L_\infty$-Norm Universal Approximation). Let $f : \mathbb{R}^{d \times n} \to \mathbb{R}^{d \times n}$ denote any continuous function on a compact domain $U \subset \mathbb{R}^{d \times n}$ and let $\epsilon > 0$ be any positive real number. There exists a self-attention $\mathrm{Attn}$ with a prepended $\mathrm{Linear}$ layer, such that

$$\|f - \mathrm{Attn} \circ \mathrm{Linear}\|_{L_\infty} \le \epsilon.$$

Theorem 4.1 indicates that a *single-layer* self-attention block, combined with a linear preprocessing layer $\mathrm{Linear}$, approximates sequence-to-sequence $f$ in the $L_\infty$-norm.

**Overview of Proof Strategy.** We adopt a proof strategy based on a key observation: self-attention is capable of approximating target functions via implicit $\mathrm{MaxAff}$ operations. Our proof consists of the following 4 steps:

- **Step 1: Partition Input Domain $U$ via $\mathrm{MaxAff}$.** Construct a max-affine function $\mathrm{MaxAff}$ over $U$ (i.e., input domain of target function $f$) such that this $\mathrm{MaxAff}$ induces a partition of size-$N_{\mathrm{ma}}$ of $U$.

- **Step 2: Configure $\mathrm{Linear}$ and $\mathrm{Attn}$ to Imitate $\mathrm{MaxAff}$ over $U$.** Use $\mathrm{Linear}$ and $W_K, W_Q$ in $\mathrm{Attn}$ to map the input $Z \in U$ to values of the affine components $\{y_i(Z) = a_i^\top \widetilde{Z} + b_i\}_{i \in [N_{\mathrm{ma}}]}$ of $\mathrm{MaxAff}$. Here we flatten the input sequence $Z \in \mathbb{R}^{d \times n}$ to $\widetilde{Z} \in \mathbb{R}^{dn}$ to compute $\mathrm{MaxAff}$.

- **Step 3: Engineer $\mathrm{Attn}$ to Generate an Indicator of Which Partition Cell the Input Belongs To.** Within self-attention $\mathrm{Attn}$, design $K^\top Q$ so that $\mathrm{Softmax}(K^\top Q)$ produces a near-one-hot vector as an indicator to the max-affine partition induced by $\mathrm{MaxAff}$ (as defined in Definition 3.2). This indicator (approximately an one-hot vector) shows which part (i.e., partitioned cell) of the partition contains the input sequence $Z$.

- **Step 4: Map the Indicator to the Target Value $f(Z)$.** Map each partition cell's indicator to the corresponding value of $f$. By continuity of $f$, refining the partitioned cell ensures $\|f - \mathrm{Attn} \circ \mathrm{Linear}\|_\infty \le \epsilon$.

*Proof Sketch.* We elaborate above in detail. Consider a continuous function $f : U \subseteq [-D, D]^{d \times n} \to \mathbb{R}^{d \times n}$ on a compact domain $U$. Let $\epsilon > 0$. We aim to construct a *single-layer, single-head* self-attention mechanism $\mathrm{Attn}$ (prepended with a linear transformation $\mathrm{Linear}$) such that

$$\|f - \mathrm{Attn} \circ \mathrm{Linear}\|_{L_\infty} \le \epsilon.$$

**Step 1: Partition Input Domain $U$ via $\mathrm{MaxAff}$.**

- **Flattening Input.** Each input $Z \in \mathbb{R}^{d \times n}$ is reshaped into a single vector $\widetilde{Z} \in \mathbb{R}^{dn}$ by stacking its rows or columns. This unifies the domain as $\widetilde{Z} \in [-D, D]^{dn}$.

- **Grid / Max-Affine Construction.** Since $f$ is uniformly continuous on the compact set $U$, choose $\delta > 0$ such that

$$\|Z_1 - Z_2\|_\infty < \delta \implies \|f(Z_1) - f(Z_2)\|_\infty < \epsilon.$$

  We subdivide $[-D, D]^{dn}$ into cubes of side $\le \delta$, yielding $G = P^{dn}$ grid centers $\{v_j\}_{j=0}^{G-1}$. We treat $\mathrm{MaxAff}$ as a piecewise (max-)affine or piecewise-constant partition: for each $\widetilde{Z}$, there's a nearest $v_j$ within $\delta/2$.

- **Technical Highlight.** This partition-based approach leverages uniform continuity to discretize $U$. The number of partitions can be large but finite, ensuring we only need a single-layer of attention to "select" the correct grid cell.

**Step 2: Configure $\mathrm{Linear}$ and $\mathrm{Attn}$ to Imitate $\mathrm{MaxAff}$ over $U$.**

- **Sum-of-Linear-Transformations Map $\mathrm{Linear}$.** Design $\mathrm{Linear} : \mathbb{R}^{d \times n} \to \mathbb{R}^M$ (for some dimension $M$) to capture the dot products $\langle v_j, \widetilde{Z} \rangle$. Essentially, $\mathrm{Linear}(Z)$ arranges these

$\{v_j^\top \widetilde{Z}\}$ in a form accessible to attention. This ensures each grid center $v_j$ can be individually "queried."

- **Encoding Affine Components.** Observe that $\max_j\{\langle v_j, \widetilde{Z}\rangle - \frac{1}{2}\|v_j\|^2\}$ is akin to a max-affine function. We store terms $v_j^\top \widetilde{Z}$, plus $-\frac{1}{2}\|v_j\|^2$, into $K$ and $Q$ for later use in $\mathrm{Softmax}(K^\top Q)$.

- **Technical Highlight.** This step demonstrates how we embed $\{\langle v_j, \widetilde{Z}\rangle\}$ into a single-head attention setting — no extra feed-forward layers required. The linear map $\mathrm{Linear}$ is carefully constructed so that each "component" is individually addressable.

**Step 3: Engineer $\mathrm{Attn}$ to Generate an Indicator of Which Partition Cell the Input Belongs to.**

- **Construct $K^\top Q$.** In the self-attention block, let $K^\top Q \approx R(\langle v_j, \widetilde{Z}\rangle - \frac{1}{2}\|v_j\|^2)$, where $R > 0$ is large. This makes $\mathrm{Softmax}(K^\top Q)$ favor the row $j^*$ maximizing

$$\langle v_j, \widetilde{Z}\rangle - \tfrac{1}{2}\|v_j\|^2.$$

- **Near-One-Hot Distribution.** Hence the $j^*$-th row obtains probability close to 1, effectively identifying which grid center $v_{j^*}$ is nearest to $\widetilde{Z}$. We interpret this as a near-one-hot "indicator" vector for the correct partition cell.

- **Technical Highlight.** This is the crux: attention's softmax can act as a *continuous* $\arg\max$ by scaling the scores with $R$. As $R \to \infty$, the distribution becomes more peaked, approximating a hard partition.

**Step 4: Map the Indicator to the Target Value $f(Z)$.**

- **Assigning Values.** We place $f(v_j)$ in the "value matrix" $W_V$, so that once row $j^*$ is selected, the attention output is $\approx f(v_{j^*})$. Since $Z$ is within $\delta/2$ of $v_{j^*}$, uniform continuity implies

$$\|f(Z) - f(v_{j^*})\| \le \epsilon, \quad \text{(for suitably chosen } \delta\text{)}.$$

- **Final Reshaping (If Needed).** A small linear projection $M$ can reshape the output back to $\mathbb{R}^{d\times n}$. The essential logic is that the correct $f(v_j)$ is "routed" to the final output via the near-one-hot attention distribution.

- **Technical Highlight.** This reveals how a single-head attention layer, armed with linear preprocessing, suffices to replicate the entire function $f$. No feed-forward sub-layer or multiple heads are needed to achieve universal approximation.

In sum, combining these steps, we see that: (i) A finite grid subdivides $U$ to handle uniform continuity. (ii) Linear encodes $\{\langle v_j, \widetilde{Z}\rangle\}$. (iii) Large-$R$ $\mathrm{Softmax}(K^\top Q)$ selects the best anchor $v_{j^*}$. (iv) A "value matrix" translates that selection into $f(v_{j^*})$. We conclude that a single-layer, single-head self-attention block approximates $f$ within $\epsilon$ in the $L_\infty$ norm. Please see Appendix E.1 for a proof. $\square$

Our result in $L_\infty$ norm can be easily extended to $L_p$ norm, where it applies to not just the continuous functions but all Lebesgue integrable functions with compact support. Please see Corollary E.1.1 for more details.

### 4.2 Single-Head Cross-Attention as a Universal Seq-to-Seq Approximator

Here we extend self-attention universal approximation results from Section 4.1 to cross-attention. Importantly, we establish the first known universal approximation in cross-attention setting. First, we state our main result in $L_\infty$-norm.

**Theorem 4.2** ($L_\infty$-Norm Universal Approximation). Let $f : U_K \times U_Q \to \mathbb{R}^{d\times n}$ denote any continuous function on a compact domain $U_K \times U_Q$ and let $\epsilon$ be any positive real number. Here $U_K, U_Q \in \mathbb{R}^{d\times n}$ stands for the compact domain of the two input sequences of cross-attention. Then

there exists a cross-attention $\mathrm{Attn}$ prepended with a $\mathrm{Linear}$ layer such that

$$\|f - \mathrm{Attn} \circ \mathrm{Linear}\|_{L_\infty} \leq \epsilon.$$

Theorem 4.2 indicates that a *single-layer cross*-attention block, prepended with a linear preprocessing layer $\mathrm{Linear}$, approximates $f : U_K \to U_Q \to \mathbb{R}^{d \times n}$ in $L_\infty$-norm.

*Proof Sketch.* Our proof follows that of Theorem 4.1 except one additional step: use $\mathrm{Attn}$ to aggregate the max-affine functions on $U_K, U_Q$ and merge into a $\mathrm{MaxAff}$ function on $U_K \times U_Q$. The proof consists of the following steps:

**Step 1: Partition the Input Domain $U_K$ and $U_Q$ with $\mathrm{MaxAff}_K$ and $\mathrm{MaxAff}_Q$ Respectively.** Construct two max-affine function $\mathrm{MaxAff}_K$ over $U_K$ and $\mathrm{MaxAff}_Q$ over $U_Q$ such that this $\mathrm{MaxAff}_K$ induces a partition of size-$N_{\mathrm{ma}}$ of $U$ and $\mathrm{MaxAff}_Q$ a same size partition on $U_Q$.

**Step 2: Configure $\mathrm{Linear}$ and $\mathrm{Attn}$ to Imitate $\mathrm{MaxAff}_K, \mathrm{MaxAff}_Q$ over $W_K, U_Q$ Respectively.** Use $\mathrm{Linear}$ and $W_K, W_Q$ in $\mathrm{Attn}$ to map the input $Z_K, Z_Q \in U$ to values of the affine components $\{y_i(Z) = a_i^\top \widetilde{Z} + b_i\}_{i \in [N_{\mathrm{ma}}]}$ of $\mathrm{MaxAff}_K$ and $\mathrm{MaxAff}_Q$ respectively. Here we flatten the input sequence $Z \in \mathbb{R}^{d \times n}$ to $\widetilde{Z} \in \mathbb{R}^{dn}$ to express $\mathrm{MaxAff}$ concisely.

**Step 3: Use $\mathrm{Attn}$ to Aggregate $\mathrm{MaxAff}_K$ and $\mathrm{MaxAff}_Q$ to Form a $\mathrm{MaxAff} : U_K \times U_Q \to \mathbb{R}$ on Both Input Sequences.** Use $\mathrm{Attn}$ to generate $\mathrm{MaxAff}(Z_K, Z_Q) := \mathrm{MaxAff}_K(Z_K) + \mathrm{MaxAff}_Q(Z_Q)$. This max-affine function merges the partition on $U_K$ and $U_Q$ to generate a unified partition on $U_K \times U_Q$.

**Step 4: Use $\mathrm{Attn}$ to Indicate the Position of the Both Input Sequence in the $\mathrm{MaxAff}$-Generated Partition.** Use $\mathrm{Attn}$ to generate an indicator to the max-affine partition generated by $\mathrm{MaxAff}$ (as defined in Definition 3.2). This indicator (approximately a one-hot vector) shows which part of the $\mathrm{MaxAff}$-generated partition contains the Cartesian product of both input sequences $Z_K \times Z_Q$.

**Step 5: Map the indicator to the Corresponding Value of $f$.** Map the indicator to the corresponding value of the target function $f$ by adding terms related to $f$ to $\mathrm{Attn}$.

Please see Appendix E.2 for a detailed proof. $\qquad\square$

## 5 Concluding Remarks

We introduce a novel interpretation of attention as a mechanism for reassigning values to a partition induced by a max-affine function. This unique perspective allows us to show that prepending a single linear layer before either self-attention or cross-attention enables the network to (i) generate indicator functions representing max-affine partitions (Proposition 3.2) and (ii) selectively reassign values to each partition cell (Proposition 3.3). As a result, we prove that both single-head self-attention and single-head cross-attention, when combined with a single layer of sum of linear transformations, achieve universal approximation of compactly supported continuous functions under $L_\infty$ norm, or integrable functions under $L_p$ norm. **Numerical validations** backup our theory in Appendix B.

**Key Insights and Results.**

- **Max-Affine Partition.** A max-affine function naturally partitions its input domain, and attention (with appropriate transformations) can approximate the indicator functions of these partitions.

- **Value Reassignment.** Self-attention reassigns output values based on partition indicators, capturing a broad class of piecewise-defined functions.

- **Universal Approximation.** With only a single linear layer and a single-head attention module, one can approximate arbitrary sequence-to-sequence maps in both the $L_\infty$ and $L_p$ senses, for both self-attention (Theorem 4.1 and Corollary E.1.1) and cross-attention (Theorem 4.2 and Corollary E.2.1) architectures.

**Limitations.** While our results highlight the surprising representational power of single-head attention with linear preprocessing, several limitations warrant discussion:

- **Large Dimensions and Network Size.** Our minimal-assumption design needs many partition regions to cover diverse targets. This follows naturally from the general setting we study. High-dimensional inputs or long sequences then inflate the parameter count and hinder practice. Appendix A eases the burden but does not eliminate it entirely.

- **Training Complexity.** Our proofs are *constructive* rather than *prescriptive* for training, meaning standard gradient-based methods may not (always) efficiently find the required weight configurations.

- **Data Distribution Shifts.** Like many universal approximation results, our approach does not account for distribution shifts or generalization beyond the compact domain used for training.

**Implications and Future Work.** Our findings explain why transformers excel at modeling heterogeneous data: attention can create flexible partitions of the input space and assign context-dependent outputs. This perspective raises open questions for future research: Can multi-head or deeper attention layers simplify representational requirements or reduce approximation constants? How might learned partitions or specialized positional encodings improve efficiency in practice? Can adaptive or data-driven strategies automatically discover near-optimal partitions for specific tasks?

Overall, our results establish a theoretical foundation for understanding attention-based architectures as universal function approximators. They illustrate how token-wise information is *partitioned and reassigned* to represent complex sequence-to-sequence functions with minimal assumptions and structural requirements on data and model.

# Acknowledgments

The authors thank Mimi Gallagher, Sara Sanchez, Dino Feng and Andrew Chen for useful discussions; and Weimin Wu, Hong-Yu Chen and Jennifer Zhang for collaborations on related topics. JH also thanks the Red Maple Family for support. The authors also thank the anonymous reviewers and program chairs for constructive comments.

Lastly, JH dedicates this work to the memory of his aunt, Lily Cheung, who passed away during its preparation (March 2025). Her loving and caring spirit will always inspire him.

JH is supported by the Walter P. Murphy Fellowship and the Terminal Year Fellowship (Paul K. Richter Memorial Award) of Northwestern University. HL is partially supported by NIH R01LM1372201, NSF AST-2421845, Simons Foundation MPS-AI-00010513, AbbVie , Dolby and Chan Zuckerberg Biohub Chicago Spoke Award. This research was supported in part through the computational resources and staff contributions provided for the Quest high performance computing facility at Northwestern University which is jointly supported by the Office of the Provost, the Office for Research, and Northwestern University Information Technology. The content is solely the responsibility of the authors and does not necessarily represent the official views of the funding agencies.

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

# Appendix

## Impact Statement

By the formal nature of this work, we do not expect any immediate negative social impact.

## A  Extension to Practical Settings

In practical scenarios, despite defined on a high dimension input domain ($\mathbb{R}^{d \times n}$), attention is often considered to approximate a function defined upon a small input domain $\mathcal{X} \subset \mathbb{R}^{d \times n}$.

To this end, we extend our method to the approximation rate of $L$-Lipschitz functions with a relatively small input domain. We state our result as the following theorem.

> **Theorem A.1.**  Let $f : \mathbb{R}^{d \times n} \to \mathbb{R}^{d \times n}$ denote an $L$-Lipschitz function (in terms of 2-norm) whose input domain is $\mathcal{X}$. For any $\epsilon > 0$, assume $\mathcal{X}$ is contained in $N_x$ spheres by the radius of $\epsilon/(3L)$ in 2-norm. Then, there exists a $\mathrm{Linear}$ layer and a $\mathrm{Attn}$ layer such that:
>
> $$\|\mathrm{Attn} \circ \mathrm{Linear} - f\|_\infty \le \epsilon.$$
>
> Furthermore, $\mathrm{Attn}$ and $\mathrm{Linear}$ have a total number of $\mathcal{O}(dnN_x)$ trainable parameters.

*Proof Sketch.* This proof only differs from the proof of Theorem 4.1 on the choice of partition. For universal approximation, we choose a partition that evenly partition the whole space. In this theorem, we change this partition to have each part centered on a different sphere described in the Theorem A.1. By characterizing our partition, we achieve a more precise approximation result.

Please see Appendix F.1 for a detailed proof.                                                   □

Theorem A.1 states that when the input domain is contained in $N_x$ spheres of $\epsilon$-level radius, there exists a single-head self-attention layer that approximates the target function with a precision of $\epsilon$.

# B  Proof-of-Concept Experiments

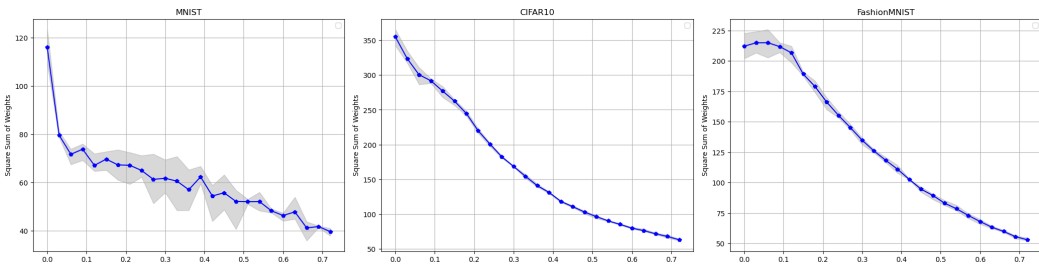

Figure 2: **Scale of Attention Weights vs. Training noise.** For MNIST, CIFAR-10, and Fashion-MNIST we plot the $\ell_2$-norm of $W_K$ and $W_Q$ against the injected label-noise ratio. In all three datasets the weight scale declines monotonically as noise increases, corroborating Proposition 3.2: higher noise hampers precise partitioning, so the model reduces the magnitude of weights that form the attention score matrix.

In Proposition 3.2, we demonstrate domain-partition mechanism of attention. In this mechanism, the temperature of the Softmax function affects the precision of the max-affine partition generated by attention, which is crucial to the complex approximations accomplished in Theorem 4.1 and Corollary E.1.1.

Since the temperature of Softmax is equivalent to the scale of the matrix involved in computing the attention score matrix $(W_K, W_Q)$, our theory suggests the scale of $W_K, W_Q$ decreases when the input data contains more noise, as a result of the rise in difficulty to form a clear partition, and an approximation based on this partition.

To verify this conjecture, we test the correlation between the scale of $W_K, W_Q$ and the noise level in the training data.

**Objectives.**   Examine the relationship between scale of matrix involved in computing the attention score matrix in attention $(W_K, W_Q)$ and the noise level (using Gaussian noise) in the dataset.

**Data.**   We perform separate experiments on the training set of the noised MNIST, CIFAR10 and FashionMNIST datasets with noise level (the coefficient multiplying the standard Gaussian noise) gradually adding from $0$ to $0.72$ by the step size of $0.03$.

**Network setups.**   Our network consists of a single-head self-attention followed by a feed-forward network. Due to the complexity and different characteristics of the selected datasets, the size of the feed-forward network slightly differs between datasets.

**Results.**   Figure 2 presents our results. As the noise level increases, a decrease in the scale of weights in $W_K, W_Q$ becomes evident in all settings. This aligns with our theory.

# C  Additional Experimental Results

In this section, we present additional experimental results to support our theoretical results.

## C.1   Numerical Justifications for Theoretical Results in Section 3

To validate our results in Proposition 3.2, we conducted the following experiment to examine whether the max-affine function generated within the attention of the form in Proposition 3.2 can learn to separate the input domain according to the values of the target function.

Specifically, we use attention to approximate a step function and observe the max-affine function generated by the weights in $K$ and $Q$ matrices in the attention score matrix. The result of this experiment is shown in Figure 3.

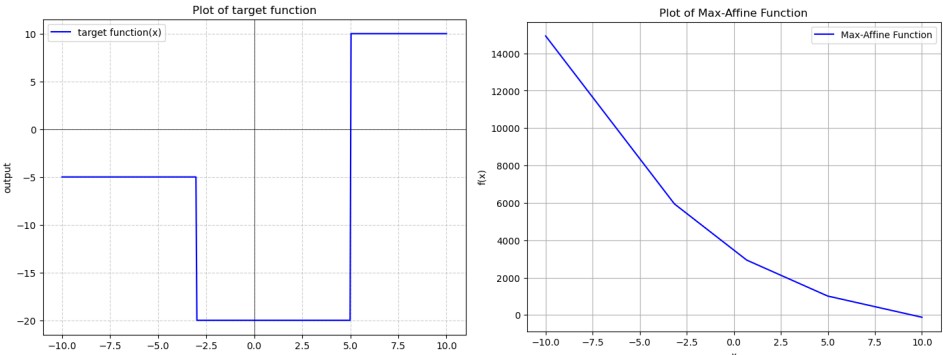

Figure 3: Result of using a single-head attention to approximate a step function. The max-affine function generated in the attention score matrix turns at points close to the switching points in the step function.

The max-affine function generated in the attention score matrix turns at points close to the switching points in the step function. This generates a partition in the input domain that resembles the distribution of the flat parts in the step function. This result aligns with our theory.

# D  Proofs of Results in Section 3

## D.1  Proof of Proposition 3.1

**Proposition D.1** (Proposition 3.1 Restated: Max-Affine Partition). Following Definition 3.1, consider a max-affine function $\text{MaxAff}(x) = \max_{i \in [N_{\text{ma}}]}\{a_i^\top x + b_i\}$, and let $\mathcal{X} \subset \mathbb{R}^{d_x}$ be its input domain. Then $\text{MaxAff}$ generates a partition on $\mathcal{X}$:

$$P_{\text{ma}} := \{U_i \mid i \in [N_{\text{ma}}]\},$$
$$U_i := \{x \in \mathcal{X} \mid \text{MaxAff}(x) = a_i^\top x + b_i\}, \quad i \in [N_{\text{ma}}].$$

We call the partition $P_{\text{ma}}$ the *max-affine partition* of $\mathcal{X}$ induced by $\text{MaxAff}$.

*Proof.* If an $x_0$ is not grouped to any $U_i$, $i \in [N_{\text{MaxAff}}]$. Since $\text{MaxAff}$ is define over $\mathcal{X}$ and thus defined on $x_0$, we have:

$$\text{MaxAff}(x_0) \neq a_i^\top x_0 + b_i, \quad i \in [N_{\text{MaxAff}}].$$

This is contradictory to the definition of $\text{MaxAff}$.

Since in Section 3 we exclude the discussion on the overlapped regions of the affine components $\{y_i = a_i^\top x + b_i\}$, $\{U_i \mid i \in [N_{\text{MaxAff}}]\}$ form a partition on $\mathcal{X}$. This completes the proof. $\square$

## D.2  Proof of Proposition 3.2

**Proposition D.2** (Proposition 3.2 Restated: Attention Approximates Indicator of Max-Affine Partition). Let $X = [X_1, X_2, \cdots, X_n] \in \mathbb{R}^{d \times n}$ denote any input sequence. We use $\mathcal{X}$ to denote the domain of all $X_i$, $i \in [n]$. Let $\text{MaxAff}$ be any max-affine function on $\mathcal{X}$ with $N_{\text{MaxAff}}$ components, and let $\epsilon > 0$ be any positive real number. We define $P_{\text{MaxAff}} = \{U_i \mid i \in [N_{\text{MaxAff}}]\}$ as the max-affine partition generated by $\text{MaxAff}$ as in Proposition 3.1. Let $E$ be the indicator of $P_{\text{MaxAff}}$ as defined in Definition 3.2. Under the above definitions, there exists a $\text{Linear}$ layer and a self-attention $\text{Attn}$ whose attention matrix satisfies

$$\| \text{Softmax}((W_K \text{Linear}(X))^\top W_Q \text{Linear}(X))W_O - [E(X_1), E(X_2), \cdots, E(X_n)]\|_\infty \leq \epsilon,$$

with exception of an arbitrarily small region. Here $W_K$, $W_Q$ are the attention weights within $\text{Attn}$.

*Proof.* We first denote that according to the premise of Section 3, the intersection region of different affine components are omitted. This means for an arbitrarily small $\delta > 0$, this proposition malfunctions on any points within a $\delta$ radius neighborhood of the intersecting lines of max-affine partitions.

Our proof consists of two parts:

1. Construct $\text{Linear}$ and $\text{Attn}$.

2. Estimate the error between the attention score matrix of $\text{Attn} \circ \text{Linear}$ and the target indicator.

For the max-affine function $\text{MaxAff}$, we denote it as follows.

**Definition D.1** (Max-Affine Function). Let $a_i \in \mathbb{R}^d, b_i \in \mathbb{R}, i \in [N_{\text{MaxAff}}]$ denote the coefficients of the affine components of $\text{MaxAff}$. In this definition, $\text{MaxAff}$ writes out as

$$\text{MaxAff}(Z) = \max_{i \in [N_{\text{MaxAff}}]}\{a_i^\top Z + b_i\}, \tag{D.1}$$

for any $Z \in \mathbb{R}^d$.

**Remark D.1.** For conciseness of presentation, we assume the top component of $\text{MaxAff}$ exceeds the second-largest by a fixed $\Delta > 0$, independent of the input and arbitrarily small.

**Construction of** Linear. Without loss of generality, assume $N_{\text{MaxAff}} \geq n$. We construct Linear (the layer of linear transformations) to be

$$\text{Linear}(Z) := \begin{bmatrix} I_d \\ 0_{n \times d} \end{bmatrix} Z \begin{bmatrix} I_n & 0_{n \times (N_{\text{MaxAff}} - n)} \end{bmatrix} + \begin{bmatrix} 0_{d \times N_{\text{MaxAff}}} \\ I_{\text{MaxAff}} \end{bmatrix}.$$

The the output of $\text{Linear}(X)$ is

$$
\begin{aligned}
\text{Linear}(X) &= \begin{bmatrix} I_d \\ 0_{n \times d} \end{bmatrix} X \begin{bmatrix} I_n & 0_{n \times (N_{\text{MaxAff}} - n)} \end{bmatrix} + \begin{bmatrix} 0_{d \times N_{\text{MaxAff}}} \\ I_{\text{MaxAff}} \end{bmatrix} \\
&= \begin{bmatrix} X & 0_{d \times (N_{\text{MaxAff}} - n)} \\ 0_{n \times n} & 0_{n \times (N_{\text{MaxAff}} - n)} \end{bmatrix} + \begin{bmatrix} 0_{d \times N_{\text{MaxAff}}} \\ I_{N_{\text{MaxAff}}} \end{bmatrix} \\
&= \begin{bmatrix} X & 0_{d \times (N_{\text{MaxAff}} - n)} \\ I_n & 0_{n \times (N_{\text{MaxAff}} - n)} \\ 0_{(N_{\text{MaxAff}} - n) \times n} & I_{N_{\text{MaxAff}} - n} \end{bmatrix}.
\end{aligned}
\tag{D.2}
$$

**Construction of** Attn. Since we only use the attention score matrix $\text{Softmax}(K^\top Q)$, we only have to construct the $W_K$ and $W_Q$ matrices.

We construct them to be as follows

$$W_K = R \begin{bmatrix} 0_{d \times d} & a_1 & a_2 & \cdots & a_{N_{\text{MaxAff}}} \\ 0 & b_1 & b_2 & \cdots & b_{N_{\text{MaxAff}}} \end{bmatrix}$$

$$W_Q = \begin{bmatrix} I_d & 0_{1 \times d} & 0_{1 \times N_{\text{MaxAff}} - d} \\ 0_{1 \times d} & 1_{1 \times d} & 0_{1 \times N_{\text{MaxAff}} - d} \end{bmatrix},$$

where $R$ is a coefficient to control the precision of the approximation. Specifically, as $R$ increases, $\text{Softmax}$ is closer to maximum function, and the approximation is more precise.

In this construction, we now calculate the $K$ and $Q$ matrices of attention

$$
\begin{aligned}
K &= W_K \text{Linear}(X) \\
&= R \begin{bmatrix} 0_{d \times d} & a_1 & a_2 & \cdots & a_{N_{\text{MaxAff}}} \\ 0 & b_1 & b_2 & \cdots & b_{N_{\text{MaxAff}}} \end{bmatrix} \cdot \begin{bmatrix} X & 0_{d \times (N_{\text{MaxAff}} - n)} \\ I_n & 0_{n \times (N_{\text{MaxAff}} - n)} \\ 0_{(N_{\text{MaxAff}} - n) \times n} & I_{N_{\text{MaxAff}} - n} \end{bmatrix} \quad \left(\text{By (D.2)}\right) \\
&= R \begin{bmatrix} a_1 & a_2 & \cdots & a_{N_{\text{MaxAff}}} \\ b_1 & b_2 & \cdots & b_{N_{\text{MaxAff}}} \end{bmatrix},
\end{aligned}
$$

and

$$
\begin{aligned}
Q &= W_Q \text{Linear}(X) \\
&= \begin{bmatrix} I_d & 0_{1 \times d} & 0_{1 \times N_{\text{MaxAff}} - d} \\ 0_{1 \times d} & 1_{1 \times d} & 0_{1 \times N_{\text{MaxAff}} - d} \end{bmatrix} \cdot \begin{bmatrix} X & 0_{d \times (N_{\text{MaxAff}} - n)} \\ I_n & 0_{n \times (N_{\text{MaxAff}} - n)} \\ 0_{(N_{\text{MaxAff}} - n) \times n} & I_{N_{\text{MaxAff}} - n} \end{bmatrix} \quad \left(\text{By (D.2)}\right) \\
&= \begin{bmatrix} X \cdot I_d & 0_{d \times N_{\text{MaxAff}}} \\ 1_{1 \times d} & 0_{1 \times N_{\text{MaxAff}}} \end{bmatrix} \\
&= \begin{bmatrix} X & 0_{d \times N_{\text{MaxAff}}} \\ 1_{1 \times d} & 0_{1 \times N_{\text{MaxAff}}} \end{bmatrix}.
\end{aligned}
$$

**Calculation of** $\text{Softmax}(K^\top Q)$. We now calculate the attention score matrix as

$$
\begin{aligned}
&\text{Softmax}(K^\top Q) \\
&= \text{Softmax}\left( R \begin{bmatrix} a_1 & a_2 & \cdots & a_{N_{\text{MaxAff}}} \\ b_1 & b_2 & \cdots & b_{N_{\text{MaxAff}}} \end{bmatrix}^\top \begin{bmatrix} X & 0_{d \times N_{\text{MaxAff}}} \\ 1_{1 \times d} & 0_{1 \times N_{\text{MaxAff}}} \end{bmatrix} \right) \\
&= \text{Softmax}\left( R \begin{bmatrix} a_1^\top & b_1 \\ a_2^\top & b_2 \\ \vdots & \vdots \\ a_{N_{\text{MaxAff}}}^\top & b_{N_{\text{MaxAff}}} \end{bmatrix} \cdot \begin{bmatrix} X & 0_{d \times N_{\text{MaxAff}}} \\ 1_{1 \times d} & 0_{1 \times N_{\text{MaxAff}}} \end{bmatrix} \right)
\end{aligned}
$$

$$= \text{Softmax}\left( R \begin{bmatrix} a_1^\top x_1 + b_1 & \cdots & a_1^\top x_n + b_1 & 0_{1\times(N_{\text{MaxAff}}-d)} \\ a_2^\top x_1 + b_2 & \cdots & a_2^\top x_n + b_2 & 0_{1\times(N_{\text{MaxAff}}-d)} \\ \vdots & \ddots & \vdots & \vdots \\ a_{N_{\text{MaxAff}}}^\top x_1 + b_{N_{\text{MaxAff}}} & \cdots & a_{N_{\text{MaxAff}}}^\top x_n + b_{N_{\text{MaxAff}}} & 0_{1\times(N_{\text{MaxAff}}-d)} \end{bmatrix} \right).$$

**Estimation of Approximation Error.** For $i \in [n]$, we have

$$\text{Softmax}\left(K^\top Q\right)_{:,i} = \text{Softmax}\left( R \begin{bmatrix} a_1^\top x_i + b_1 \\ a_2^\top x_i + b_2 \\ \vdots \\ a_{N_{\text{MaxAff}}}^\top x_i + b_{N_{\text{MaxAff}}} \end{bmatrix} \right)$$

$$= \frac{1}{\sum_{\eta=1}^{N_{\text{MaxAff}}} \exp\left(Ra_\eta^\top x_i + Rb_\eta\right)} \begin{bmatrix} \exp\left(Ra_1^\top x_i + Rb_1\right) \\ \exp\left(Ra_2^\top x_i + Rb_2\right) \\ \vdots \\ \exp\left(Ra_{N_{\text{MaxAff}}}^\top x_i + Rb_{N_{\text{MaxAff}}}\right) \end{bmatrix}.$$

This yields the entry on the $k$-th row of $\text{Softmax}\, K^\top Q_{:,i}$ to be

$$\text{Softmax}\left(K^\top Q\right)_{k,i} = \frac{\exp\left(Ra_k^\top x_i + Rb_k\right)}{\sum_{\eta=1}^{N_{\text{MaxAff}}} \exp\left(Ra_\eta^\top x_i + Rb_\eta\right)}.$$

When $a_k^\top x_i + b_k$ is the maximal affine component and $a_{k'}^\top x_i + b_{k'}$ is the second largest, we have

$$\text{Softmax}\left(K^\top Q\right)_{k,i} = 1 - \frac{\sum_{\eta\in[N_{\text{MaxAff}}],\eta\neq k} \exp\left(Ra_\eta^\top x_i + Rb_\eta\right)}{\sum_{\eta=1}^{N_{\text{MaxAff}}} \exp\left(Ra_\eta^\top x_i + Rb_\eta\right)}$$

$$\geq 1 - \frac{\sum_{\eta\in[N_{\text{MaxAff}}],\eta\neq k} \exp\left(Ra_\eta^\top x_i + Rb_\eta\right)}{\sum_{\eta=1}^{N_{\text{MaxAff}}} \exp\left(Ra_k^\top x_i + Rb_k\right)}$$

$$\geq 1 - (N_{\text{MaxAff}} - 1)\frac{\exp\left(Ra_{k'}^\top x_i + Rb_{k'}\right)}{\exp\left(Ra_k^\top x_i + Rb_k\right)}$$

$$= 1 - \frac{N_{\text{MaxAff}} - 1}{\exp\left(Ra_k^\top x_i + Rb_k - (Ra_{k'}^\top x_i + Rb_{k'})\right)}$$

$$\geq 1 - \frac{N_{\text{MaxAff}} - 1}{\exp(R\Delta)}.$$

Thus when

$$R \geq \Delta \cdot (\ln(N_{\text{MaxAff}} - 1) - \ln \epsilon),$$

we have

$$\frac{N_{\text{MaxAff}} - 1}{\exp(R\Delta)} \leq \epsilon,$$

which means

$$\text{Softmax}\, K^\top Q_{k,i} \geq 1 - \epsilon. \tag{D.3}$$

Moreover, since the sum of all entries in $\text{Softmax}\, K^\top Q_{:,i}$ is 1, we have

$$\text{Softmax}\left(K^\top Q\right)_{h,i} \leq 1 - \text{Softmax}\, K^\top Q_{k,i} \leq 1 - (1-\epsilon) = \epsilon, \quad h \neq k. \tag{D.4}$$

(D.3) and (D.3) are equivalent to

$$\| \text{Softmax}\, K^\top Q_{k,i} - 1\|_\infty \leq \epsilon$$

$$\| \operatorname{Softmax} K^\top Q_{h,i} - 0\|_\infty \le \epsilon, \quad h \ne k.$$

This yields

$$\|\operatorname{Softmax}\left(K^\top Q\right)_{:,i} - E(X_i)\|_\infty \le \epsilon.$$

Thus, by the nature of $\|\cdot\|_\infty$,

$$\|\operatorname{Softmax}\left(K^\top Q\right)_{:,i} - [E(X_1), E(X_2), \cdots, E(X_n)]\|_\infty \le \epsilon.$$

We construct $W_O$ to discard $\operatorname{Softmax} K^\top Q_{n+1:N_{\mathrm{MaxAff}},i}$ in $\operatorname{Softmax} K^\top Q$:

$$\begin{bmatrix} I_n \\ 0_{(N_{\mathrm{MaxAff}}-n)\times n} \end{bmatrix}.$$

Thus

$$\|\operatorname{Softmax}\left(K^\top Q\right) W_O - [E(X_1), E(X_2), \cdots, E(X_n)]\|_\infty$$
$$= \|\operatorname{Softmax}\left(K^\top Q\right)_{1:n,i} - [E(X_1), E(X_2), \cdots, E(X_n)]\|_\infty$$
$$\le \epsilon.$$

This completes the proof. $\qquad\square$

### D.3 Proof of Proposition 3.3

**Proposition D.3** (Proposition 3.3 Restated: Attention Reassigns Value to Max-Affine Partition)**.** Following the notation in Proposition 3.2, let $F : \mathbb{R}^d \to \mathbb{R}^d_{\mathrm{out}}$ be a piece-wise constant function which is separately constant on each $U_i, i \in [N_{\mathrm{MaxAff}}]$. We show that for any $\epsilon > 0$, there exists an self-attention $\operatorname{Attn}$ such that

$$\|\operatorname{Attn}(X) - [F(X_1), F(X_2), \cdots, F(X_n)]\|_\infty \le \epsilon,$$

for every $X$ in $\mathcal{X}$ with exception of a region of arbitrarily small Lebesgue measure in $\mathbb{R}^n$.

*Proof.* Let Linear and the $W_K$, $W_Q$ and $W_O$ matrices be the same as in Appendix D.2. Then by Appendix D.2, we have

$$\|\operatorname{Softmax}\left(K^\top Q\right) W_O - [E(X_1), E(X_2), \cdots, E(X_n)]\|_\infty \le \epsilon_0,$$

for any $\epsilon_0 > 0$.

Let $V_i$ denote the value of $F$ on $U_i$.

**Construction of $W_V$.** We construct $W_V$ to be

$$W_V := \begin{bmatrix} 0_{1\times d} & V_1 & V_2 & \cdots & V_{N_{\mathrm{MaxAff}}} \end{bmatrix}.$$

Thus $V$ equals to

$$V := W_V \operatorname{Linear}(X)$$

$$= \begin{bmatrix} 0_{1\times d} & V_1 & V_2 & \cdots & V_{N_{\mathrm{MaxAff}}} \end{bmatrix} \begin{bmatrix} X & 0_{d\times(N_{\mathrm{MaxAff}}-n)} \\ I_n & 0_{n\times(N_{\mathrm{MaxAff}}-n)} \\ 0_{(N_{\mathrm{MaxAff}}-n)\times n} & I_{N_{\mathrm{MaxAff}}-n} \end{bmatrix}$$

$$= \begin{bmatrix} V_1 & V_2 & \cdots & V_{N_{\mathrm{MaxAff}}} \end{bmatrix}.$$

Thus we have

$$\|V\operatorname{Softmax}\left(K^\top Q\right) W_O - [F(X_1), F(X_2), \cdots, F(X_n)]\|_\infty$$
$$= \|\begin{bmatrix} V_1 & V_2 & \cdots & V_{N_{\mathrm{MaxAff}}} \end{bmatrix} \operatorname{Softmax}\left(K^\top Q\right) W_O - [F(X_1), F(X_2), \cdots, F(X_n)]\|_\infty$$
$$= \|\begin{bmatrix} V_1 & V_2 & \cdots & V_{N_{\mathrm{MaxAff}}} \end{bmatrix} \operatorname{Softmax}\left(K^\top Q\right) W_O - \begin{bmatrix} V_1 & V_2 & \cdots & V_{N_{\mathrm{MaxAff}}} \end{bmatrix} [E(X_1), E(X_2), \cdots, E(X_n)]\|_\infty$$
$$\le \|V\|_\infty \epsilon_0.$$

Let $\|V\|_\infty \epsilon_0 \le \epsilon$ yields the final result. This completes the proof. $\qquad\square$

# E  Proof of Results in Section 4

## E.1  Proof of Theorem 4.1

In this section we give the proofs of our universal approximation theorems of self-attention. We first prove the $L_\infty$ norm version whose target function are continuous. Then we combine this result with the well known Lusin's theorem and extend our result to Lebesgue integrable functions in terms of $L_p$ norm.

> **Theorem E.1** (Theorem 4.1 Restated: $L_\infty$-Norm Universal Approximation of Self-Attention). Let $f : \mathbb{R}^{d \times n} \to \mathbb{R}^{d \times n}$ denote any continuous function on a compact domain $U \subset \mathbb{R}^{d \times n}$ and let $\epsilon > 0$ be any positive real number. Then, there exists a self-attention $\mathrm{Attn}$ with a prepended $\mathrm{Linear}$ layer, such that
> $$\|f - \mathrm{Attn} \circ \mathrm{Linear}\|_{L_\infty} \leq \epsilon.$$

*Proof Sketch.* Our proof consists of four conceptual steps.

**Step 1: Partition Input Domain $U$ via $\mathrm{MaxAff}$.**

- **Flattening Input.** Each input $Z \in \mathbb{R}^{d \times n}$ is reshaped into a single vector $\widetilde{Z} \in \mathbb{R}^{dn}$ by stacking its rows or columns. This unifies the domain as $\widetilde{Z} \in [-D, D]^{dn}$.

- **Grid / Max-Affine Construction.** Since $f$ is uniformly continuous on the compact set $U$, choose $\delta > 0$ such that
$$\|Z_1 - Z_2\|_\infty < \delta \implies \|f(Z_1) - f(Z_2)\|_\infty < \epsilon.$$
We subdivide $[-D, D]^{dn}$ into cubes of side $\leq \delta$, yielding $G = P^{dn}$ grid centers $\{v_j\}_{j=0}^{G-1}$. We treat $\mathrm{MaxAff}$ as a piecewise (max-)affine or piecewise-constant partition: for each $\widetilde{Z}$, there's a nearest $v_j$ within $\delta/2$.

**Step 2: Configure $\mathrm{Linear}$ and $\mathrm{Attn}$ to Imitate $\mathrm{MaxAff}$ over $U$.**

- **Sum-of-Linear-Transformations Map $\mathrm{Linear}$.** Design $\mathrm{Linear} : \mathbb{R}^{d \times n} \to \mathbb{R}^M$ (for some dimension $M$) to capture the dot products $\langle v_j, \widetilde{Z} \rangle$. Essentially, $\mathrm{Linear}(Z)$ arranges these $\{v_j^\top \widetilde{Z}\}$ in a form accessible to attention. This ensures each grid center $v_j$ can be individually "queried."

- **Encoding Affine Components.** Observe that $\max_j\{\langle v_j, \widetilde{Z} \rangle - \frac{1}{2}\|v_j\|^2\}$ is akin to a max-affine function. We store terms $v_j^\top \widetilde{Z}$, plus $-\frac{1}{2}\|v_j\|^2$, into $K$ and $Q$ for later use in $\mathrm{Softmax}(K^\top Q)$.

**Step 3: Enginner $\mathrm{Attn}$ to Generate an Indicator of Which Partition Cell the Input Belongs To.**

- **Construct $K^\top Q$.** In the self-attention block, let $K^\top Q \approx R(\langle v_j, \widetilde{Z} \rangle - \frac{1}{2}\|v_j\|^2)$, where $R > 0$ is large. This makes $\mathrm{Softmax}(K^\top Q)$ favor the row $j^*$ maximizing
$$\langle v_j, \widetilde{Z} \rangle - \tfrac{1}{2}\|v_j\|^2.$$

- **Near-One-Hot Distribution.** Hence the $j^*$-th row obtains probability close to 1, effectively identifying which grid center $v_{j^*}$ is nearest to $\widetilde{Z}$. We interpret this as a near-one-hot "indicator" vector for the correct partition cell.

**Step 4: Map the Indicator to the Target Value $f(Z)$.**

- **Assigning Values.** We place $f(\widetilde{v}_j)$ in the "value matrix" $W_V$, so that once row $j^*$ is selected, the attention output is $\approx f(\widetilde{v}_{j^*})$. Since $Z$ is within $\delta/2$ of $v_{j^*}$, uniform continuity implies

$$\|f(Z) - f(\widetilde{v}_{j^*})\| \le \epsilon, \text{ (for suitably chosen } \delta).$$

- **Final Reshaping (If Needed).** A small linear projection $M$ can reshape the output back to $\mathbb{R}^{d \times n}$. The essential logic is that the correct $f(\widetilde{v}_j)$ is "routed" to the final output via the near-one-hot attention distribution.

Thus, a single-head attention block with a minimal linear layer can approximate any continuous function on the domain. This completes the proof. $\qquad\square$

*Proof.* We divide our proof into two parts:

- **Part 1: Construction of** $\mathrm{Attn}$ **and** $\mathrm{Linear}$**.** We construct $\mathrm{Attn}$ and $\mathrm{Linear}$ in accordance with the steps shown in the **proof sketch**, and calculate the precise output of our construction.

- **Part 2: Estimation of Approximation Error between** $\mathrm{Attn} \circ \mathrm{Linear}$ **and** $f$**.** We calculate the difference between the output calculated in previous part and the target function to

**Part 1: Construction of** $\mathrm{Attn}$ **and** $\mathrm{Linear}$**.**

We first construct the grid points in $[-D, D]^{dn}$ used in the construction of $\mathrm{Linear}$ and $\mathrm{Attn}$.

These grid points are used to construct the max-affine partition. Specifically, the max-affine partition we use is a grid-partition and these points are the center points of these grids.

**Construction of Grid Centers in** $[-D, D]^{dn}$**.** Let $Z = [z_1, z_2, \cdots, z_n] \in \mathbb{R}^{d \times n}$ denote the input to $\mathrm{Linear}$. Define $\widetilde{Z} := [z_1^\top, z_2^\top, \cdots, z_n^\top]^\top$. $P \in N_+$ is a parameter that controls the size of the attention block and the error of our approximation.

**Definition E.1** (Grid Centers in $[-D, D]^{dn}$). Define $v_{k_1, k_2, \cdots, k_{dn}} \in \mathbb{R}^{dn}$ as

$$v_{k_1, k_2, \cdots, k_{dn}} := \left[ \frac{2Dk_1 - DP}{P}, \frac{2Dk_2 - DP}{P}, \cdots, \frac{2Dk_{dn} - DP}{P} \right]^\top,$$

for $k_i \in \{0, 1, 2, \cdots, P - 1\}$, $i \in [dn]$.

**Remark E.1** (Scalar-Labeled Grid Centers). For each multi-index $(k_1, \ldots, k_{dn})$ with $k_i \in \{0, \ldots, P - 1\}$, we define

$$s := \sum_{i=1}^{dn} k_i \, P^{i-1}, \quad s \in \{0, \ldots, P^{dn} - 1\}.$$

This base-$P$ expansion gives a one-to-one map between the tuple and the scalar. This notation allows us to define another representation of the grid center:

$$v_s := v_{k_1, \ldots, k_{dn}}.$$

For every $v \in V$, we define

$$\widetilde{v} := \underbrace{\left[ v_{1:d}, v_{d+1:2d}, \cdots, v_{(n-1)d+1:nd} \right]}_{d \times n}.$$

We now construct functions $E$ and $T$. They are linear functions of $f : \mathbb{R}^{d \times n} \to \mathbb{R}^{d \times n}$ playing crucial roles in the constructions of $W_K$ and $W_Q$ in $\mathrm{Attn}(\cdot)$.

**Construction of $E$ and $T$.** We first show that $f$ is bounded. Because $f$ is continuous within a closed region, its output value is bounded $\infty$-norm. Let $B_0$ denote this bound

$$B_0 := \|f\|_{L_\infty}.$$

We now construct two functions $E(\cdot), T(\cdot)$ related to $f$. Their sum is a constant while their subtraction is scaled $f$. For any $Z \in \mathbb{R}^{d \times n}$, we define

$$E(Z) := 1_{d \times n} - \frac{f(Z)}{B_0}, \tag{E.1}$$

$$T(Z) := 1_{d \times n} + \frac{f(Z)}{B_0}, \tag{E.2}$$

and

$$(E+T)(Z) := E(Z) + T(Z),$$
$$(E-T)(Z) := E(Z) - T(Z).$$

By the definition of $E(\cdot)$ and $T(\cdot)$, we have

$$(E+T)(Z) \equiv 2_{d \times n} \tag{E.3}$$

$$(E-T)(Z) = \frac{2f(Z)}{B_0}. \tag{E.4}$$

for any $Z \in \mathbb{R}^{d \times n}$.

**Construction of the Layer of Sum of Linear Transformations.** We now construct the Linear layer to be

$$\text{Linear}(Z) := \sum_{j=0}^{G-1} \left( \sum_{k=0}^{(n-1)} \underbrace{(Ze_{k+1}^{(n)})^\top (v_j)_{kd+1:kd+d}}_{d \times 1} \right) e_1^{(2dG+1)} \sum_{s=0}^{d-1} \left( e_{j+s+1}^{(2dG)} + e_{j+s+dG+1}^{(2dG)} \right)^\top + \begin{bmatrix} 0_{1 \times 2dG} \\ I_{2dG} \end{bmatrix},$$

$$\left( e_{j+s+dG+1}^{(2dG)} \text{ is shifting the 1 in } e_{j+s+1}^{(2dG)} \text{ down for } dG \text{ rows.} \right)$$

where $G = P^{dn}$.

This layer multiplies the flattened input with the grid centers in Definition E.1 and append a $2dG$-dimensional identity matrix below the matrix containing these multiplications.

We now express the output of Linear in a simpler form in the following discussion.

First, we show that

$$\sum_{k=0}^{(n-1)} (\underbrace{Ze_{k+1}^{(n)}}_{\text{retrieve the } (k+1)\text{-th token}})^\top (v_j)_{kd+1:kd+d} = \sum_{k=0}^{(n-1)} z_{k+1}^\top (v_j)_{kd+1:kd+d}$$

$$= [z_1^\top, z_2^\top, \cdots, z_n^\top] v_j$$

$$= \widetilde{Z}^\top v_j \qquad \left( \text{By } \widetilde{Z} \text{ being the flattened input} \right)$$

$$= v_j^\top \widetilde{Z} \in \mathbb{R}, \ j \in \{0, 1, 2, \cdots, G-1\}.$$

This yields

$$\text{Linear}(Z) = \sum_{j=0}^{G-1} v_j^\top \widetilde{Z} \sum_{s=0}^{d-1} \left( e_{j+s+1}^{(2dG)} + e_{j+s+dG+1}^{(2dG)} \right)^\top e_1^{(2dG+1)} + \begin{bmatrix} 0_{1 \times 2dG} \\ I_{2dG} \end{bmatrix}$$

$$= \begin{bmatrix} X_0 & X_0 \\ I_{dG} & 0_{dG \times dG} \\ 0_{dG \times dG} & I_{dG} \end{bmatrix}. \tag{E.5}$$

Explicitly, the last line is by

$$\sum_{j=0}^{G-1} v_j^\top \widetilde{Z} \sum_{s=0}^{d-1} \left( e_{j+s+1}^{(2dG)} \right)^\top = X_0,$$

which implies

$$\sum_{j=0}^{G-1} v_j^\top \widetilde{Z} \sum_{s=0}^{d-1} \left( e_{j+s+1}^{(2dG)} + e_{j+s+dG+1}^{(2dG)} \right)^\top = [X_0\ X_0].$$

Here

$$X_0 := \left[ v_0^\top \widetilde{Z} 1_{1\times d} \quad v_1^\top \widetilde{Z} 1_{1\times d} \quad v_2^\top \widetilde{Z} 1_{1\times d} \quad \cdots \quad v_{G-1}^\top \widetilde{Z} 1_{1\times d} \right].$$

To summarize, in the output of the first layer of linear transformations, the first row consists of linear transformations of the flattened input, while the other rows are together an identity matrix ($I_{2dG}$).

**Construction of $K$ and $Q$ Matrices.** We now construct the $W_k$ and $W_Q$ matrices in the self-attention block and calculate the output of $\mathrm{Softmax}\left(K^\top Q\right)$.

We define $W_K$ as follows

$$W_K := \begin{bmatrix} 1 & 0 & \cdots & 0 & 0 & \cdots & 0 \\ 0 & -\frac{\|v_0\|_2^2}{2}1_{1\times d} & \cdots & -\frac{\|v_{G-1}\|_2^2}{2}1_{1\times d} & -\frac{\|v_0\|_2^2}{2}1_{1\times d} & \cdots & -\frac{\|v_{G-1}\|_2^2}{2}1_{1\times d} \\ 0_n & \ln(T(\widetilde{v}_0))^\top & \cdots & \ln(T(\widetilde{v}_{G-1}))^\top & \ln(E(\widetilde{v}_0))^\top & \cdots & \ln(E(\widetilde{v}_{G-1}))^\top \end{bmatrix}.$$

The definition of $W_K$ yields that

$$K := W_K \mathrm{Linear}(Z)$$
$$= \begin{bmatrix} 1 & 0 & 0 & \cdots & 0 & 0 & 0 & \cdots & 0 \\ 0 & -\frac{\|v_0\|_2^2}{2}1_{1\times d} & -\frac{\|v_1\|_2^2}{2}1_{1\times d} & \cdots & -\frac{\|v_{G-1}\|_2^2}{2}1_{1\times d} & -\frac{\|v_0\|_2^2}{2}1_{1\times d} & -\frac{\|v_1\|_2^2}{2}1_{1\times d} & \cdots & -\frac{\|v_{G-1}\|_2^2}{2}1_{1\times d} \\ 0_n & \ln(T(\widetilde{v}_0))^\top & \ln(T(\widetilde{v}_1))^\top & \cdots & \ln(T(\widetilde{v}_{G-1}))^\top & \ln(E(\widetilde{v}_0))^\top & \ln(E(\widetilde{v}_1))^\top & \cdots & \ln(E(\widetilde{v}_{G-1}))^\top \end{bmatrix}$$
$$\cdot \begin{bmatrix} X_0 & X_0 \\ I_{dG} & 0_{dG\times dG} \\ 0_{dG\times dG} & I_{dG} \end{bmatrix}$$
$$= \begin{bmatrix} v_0^\top \widetilde{Z} 1_{1\times d} & v_1^\top \widetilde{Z} 1_{1\times d} & \cdots & v_{G-1}^\top \widetilde{Z} 1_{1\times d} & v_0^\top \widetilde{Z} 1_{1\times d} & v_1^\top \widetilde{Z} 1_{1\times d} & \cdots & v_{G-1}^\top \widetilde{Z} 1_{1\times d} \\ -\frac{\|v_0\|_2^2}{2}1_{1\times d} & -\frac{\|v_1\|_2^2}{2}1_{1\times d} & \cdots & -\frac{\|v_{G-1}\|_2^2}{2}1_{1\times d} & -\frac{\|v_0\|_2^2}{2}1_{1\times d} & -\frac{\|v_1\|_2^2}{2}1_{1\times d} & \cdots & -\frac{\|v_{G-1}\|_2^2}{2}1_{1\times d} \\ \ln(T(\widetilde{v}_0))^\top & \ln(T(\widetilde{v}_1))^\top & \cdots & \ln(T(\widetilde{v}_{G-1}))^\top & \ln(E(\widetilde{v}_0))^\top & \ln(E(\widetilde{v}_1))^\top & \cdots & \ln(E(\widetilde{v}_{G-1}))^\top \end{bmatrix},$$
$$\text{(By (E.5))}$$

where the last line follows from $X_0$ being multiplied by $1$ and thus appearing in the first row of the output.

Next, we construct $W_Q$ to be

$$W_Q := \begin{bmatrix} 0 & R1_{1\times n} & 0_{1\times(2dG-n)} \\ 0 & R1_{1\times n} & 0_{1\times(2dG-n)} \\ 0_n & I_n & 0_{n\times(2dG-n)} \end{bmatrix}.$$

This yields that

$$Q = W_Q \mathrm{Linear}(Z)$$
$$= \begin{bmatrix} 0 & R1_{1\times n} & 0_{1\times(2dG-n)} \\ 0 & R1_{1\times n} & 0_{1\times(2dG-n)} \\ 0_n & I_n & 0_{n\times(2dG-n)} \end{bmatrix} \cdot \begin{bmatrix} X_0 & X_0 \\ I_{dG} & 0_{dG\times dG} \\ 0_{dG\times dG} & I_{dG} \end{bmatrix}$$
$$= \begin{bmatrix} R1_{1\times n} & 0_{1\times(2dG-n)} \\ R1_{1\times n} & 0_{1\times(2dG-n)} \\ I_n & 0_{n\times(2dG-n)} \end{bmatrix}.$$

We now calculate the attention matrix $\mathrm{Softmax}\left(K^\top Q\right)$.

**Calculation of** $\mathrm{Softmax}(K^\top Q)$. First, $K^\top Q$ writes out as

$$
K^\top Q = \begin{bmatrix}
v_0^\top \widetilde{Z} 1_d & \frac{\|v_0\|_2^2}{2} 1_d & \ln(T(\widetilde{v}_0)) \\
v_1^\top \widetilde{Z} 1_d & \frac{\|v_1\|_2^2}{2} 1_d & \ln(T(\widetilde{v}_1)) \\
& \vdots & \\
v_{G-1}^\top \widetilde{Z} 1_d & \frac{\|v_1\|_2^2}{2} 1_d & \ln(T(\widetilde{v}_{G-1})) \\
v_0^\top \widetilde{Z} 1_d & \frac{\|v_0\|_2^2}{2} 1_d & \ln(E(\widetilde{v}_0)) \\
v_1^\top \widetilde{Z} 1_d & \frac{\|v_1\|_2^2}{2} 1_d & \ln(E(\widetilde{v}_1)) \\
& \vdots & \\
v_{G-1}^\top \widetilde{Z} 1_d & \frac{\|v_1\|_2^2}{2} 1_d & \ln(E(\widetilde{v}_{G-1}))
\end{bmatrix} \cdot \begin{bmatrix}
R1_{1\times n} & 0_{1\times(2dG-n)} \\
R1_{1\times n} & 0_{1\times(2dG-n)} \\
I_n & 0_{n\times(2dG-n)}
\end{bmatrix}
$$

$$
= \begin{bmatrix}
R(v_0^\top \widetilde{Z} - \frac{\|v_0\|_2^2}{2})1_{d\times n} + \ln(T(\widetilde{v}_0)) & 0_{d\times(2dG-n)} \\
R(v_1^\top \widetilde{Z} - \frac{\|v_1\|_2^2}{2})1_{d\times n} + \ln(T(\widetilde{v}_1)) & 0_{d\times(2dG-n)} \\
\vdots & \vdots \\
R(v_{G-1}^\top \widetilde{Z} - \frac{\|v_{G-1}\|_2^2}{2})1_{d\times n} + \ln(T(\widetilde{v}_{G-1})) & 0_{d\times(2dG-n)} \\
R(v_0^\top \widetilde{Z} - \frac{\|v_0\|_2^2}{2})1_{d\times n} + \ln(E(\widetilde{v}_0)) & 0_{d\times(2dG-n)} \\
R(v_1^\top \widetilde{Z} - \frac{\|v_1\|_2^2}{2})1_{d\times n} + \ln(E(\widetilde{v}_1)) & 0_{d\times(2dG-n)} \\
\vdots & \vdots \\
R(v_{G-1}^\top \widetilde{Z} - \frac{\|v_{G-1}\|_2^2}{2})1_{d\times n} + \ln(E(\widetilde{v}_{G-1})) & 0_{d\times(2dG-n)}
\end{bmatrix}, \tag{E.6}
$$

where the last line follows from the multiplication of block matrices. This multiplication between $K^\top$ and $Q$ is equivalent to first multiplying the first 2 columns in $K^\top$ with $R$ and then broadcasting their sum to the first $n$ columns, and then adding the result with $T$ and $E$ related blocks. Columns are all filled with $0$ except for the first $n$ columns.

**Remark E.2** (Interpretation of $K^\top Q$). The non-zero entries of $K^\top Q$ is an aggregation of two matrices

$$
\begin{bmatrix}
R(v_0^\top \widetilde{Z} - \frac{\|v_0\|_2^2}{2})1_{d\times n} \\
R(v_1^\top \widetilde{Z} - \frac{\|v_1\|_2^2}{2})1_{d\times n} \\
\vdots \\
R(v_{G-1}^\top \widetilde{Z} - \frac{\|v_{G-1}\|_2^2}{2})1_{d\times n} \\
R(v_0^\top \widetilde{Z} - \frac{\|v_0\|_2^2}{2})1_{d\times n} \\
R(v_1^\top \widetilde{Z} - \frac{\|v_1\|_2^2}{2})1_{d\times n} \\
\vdots \\
R(v_{G-1}^\top \widetilde{Z} - \frac{\|v_{G-1}\|_2^2}{2})1_{d\times n}
\end{bmatrix}
\tag{E.7}
$$

and

$$
\begin{bmatrix}
\ln(T(\widetilde{v}_0)) \\
\ln(T(\widetilde{v}_1)) \\
\vdots \\
\ln(T(\widetilde{v}_{G-1})) \\
\ln(E(\widetilde{v}_0)) \\
\ln(E(\widetilde{v}_1)) \\
\vdots \\
\ln(E(\widetilde{v}_{G-1}))
\end{bmatrix}.
\tag{E.8}
$$

In these two matrices, (E.7) is identical between columns and has the precision coefficient $R$ free of our choice. In later discussions, we set $R$ to be sufficiently large so that the Softmax approximates a maximum function, and "selects" the $i$ of the maximal $R(v_i^\top \widetilde{Z} - \frac{\|v_i\|_2^2}{2})1_{d\times n}$ for $i \in \{0, 1, \cdots, G-1\}$. By "select" we mean only the entries with the selected label has a value not close to 0 in each column of $\mathrm{Softmax}(K^\top Q)$.
(E.8) does not include $R$ related terms. Thus when $R$ is set to be sufficiently large in our later discussions, (E.8) does not affect the selection made by (E.7).
If we exclude the (E.8) in the attention score matrix $\mathrm{Softmax}(K^\top Q)$, the output approximates a matrix whose columns are all-zero except for two sub-vector equal to $1/2d \cdot 1_d$. This writes out as (here we only show the first $n$ non-constant columns)

$$
\begin{bmatrix}
0_{(s-1)d\times n} \\
\frac{1}{2d}1_{d\times n} \\
0_{(G-s)d\times n} \\
0_{(s-1)d\times n} \\
\frac{1}{2d}1_{d\times n} \\
0_{(G-s)d\times n}
\end{bmatrix},
\tag{E.9}
$$

for any $s \in [G]$. The addition of (E.8) change the $1_d$ in (E.9) to

$$
\begin{bmatrix}
0_{(s-1)d\times n} \\
\frac{1}{2d}T(\widetilde{v}_{s-1}) \\
0_{(G-s)d\times n} \\
0_{(s-1)d\times n} \\
\frac{1}{2d}E(\widetilde{v}_{s-1}) \\
0_{(G-s)d\times n}
\end{bmatrix}.
\tag{E.10}
$$

In later discussion, we use $V$ to transform (E.10) to $T(\widetilde{v}_{s-1}) - E(\widetilde{v}_{s-1}) = 2f(\widetilde{v}_{s-1})/2dB_0$ to obtain the final output.

Now, we divide the calculation of $\mathrm{Softmax}\left(K^\top Q\right)$ into two parts: the calculation of $\exp\left(K^\top Q\right)$ and the calculation of the denominator of every column of $\mathrm{Softmax}\left(K^\top Q\right)$. This denominator explicitly writes out as $\sum_{j=1}^{2dG} \exp\left(K^\top Q\right)_{ij}$ for each $i \in [2dG]$.

For $\exp\left(K^\top Q\right)$, by (E.6), we have

$$
\exp\left(K^\top Q\right) =
\begin{bmatrix}
\exp\left(R(v_0^\top \widetilde{Z} - \frac{\|v_0\|_2^2}{2})\right)T(\widetilde{v}_0) & 1_{d\times(2dG-n)} \\
\exp\left(R(v_1^\top \widetilde{Z} - \frac{\|v_1\|_2^2}{2})\right)T(\widetilde{v}_1) & 1_{d\times(2dG-n)} \\
\vdots & \\
\exp\left(R(v_{G-1}^\top \widetilde{Z} - \frac{\|v_{G-1}\|_2^2}{2})\right)T(\widetilde{v}_{G-1}) & 1_{d\times(2dG-n)} \\
\exp\left(R(v_0^\top \widetilde{Z} - \frac{\|v_0\|_2^2}{2})\right)E(\widetilde{v}_0) & 1_{d\times(2dG-n)} \\
\exp\left(R(v_1^\top \widetilde{Z} - \frac{\|v_1\|_2^2}{2})\right)E(\widetilde{v}_1) & 1_{d\times(2dG-n)} \\
\vdots & \\
\exp\left(R(v_{G-1}^\top \widetilde{Z} - \frac{\|v_{G-1}\|_2^2}{2})\right)E(\widetilde{v}_{G-1}) & 1_{d\times(2dG-n)}
\end{bmatrix}. \qquad \text{(E.11)}
$$

For the denominator, we calculate it in columns. Let $i$ denote the column which we calculate the denominator in Softmax. When $i \in \{n+1, n+2, \cdots, 2dG\}$, there are $1 \cdot 2dG = 2dG$ columns. And when $i \in [n]$, we denote that

$$
\sum_{j=1}^{2dG} \exp\left(K^\top Q\right)_{i,j} = \sum_{j=1}^{G}\left[\left(1_{1\times d}T(\widetilde{v}_{j-1})_{:,i} + 1_{1\times d}E(\widetilde{v}_{j-1})_{:,i}\right) \cdot \exp\left(R\left(v_{j-1}^\top \widetilde{Z} - \frac{\|v_{j-1}\|_2^2}{2}\right)\right)\right]
$$
$$
\left(\text{By (E.11)}\right)
$$
$$
= \sum_{j=1}^{G}\left[\left(1_{1\times d}(E+T)(v_{j-1})_{:,i}\right) \cdot \exp\left(R\left(v_{j-1}^\top \widetilde{Z} - \frac{\|v_{j-1}\|_2^2}{2}\right)\right)\right]
$$
$$
= \sum_{j=1}^{G}\left[\left(1_{1\times d}(2_{d\times n})_{:,i}\right) \cdot \exp\left(R\left(v_{j-1}^\top \widetilde{Z} - \frac{\|v_{j-1}\|_2^2}{2}\right)\right)\right]
$$
$$
= \sum_{j=1}^{G} 2d \cdot \exp\left(R\left(v_{j-1}^\top \widetilde{Z} - \frac{\|v_{j-1}\|_2^2}{2}\right)\right), \quad i \in [n]. \qquad \text{(E.12)}
$$

Observing from (E.12), $\sum_{j=1}^{2dG} \exp\left(K^\top Q\right)_{i,j}$ is *invariant* of $i$ for $i \in [n]$. In this case, we define

$$
\alpha(Z) := \frac{1}{2d}\sum_{j=1}^{2dG} \exp\left(K^\top Q\right)_{i,j} = \sum_{j=1}^{G}\exp\left(R\left(v_{j-1}^\top \widetilde{Z} - \frac{\|v_{j-1}\|_2^2}{2}\right)\right) \in \mathbb{R}, \quad i \in [n].
$$

From (E.11) and (E.12), we have

Softmax $\left(K^\top Q\right)$

$$
= \exp\left(K^\top Q\right) \odot \left[\frac{1}{\sum_{j=1}^{2dG}\exp(K^\top Q)_{1j}}1_{2dG\times n} \quad \frac{1}{2dG}1_{2dG\times(2dG-n)}\right]
$$
$$
\left(\text{By } \frac{1}{\sum_{j=1}^{2dG}\exp\left(K^\top Q\right)_{ij}} \text{ is invariant of } i \text{ for } i \in [n]\right)
$$
$$
=
\begin{bmatrix}
\exp\left(R(v_0^\top \widetilde{Z} - \frac{\|v_0\|_2^2}{2})\right)T(\widetilde{v}_0) & 1_{d\times(2dG-n)} \\
\exp\left(R(v_1^\top \widetilde{Z} - \frac{\|v_1\|_2^2}{2})\right)T(\widetilde{v}_1) & 1_{d\times(2dG-n)} \\
\vdots & \\
\exp\left(R(v_{G-1}^\top \widetilde{Z} - \frac{\|v_{G-1}\|_2^2}{2})\right)T(\widetilde{v}_{G-1}) & 1_{d\times(2dG-n)} \\
\exp\left(R(v_0^\top \widetilde{Z} - \frac{\|v_0\|_2^2}{2})\right)E(\widetilde{v}_0) & 1_{d\times(2dG-n)} \\
\exp\left(R(v_1^\top \widetilde{Z} - \frac{\|v_1\|_2^2}{2})\right)E(\widetilde{v}_1) & 1_{d\times(2dG-n)} \\
\vdots & \\
\exp\left(R(v_{G-1}^\top \widetilde{Z} - \frac{\|v_{G-1}\|_2^2}{2})\right)E(\widetilde{v}_{G-1}) & 1_{d\times(2dG-n)}
\end{bmatrix} \odot \left[\frac{1}{2d\alpha(Z)}1_{2dG\times n} \quad \frac{1}{2dG}1_{2dG\times(2dG-n)}\right]
$$

$$
= \frac{1}{2d}
\begin{bmatrix}
\frac{\exp\left(R(v_0^\top \widetilde{Z} - \frac{\|v_0\|_2^2}{2})\right)}{\alpha(Z)} T(\widetilde{v}_0) & \frac{1}{G} 1_{d \times (2dG - n)} \\[2ex]
\frac{\exp\left(R(v_1^\top \widetilde{Z} - \frac{\|v_1\|_2^2}{2})\right)}{\alpha(Z)} T(\widetilde{v}_1) & \frac{1}{G} 1_{d \times (2dG - n)} \\[2ex]
\vdots & \\[1ex]
\frac{\exp\left(R(v_{G-1}^\top \widetilde{Z} - \frac{\|v_{G-1}\|_2^2}{2})\right)}{\alpha(Z)} T(\widetilde{v}_{G-1}) & \frac{1}{G} 1_{d \times (2dG - n)} \\[2ex]
\frac{\exp\left(R(v_0^\top \widetilde{Z} - \frac{\|v_0\|_2^2}{2})\right)}{\alpha(Z)} E(\widetilde{v}_0) & \frac{1}{G} 1_{d \times (2dG - n)} \\[2ex]
\frac{\exp\left(R(v_1^\top \widetilde{Z} - \frac{\|v_1\|_2^2}{2})\right)}{\alpha(Z)} E(\widetilde{v}_1) & \frac{1}{G} 1_{d \times (2dG - n)} \\[2ex]
\vdots & \\[1ex]
\frac{\exp\left(R(v_{G-1}^\top \widetilde{Z} - \frac{\|v_{G-1}\|_2^2}{2})\right)}{\alpha(Z)} E(\widetilde{v}_{G-1}) & \frac{1}{G} 1_{d \times (2dG - n)}
\end{bmatrix}. \tag{E.13}
$$

**Construction of $W_V$ and $W_O$.** We now construct the $W_V$ matrix and calculate the $V$ matrix of the self-attention.

We define $W_V$ as:

$$
W_V := [0_d \quad X_1 \quad -X_1]_{d \times (1 + 2dG)},
$$

where

$$
X_1 := [I_d \quad I_d \quad \cdots \quad I_d]_{d \times dG},
$$

is a matrix formed by stacking $G$ $I_d$ matrix horizontally.

With this definition, we compute $V$ matrix as follows

$$
\begin{aligned}
V &:= W_V \mathrm{Linear}(Z) \\
&= [0_d \quad X_1 \quad -X_1] \cdot
\begin{bmatrix}
X_0 & X_0 \\
I_{dG} & 0_{dG \times dG} \\
0_{dG \times dG} & I_{dG}
\end{bmatrix} \\
&= [X_1 \quad -X_1]. \tag{E.14}
\end{aligned}
$$

After the construction and calculation of $V$, we go on to construct $W_O$ as:

$$
W_O =
\begin{bmatrix}
dB_0 I_n \\
0_{(2dG - n) \times n}
\end{bmatrix}.
$$

The sole purpose of $W_O$ is to extract the non-zero entries of the final output.

**Calculation of the Output of** $\mathrm{Attn} \circ \mathrm{Linear}$**.** We now compute the final output of the self-attention block

$$
\mathrm{Attn} \circ \mathrm{Linear}(Z)
$$

$$= \frac{1}{2d} \begin{bmatrix} X_1 & -X_1 \end{bmatrix} \cdot \begin{bmatrix} \dfrac{\exp\left(R(v_0^\top \widetilde{Z} - \frac{\|v_0\|_2^2}{2})\right)}{\alpha(Z)} T(\widetilde{v}_0) & \frac{1}{G} 1_{d\times(2dG-n)} \\ \dfrac{\exp\left(R(v_1^\top \widetilde{Z} - \frac{\|v_1\|_2^2}{2})\right)}{\alpha(Z)} T(\widetilde{v}_1) & \frac{1}{G} 1_{d\times(2dG-n)} \\ \vdots & \\ \dfrac{\exp\left(R(v_{G-1}^\top \widetilde{Z} - \frac{\|v_{G-1}\|_2^2}{2})\right)}{\alpha(Z)} T(\widetilde{v}_{G-1}) & \frac{1}{G} 1_{d\times(2dG-n)} \\ \dfrac{\exp\left(R(v_0^\top \widetilde{Z} - \frac{\|v_0\|_2^2}{2})\right)}{\alpha(Z)} E(\widetilde{v}_0) & \frac{1}{G} 1_{d\times(2dG-n)} \\ \dfrac{\exp\left(R(v_1^\top \widetilde{Z} - \frac{\|v_1\|_2^2}{2})\right)}{\alpha(Z)} E(\widetilde{v}_1) & \frac{1}{G} 1_{d\times(2dG-n)} \\ \vdots & \\ \dfrac{\exp\left(R(v_{G-1}^\top \widetilde{Z} - \frac{\|v_{G-1}\|_2^2}{2})\right)}{\alpha(Z)} E(\widetilde{v}_{G-1}) & \frac{1}{G} 1_{d\times(2dG-n)} \end{bmatrix} W_O$$

(By (E.14) and (E.13))

$$= \frac{1}{2d} X_1 \begin{bmatrix} \dfrac{\exp\left(R(v_0^\top \widetilde{Z} - \frac{\|v_0\|_2^2}{2})\right)}{\alpha(Z)} (T(\widetilde{v}_0) - E(\widetilde{v}_0)) & 0_{d\times(2dG-n)} \\ \dfrac{\exp\left(R(v_1^\top \widetilde{Z} - \frac{\|v_1\|_2^2}{2})\right)}{\alpha(Z)} (T(\widetilde{v}_1) - E(\widetilde{v}_1)) & 0_{d\times(2dG-n)} \\ \vdots & \vdots \\ \dfrac{\exp\left(R(v_{G-1}^\top \widetilde{Z} - \frac{\|v_{G-1}\|_2^2}{2})\right)}{\alpha(Z)} (T(\widetilde{v}_{G-1}) - E(\widetilde{v}_{G-1})) & 0_{d\times(2dG-n)} \end{bmatrix} W_O$$

(Sum of $X_1$ multiplied by $T$ related blocks and $-X_1$ multiplied by $E$ related ones)

$$= \frac{1}{2d} X_1 \begin{bmatrix} \dfrac{\exp\left(R(v_0^\top \widetilde{Z} - \frac{\|v_0\|_2^2}{2})\right)}{\alpha(Z)} \cdot \dfrac{2f(\widetilde{v}_0)}{B_0} & 0_{d\times(2dG-n)} \\ \dfrac{\exp\left(R(v_1^\top \widetilde{Z} - \frac{\|v_1\|_2^2}{2})\right)}{\alpha(Z)} \cdot \dfrac{2f(\widetilde{v}_1)}{B_0} & 0_{d\times(2dG-n)} \\ \vdots & \\ \dfrac{\exp\left(R(v_{G-1}^\top \widetilde{Z} - \frac{\|v_{G-1}\|_2^2}{2})\right)}{\alpha(Z)} \dfrac{2f(\widetilde{v}_{G-1})}{B_0} & 0_{d\times(2dG-n)} \end{bmatrix} W_O. \qquad \text{(By (E.3))}$$

Let $I_d$ denote the $d$-dimensional identity matrix. We have

$$X_1 \begin{bmatrix} \dfrac{\exp\left(R(v_0^\top \widetilde{Z} - \frac{\|v_0\|_2^2}{2})\right)}{\alpha(Z)} \dfrac{2f(\widetilde{v}_0)}{B_0} \\ \dfrac{\exp\left(R(v_1^\top \widetilde{Z} - \frac{\|v_1\|_2^2}{2})\right)}{\alpha(Z)} \dfrac{2f(\widetilde{v}_0)}{B_0} \\ \vdots \\ \dfrac{\exp\left(R(v_{G-1}^\top \widetilde{Z} - \frac{\|v_{G-1}\|_2^2}{2})\right)}{\alpha(Z)} \dfrac{2f(\widetilde{v}_{G-1})}{B_0} \end{bmatrix}$$

$$= \begin{bmatrix} I_d & I_d & \cdots & I_d \end{bmatrix}_{d \times dG} \cdot \underbrace{\begin{bmatrix} \dfrac{\exp\left(R(v_0^\top \widetilde{Z} - \frac{\|v_0\|_2^2}{2})\right)}{\alpha(Z)} \dfrac{2f(\widetilde{v}_0)}{B_0} \\ \dfrac{\exp\left(R(v_1^\top \widetilde{Z} - \frac{\|v_1\|_2^2}{2})\right)}{\alpha(Z)} \dfrac{2f(\widetilde{v}_1)}{B_0} \\ \vdots \\ \dfrac{\exp\left(R(v_{G-1}^\top \widetilde{Z} - \frac{\|v_{G-1}\|_2^2}{2})\right)}{\alpha(Z)} \dfrac{2f(\widetilde{v}_{G-1})}{B_0} \end{bmatrix}}_{:=S}$$

(Equivalent to summing all blocks in $S$)

$$= \sum_{j=0}^{G-1} I_d \cdot \frac{\exp\left(R(v_j^\top \widetilde{Z} - \frac{\|v_{j-1}\|_2^2}{2})\right)}{\alpha(Z)} \frac{2f(\widetilde{v}_j)}{B_0}$$

$$= \sum_{j=0}^{G-1} \frac{1}{\alpha(Z)} \exp\left(R(v_j^\top \widetilde{Z} - \frac{\|v_j\|_2^2}{2})\right) \frac{2f(\widetilde{v}_j)}{B_0}.$$

This yields

$$\text{Attn} \circ \text{Linear}(Z) = \begin{bmatrix} \sum_{j=0}^{G-1} \frac{\exp\left(R(v_j^\top \widetilde{Z} - \frac{\|v_j\|_2^2}{2})\right)}{\alpha(Z)} \frac{2f(\widetilde{v}_j)}{B_0} & 0_{d \times (2dG-n)} \end{bmatrix} W_O$$

$$= \begin{bmatrix} \sum_{j=0}^{G-1} \frac{\exp\left(R(v_j^\top \widetilde{Z} - \frac{\|v_j\|_2^2}{2})\right)}{\alpha(Z)} \frac{2f(\widetilde{v}_j)}{B_0} & 0_{d \times (2dG-n)} \end{bmatrix} \cdot \begin{bmatrix} dB_0 I_n \\ 0_{(2dG-n) \times n} \end{bmatrix}$$

$$= \sum_{j=0}^{G-1} \frac{1}{\alpha(Z)} \exp\left(R(v_j^\top \widetilde{Z} - \frac{1}{2}\|v_j\|_2^2)\right) f(\widetilde{v}_j). \tag{E.15}$$

**Part 2: Estimation of the Approximation Error between $\text{Attn} \circ \text{Linear}$ and $f$.**

With above calculations of the output of $\text{Attn} \circ \text{Linear}$, we now demonstrate how this output approximates our target function.

Essentially, we demonstrate that each term in the summation of (E.15), given by

$$\frac{1}{\alpha(Z)} \exp\left(R(v_j^\top \widetilde{Z} - \frac{1}{2}|v_j|_2^2)\right),$$

approximates a max-affine indicator as $R$ becomes sufficiently large. They are each multiplied with $f(\widetilde{v}_j)$, which is the value of the target function at the center point of the indicated region.

> **Definition E.2** (Max-Affine Function on $\widetilde{Z}$). Let $\text{Aff}_j \in \mathbb{R}^{dn} \to \mathbb{R}$ with $j \in \{0, 1, 2, \cdots, G-1\}$ denote a group of affine functions defined as:
>
> $$\text{Aff}_j(\widetilde{Z}) = v_j^\top \widetilde{Z} - \frac{1}{2}\|v_j\|_2^2, \quad j \in \{0, 1, 2, \cdots, G-1\}.$$
>
> Then let $\text{MaxAff} \in \mathbb{R}^{dn} \to \mathbb{R}$ denote a max affine function whose affine components are $\{\text{Aff}_j \mid j \in \{0, 1, 2, \cdots, G-1\}\}$. Explicitly defined as:
>
> $$\text{MaxAff}(\widetilde{Z}) = \max_{j \in \{0,1,2,\cdots,G-1\}} \left\{ \text{Aff}_j(\widetilde{Z}) \right\}.$$

Because the target function $f$ is a continuous function on a closed domain, the function $f$ is uniformly continuous. Thus for $\epsilon$, there exists a $\delta > 0$ such that for any $Z_1, Z_2$, as long as $\|\widetilde{Z}_1 - \widetilde{Z}_2\|_\infty \leq \delta$, we have $\|f(Z_1) - f(Z_2)\|_\infty \leq \epsilon/3$.

According to this $\delta$, we divide the affine components of $\text{MaxAff}$ into three parts:

1. The maximal component, which has the smallest label $j_m$.

2. All affine components that match the maximal component or fall within $\delta$ of it ($J_0$ as defined below).

3. The remaining $\mathrm{Aff}_j$ for $j \in \{0, 1, \ldots, G-1\}$ ($J_1$ as defined below).

We write out the labels of these groups of components as follows

$$j_m := \min_{j \in \{0,1,2,\cdots,G-1\}} \{\mathrm{Aff}_j(\widetilde{Z}) = \mathrm{MaxAff}(\widetilde{Z})\},$$

$$J_0 := \{j \mid \mathrm{MaxAff}(\widetilde{Z}) - \mathrm{Aff}_j(\widetilde{Z}) \le \delta\},$$

$$J_1 := \{j \mid \mathrm{MaxAff}(\widetilde{Z}) - \mathrm{Aff}_j(\widetilde{Z}) > \delta\}.$$

For any pair of $i, j \in \{0, 1, \cdots, G-1\}$, we have

$$\mathrm{Aff}_i(\widetilde{Z}) - \mathrm{Aff}_j(\widetilde{Z}) = v_i^\top \widetilde{Z} - \frac{\|v_i\|_2^2}{2} - \left(v_j^\top \widetilde{Z} - \frac{\|v_j\|_2^2}{2}\right)$$

$$= -\frac{\|\widetilde{Z}\|_2^2}{2} + v_i^\top \widetilde{Z} - \frac{\|v_i\|_2^2}{2} - \left(-\frac{\|\widetilde{Z}\|_2^2}{2} + v_j^\top \widetilde{Z} - \frac{\|v_j\|_2^2}{2}\right)$$

$$= -\frac{1}{2}\|\widetilde{Z} - v_i\|_2^2 + \frac{1}{2}\|\widetilde{Z} - v_j\|_2^2.$$

Thus for $j_m$, we have

$$-\frac{1}{2}\|\widetilde{Z} - v_{j_m}\|_2^2 + \frac{1}{2}\|\widetilde{Z} - v_j\|_2^2 = \mathrm{Aff}_{j_m}(\widetilde{Z}) - \mathrm{Aff}_j(\widetilde{Z}) \ge 0, \quad j \in \{0, 1, \cdots, G-1\}.$$

This yields

$$\|\widetilde{Z} - v_{j_m}\|_2^2 \le \|\widetilde{Z} - v_j\|_2^2,$$

for all $j \in \{0, 1, \cdots, G-1\}$.

This denotes $j_m$ is also the label of the closest $v_i$ to $\widetilde{Z}$ among all $v_i$, $i \in \{0, 1, \cdots, G-1\}$. Thus we have

$$\|v_{j_m} - \widetilde{Z}\|_2 = \min_{i \in \{0,1,\cdots,G-1\}} \{\|v_i - \widetilde{Z}\|_2\}. \tag{E.16}$$

Now, we prove $v_{j_m}$ (the grid point nearest to $\widetilde{Z}$) has a distance to $\widetilde{Z}$ smaller than half of the grid width (e.g., $D/g$) in infinite norm.

Let $\mathcal{D} := 2D/g \times \{-1, 0, 1\}^{dn}$ denote a set differences to $v_{j_m}$ from the set of all $v_i$ ($i \in \{0, 1, \cdots, G-1\}$) neighboring $v_{j_m}$. For any $\Delta$ in $\mathcal{D}$, from (E.16) we have

$$\|v_{j_m} - \widetilde{Z}\|_2^2 \le \|v_{j_m} + \Delta - \widetilde{Z}\|_2^2.$$

This yields

$$2\Delta^\top (\widetilde{Z} - v_{j_m}) \le \|\Delta\|_2^2.$$

This means that, for any $k \in [dn]$, by selecting $\Delta$ to be $\pm 2D/g e_k^{(dn)}$, we have:

$$\pm 2 \cdot \frac{2D}{g}(\widetilde{Z} - v_{j_m})_k = 2\Delta^\top (\widetilde{Z} - v_{j_m}) \le \|\Delta\|_2^2 = \frac{4D^2}{g^2}.$$

Thus we have

$$\left(\left|\widetilde{Z} - v_{j_m}\right|\right)_k \le \frac{D}{g}, \ k \in [dn],$$

which implies

$$\|\widetilde{Z} - v_{j_m}\|_\infty \le \frac{D}{g}.$$

Set $g$ to be larger than $2D/\delta$; we have

$$\|\widetilde{Z} - v_{j_m}\|_\infty \leq \frac{\delta}{2},$$

thus

$$\|f(Z) - f(\widetilde{v}_{j_m})\|_\infty \leq \frac{\epsilon}{3}, \tag{E.17}$$

where the inequality holds by $\delta/2 < \delta$.

**Calculation of** $\|\text{Attn} \circ \text{Linear} - f\|_\infty$. We now calculate the difference between the output in (E.15) and target function $f$

$$\|\text{Attn} \circ \text{Linear}(Z) - f(Z)\|_\infty$$

$$= \|\sum_{j=0}^{G-1} \frac{\exp\left(R(v_j^\top \widetilde{Z} - \frac{\|v_j\|_2^2}{2})\right)}{\alpha(Z)} f(\widetilde{v}_j) - f(Z)\|_\infty$$

$$= \|\sum_{j=0}^{G-1} \frac{\exp\left(R(v_j^\top \widetilde{Z} - \frac{\|v_j\|_2^2}{2})\right)}{\alpha(Z)} (f(\widetilde{v}_j) - f(Z))\| \qquad \left(\text{By } \sum_{j=0}^{G-1} \frac{\exp\left(R(v_j^\top \widetilde{Z} - \frac{\|v_j\|_2^2}{2})\right)}{\alpha(Z)} = 1\right)$$

$$\leq \sum_{j=0}^{G-1} \frac{\exp\left(R(v_j^\top \widetilde{Z} - \frac{\|v_j\|_2^2}{2})\right)}{\alpha(Z)} \|f(\widetilde{v}_j) - f(Z)\|_\infty \qquad \left(\text{By property of infinite norm}\right)$$

$$= \frac{\exp\left(R(v_{j_m}^\top \widetilde{Z} - \frac{\|v_{j_m}\|_2^2}{2})\right)}{\alpha(Z)} \|f(\widetilde{v}_{j_m}) - f(Z)\|_\infty$$

$$+ \sum_{j \in J_0} \frac{\exp\left(R(v_j^\top \widetilde{Z} - \frac{\|v_j\|_2^2}{2})\right)}{\alpha(Z)} \|f(\widetilde{v}_j) - f(Z)\|_\infty$$

$$+ \sum_{j \in J_1} \frac{\exp\left(R(v_j^\top \widetilde{Z} - \frac{\|v_j\|_2^2}{2})\right)}{\alpha(Z)} \|f(\widetilde{v}_j) - f(Z)\|_\infty. \tag{E.18}$$

We now calculate each part in (E.18).

As previously stated, for any $Z_1, Z_2$, as long as $\|\widetilde{Z}_1 - \widetilde{Z}_2\|_\infty \leq \delta$, we have $\|f(Z_1) - f(Z_2)\|_\infty \leq \epsilon/3$. Thus when we designate $Z_1 = v_j$ for any $j \in J_0$ and $Z_2 = v_{j_m}$, along with (E.17) we have

$$\sum_{j \in J_0} \frac{\exp\left(R(v_j^\top \widetilde{Z} - \frac{\|v_j\|_2^2}{2})\right)}{\alpha(Z)} \|f(\widetilde{v}_j) - f(Z)\|_\infty$$

$$\leq \sum_{j \in J_0} \frac{\exp\left(R(v_j^\top \widetilde{Z} - \frac{\|v_j\|_2^2}{2})\right)}{\alpha(Z)} (\|f(\widetilde{v}_j) - f(\widetilde{v}_{j_m})\|_\infty + \|f(\widetilde{v}_{j_m}) - f(Z)\|_\infty)$$

$$\leq \sum_{j \in J_0} \frac{\exp\left(R(v_j^\top \widetilde{Z} - \frac{\|v_j\|_2^2}{2})\right)}{\alpha(Z)} \cdot (\frac{\epsilon}{3} + \frac{\epsilon}{3})$$

$$= \sum_{j \in J_0} \frac{\exp\left(R(v_j^\top \widetilde{Z} - \frac{\|v_j\|_2^2}{2})\right)}{\alpha(Z)} \cdot \frac{2\epsilon}{3}. \tag{E.19}$$

For $j_m$, we have

$$\frac{\exp\left(R(v_{j_m}^\top \widetilde{Z} - \frac{1}{2}\|v_{j_m}\|_2^2)\right)}{\alpha(Z)} \|f(\widetilde{v}_{j_m}) - f(Z)\|_\infty \leq \frac{\exp\left(R(v_{j_m}^\top \widetilde{Z} - \frac{1}{2}\|v_{j_m}\|_2^2)\right)}{\alpha(Z)} \cdot \frac{\epsilon}{3}. \tag{E.20}$$

When $R$ is larger than $\frac{8}{3\delta^2} \ln\left(\frac{3}{2} B_0 G \epsilon\right)$, we have

$$\sum_{j \in J_1} \frac{\exp\left(R(v_j^\top \widetilde{Z} - \frac{\|v_j\|_2^2}{2})\right)}{\alpha(Z)} \|f(\widetilde{v}_j) - f(Z)\|_\infty$$

$$\leq \sum_{j \in J_1} \frac{\exp\left(R(v_j^\top \widetilde{Z} - \frac{\|v_j\|_2^2}{2})\right)}{\alpha(Z)} \cdot 2B_0 \qquad \left(\text{By } \|f\|_{L_\infty} = B_0\right)$$

$$\leq 2B_0 \frac{\sum_{j \in J_1} \exp\left(R(v_j^\top \widetilde{Z} - \frac{\|v_j\|_2^2}{2})\right)}{\alpha(Z)}$$

$$< 2B_0 \frac{\sum_{j \in J_1} \exp\left(R(v_j^\top \widetilde{Z} - \frac{\|v_j\|_2^2}{2})\right)}{\exp\left(R(v_{j_m}^\top \widetilde{Z} - \frac{\|v_{j_m}\|_2^2}{2})\right)}$$

$$\left(\alpha(Z) \text{ is the sum of all } \exp\left(R(v_j^\top \widetilde{Z} - \frac{\|v_j\|_2^2}{2})\right), \text{ thus larger than } \exp\left(R(v_{j_m}^\top \widetilde{Z} - \frac{\|v_{j_m}\|_2^2}{2})\right)\right)$$

$$= 2B_0 \sum_{j \in J_1} \exp\left(\frac{R}{2}(\|v_{j_m} - Z\|_2^2 - \|v_j - Z\|_2^2)\right)$$

$$\leq 2B_0 \|J_1\| \exp\left(\frac{R}{2}\left[(\frac{\delta}{2})^2 - \delta^2\right]\right)$$

$$< 2B_0 G \exp\left(\frac{-3R\delta^2}{8}\right)$$

$$\leq 2B_0 G \exp\left(\frac{-3\delta^2 \cdot \frac{8 \ln\left(\frac{2}{3} B_0 G \epsilon\right)}{3\delta^2}}{8}\right) \qquad \left(\text{By } R \geq \frac{8}{3\delta^2} \ln\left(\frac{3}{2} B_0 G \epsilon\right)\right)$$

$$= \frac{\epsilon}{3}. \tag{E.21}$$

Combining (E.19) and (E.20) yields

$$\sum_{j \in J_0 \cup \{j_m\}} \frac{\exp\left(R(v_j^\top \widetilde{Z} - \frac{\|v_j\|_2^2}{2})\right)}{\alpha(Z)} \|f(\widetilde{v}_j) - f(Z)\|_\infty$$

$$\leq \sum_{j \in J_0} \frac{\exp\left(R(v_j^\top \widetilde{Z} - \frac{\|v_j\|_2^2}{2})\right)}{\alpha(Z)} \frac{2\epsilon}{3} + \frac{\exp\left(R(v_{j_m}^\top \widetilde{Z} - \frac{\|v_{j_m}\|_2^2}{2})\right)}{\alpha(Z)} \frac{\epsilon}{3} \qquad \left(\text{By (E.19) and (E.20)}\right)$$

$$\leq \sum_{j \in J_0 \cup \{j_m\}} \frac{\exp\left(R(v_j^\top \widetilde{Z} - \frac{\|v_j\|_2^2}{2})\right)}{\alpha(Z)} \frac{2\epsilon}{3}$$

$$\leq \frac{2\epsilon}{3}, \tag{E.22}$$

where the last line is by $\sum_{j \in J_0 \cup \{j_m\}} \frac{1}{\alpha(Z)} \exp\left(R(v_j^\top \widetilde{Z} - \frac{1}{2}\|v_j\|_2^2)\right) \leq 1$.

We plug (E.21) and (E.22) to (E.18) and get

$$\|\text{Attn} \circ \text{Linear}(Z) - f(Z)\|_\infty \leq \frac{\exp\left(R(v_{j_m}^\top \widetilde{Z} - \frac{\|v_{j_m}\|_2^2}{2})\right)}{\alpha(Z)} \|f(\widetilde{v}_{j_m}) - f(Z)\|_\infty$$

$$+ \sum_{j \in J_0} \frac{\exp\left(R(v_j^\top \widetilde{Z} - \frac{\|v_j\|_2^2}{2})\right)}{\alpha(Z)} \|f(\widetilde{v}_j) - f(Z)\|_\infty$$

$$+ \sum_{j \in J_1} \frac{\exp\left(R(v_j^\top \widetilde{Z} - \frac{\|v_j\|_2^2}{2})\right)}{\alpha(Z)} \|f(\widetilde{v}_j) - f(Z)\|_\infty$$

$$\leq \frac{2\epsilon}{3} + \frac{\epsilon}{3}$$
$$= \epsilon.$$

This completes the proof. □

We also extend this $L_\infty$-Norm result we just proved to $L_p$-Norm.

**Corollary E.1.1** ($L_p$-Norm Universal Approximation). Let $f : \mathbb{R}^{d\times n} \to \mathbb{R}^{d\times n}$ denote any Lebesgue integrable function on a compact domain $U \in \mathbb{R}^{d\times n}$ and let $\epsilon > 0$ be any positive real number. Then, there exists a self-attention $\text{Attn}$ prepended with a $\text{Linear}$ layer such that
$$\|f - \text{Attn} \circ \text{Linear}\|_{L_p} \leq \epsilon.$$

*Proof Sketch.* The same partition-based construction applies almost everywhere; outside a negligible set, $f$ is continuous (Lusin's theorem). Thus the $L_\infty$ argument extends. □

*Proof.* Since $f$ is Lebesgue integrable on a compact set, $f$ is bounded almost every where. Let $B_p$ denote the bound of $\|f\|_p$.

By Lusin's theorem, for $f$ on a compact domain $U$, there exists a continuous function $g$ which is equal to $f$ in $U$ except for a region $D_\delta$ such that $\mu(D_\delta) \leq \Delta$. This can be written as

$$D_\delta = \{Z | f(Z) \neq g(Z)\}, \tag{E.23}$$
$$\mu(D_\delta) \leq \Delta. \tag{E.24}$$

Here $\mu$ stands for the Lebesgue measure of a set.

By Theorem 4.1, there exists a net work $\text{Attn} \circ \text{Linear}$, consists of a self-attention $\text{Attn}$ and a layer of sum of linear transformation $\text{Linear}$ such that

$$\|\text{Attn} \circ \text{Linear} - g\|_{L_\infty} \leq \epsilon_0,$$

for any $\epsilon_0 > 0$.

This denote that for any $Z \in U$

$$\|\text{Attn} \circ \text{Linear}(Z) - g(Z)\|_p \leq (dn \cdot \epsilon^p)^{\frac{1}{p}} = \epsilon_0(dn)^{\frac{1}{p}}.$$

Combine this with (E.23) and (E.24), we get

$$\mu(\{Z | \|\text{Attn} \circ \text{Linear}(Z) - g(Z)\|_\infty > \epsilon_0\}) \leq \mu(\{f(Z) \neq g(Z)\})$$
$$\leq \Delta, \tag{E.25}$$

since that $f(Z) = g(Z)$, $\|\text{Attn} \circ \text{Linear}(Z) - g(Z)\| = \|\text{Attn} \circ \text{Linear}(Z) - f(Z)\| \leq \epsilon_0$.

This yields

$$\|f - \text{Attn} \circ \text{Linear}\|_{L_p} = \left(\int_{Z\in U} \|f - \text{Attn} \circ \text{Linear}\|_p^p \, dx\right)^{\frac{1}{p}}$$
$$\leq \left(\int_{Z\in U\setminus D_\delta} \|f - \text{Attn} \circ \text{Linear}\|_p^p \, dx + \int_{Z\in D_\delta} \|f - \text{Attn} \circ \text{Linear}\|_p^p \, dx\right)^{\frac{1}{p}}$$
$$= \left(\int_{Z\in U\setminus D_\delta} \|g - \text{Attn} \circ \text{Linear}\|_p^p \, dx + \int_{Z\in D_\delta} \|f - \text{Attn} \circ \text{Linear}\|_p^p \, dx\right)^{\frac{1}{p}}$$
$$\leq \left(\mu(U\setminus D_\delta)(\epsilon_0(dn)^{\frac{1}{p}})^p + \Delta \cdot B_p^p\right)^{\frac{1}{p}} \qquad (\text{By (E.25)})$$
$$\leq \epsilon_0(dn\mu(U))^{\frac{1}{p}} + \Delta^{\frac{1}{p}} B_p.$$

Set

$$\epsilon_0 \leq \frac{\epsilon}{2(dn\mu(U))^{\frac{1}{p}}}$$
$$\Delta \leq \frac{\epsilon^p}{B_p \cdot 2^p}.$$

We have

$$\|f - \mathrm{Attn} \circ \mathrm{Linear}\|_{L_p} \leq \epsilon_0 (dn\mu(U))^{\frac{1}{p}} + \Delta^{\frac{1}{p}} B_p$$

$$\leq (dn\mu(U))^{\frac{1}{p}} \cdot \frac{\epsilon}{2(dn\mu(U))^{\frac{1}{p}}} + \left(\frac{\epsilon^p}{B_p \cdot 2^p}\right)^{\frac{1}{p}} B_p$$

$$= \frac{\epsilon}{2} + \frac{\epsilon}{2}$$

$$= \epsilon.$$

This completes our proof. □

## E.2 Proof of Theorem 4.2

**Theorem E.2.** Let $U_K \subset \mathbb{R}^{d \times n}$ and $U_Q \subset \mathbb{R}^{d \times n}$ be two compact domains, and let $f : U_K \times U_Q \to \mathbb{R}^{d \times n}$ be any continuous function that takes input from both domains. We use $Z_K, Z_Q \in \mathbb{R}^{d \times n}$ to denote the two inputs of $f$ from $U_K$ and $U_Q$ respectively. Without loss of generality, suppose both input domains to be $[-D, D]^{d \times n}$, where $D \in \mathbb{R}_+$. Then for any $\epsilon > 0$, there exists a single-head cross-attention $\mathrm{Attn}$ and two layers of sum of linear transformations, $\mathrm{Linear}_K$ and $\mathrm{Linear}_Q$ such that:

$$\|\mathrm{Attn}\left(\mathrm{Linear}_K(Z_K), \mathrm{Linear}_Q(Z_Q)\right) - f(Z_K, Z_Q)\|_\infty \leq \epsilon,$$

for any $Z_K, Z_Q \in [-D, D]^{d \times n}$.

*Proof.* Without loss of generality, assume $U_K = U_Q = [-D, D]^{d \times n}$ for a $D \in R_+$.

**Construction of Grid Centers in** $U_K, U_Q$**.** Same as in Appendix E.1, we define $\widetilde{Z} := [z_1^\top, z_2^\top, \cdots, z_n^\top]^\top$. $P \in N_+$ is a parameter that controls the size of the attention block and the error of our approximation. Define $v_{k_1, k_2, \cdots, k_{dn}} \in \mathbb{R}^{dn}$ to be

$$v_{k_1, k_2, \cdots, k_{dn}} := \left[\frac{2Dk_1 - DP}{P}, \frac{2Dk_2 - DP}{P}, \cdots, \frac{2Dk_{dn} - DP}{P}\right]^\top, \ k_i \in \{0, 1, 2, \cdots, P-1\}, \ i \in [dn].$$

Let $V := \{v_{k_1, k_2, \cdots, k_{dn}} | k_i \in \{0, 1, 2, \cdots, P-1\}, \ i \in [dn]\}$ be the set of all $v_{k_1, k_2, \cdots, k_{dn}}$. We also define another way to refer to a vector in $V$, denoted as

$$v_{\sum_{i=1}^{dn} k_i P^{(i-1)}} := v_{k_1, k_2, \cdots, k_{dn}}.$$

Please see Remark E.1 for the reason for the feasibility of such expression.

Following the notation in Appendix E.1, for every $v \in V$, we define

$$\widetilde{v} := \underbrace{[v_{1:d}, v_{d+1:2d}, \cdots, v_{(n-1)d+1, nd}]}_{d \times n}$$

as a $d \times n$ matrix-form representation of $v$.

**Construction of** $f$ **Related Function** $E$ **and** $T$**.** The continuity of $f$ within a closed region guarantees it to be bounded in $\infty$-norm. Let $B_0$ denote this bound. For any $a_K, a_Q \in \mathbb{R}^{d \times n}$, we define

$$E(a_K, a_Q) := 1_{d \times n} - \frac{f(a_K, a_Q)}{B_0}$$

$$T(a_K, a_Q) := 1_{d \times n} + \frac{f(a_K, a_Q)}{B_0}.$$

We define $(E + T)(a_K, a_Q) = E(a_K, a_Q) + T(a_K, a_Q)$. By the definition of $E$ and $T$, $(E + T)(a_K, a_Q) \equiv 2_{d \times n}$ for any $a_K, a_Q \in \mathbb{R}^{d \times n}$.

**Remark E.3** (Intuition behind $E$ and $T$). $\quad E$ and $T$ are constructed to satisfy 3 conditions:
- $T(Z_k, Z_Q) + E(Z_K, Z_Q) \equiv 2$.
- $T(Z_k, Z_Q) - E(Z_K, Z_Q) = 2f(Z_K, Z_Q)/B_0$.
- $T, E > 0$ for any input.

The first condition is used to configure the denominator in the $\mathrm{Softmax}$ expression of attention to a constant value. The second condition is used to form the value of $f$ in the

For simplicity, same as in Appendix E.1, define

$$G := P^{dn}.$$

We now construct the $\mathrm{Linear}_K$ and $\mathrm{Linear}_Q$ layers to be

$$
\mathrm{Linear}_K(Z_K) := \sum_{j=0}^{G-1} \left( \sum_{k=0}^{(n-1)} (Z_K e_{k+1}^{(n)})^\top (v_j)_{kd+1:kd+d} \right) e_{2dG^2+j+1}^{(2dG^2+G)} \sum_{s=0}^{dG-1} \left( e_{j+s+1}^{(2dG^2)} + e_{j+s+dG^2+1}^{(2dG^2)} \right)^\top
$$

$$
+ \begin{bmatrix} I_{2dG^2} \\ 0_{G \times 2dG^2} \end{bmatrix},
$$

$$
\mathrm{Linear}_Q(Z_Q) := \sum_{j=0}^{G-1} \left( \sum_{k=0}^{(n-1)} (Z_Q e_{k+1}^{(n)})^\top (v_j)_{kd+1:kd+d} \right) e_{j+1}^{(n+G)} \begin{bmatrix} 1_{1 \times n} & 0_{1 \times (2dG^2-n)} \end{bmatrix}
$$

$$
+ \begin{bmatrix} 0_{G \times n} & 0_{G \times (2dG^2-n)} \\ I_n & 0_{n \times (2dG^2-n)} \end{bmatrix}.
$$

Same as that in Theorem E.1, we have

$$
\sum_{k=0}^{(n-1)} (Z_K e_{k+1}^{(n)})^\top (v_j)_{kd+1:kd+d} = v_j^\top \widetilde{Z}_K, \tag{E.26}
$$

$$
\sum_{k=0}^{(n-1)} (Z_Q e_{k+1}^{(n)})^\top (v_j)_{kd+1:kd+d} = v_j^\top \widetilde{Z}_Q, \tag{E.27}
$$

for $j \in \{0, 1, 2, \cdots, G-1\}$.

We now calculate the output of $\mathrm{Linear}_K$ and $\mathrm{Linear}_Q$.

For $\mathrm{Linear}_K$, we have

$$
\mathrm{Linear}_K(Z_K) = \sum_{j=0}^{G-1} v_j^\top \widetilde{Z}_K e_{2dG^2+j+1}^{(2dG^2+G)} \sum_{s=0}^{dG-1} \left( e_{j+s+1}^{(2dG^2)} + e_{j+s+dG^2+1}^{(2dG^2)} \right)^\top + \begin{bmatrix} I_{2dG^2} \\ 0_{G \times 2dG^2} \end{bmatrix}
$$

$$
= \begin{bmatrix} I_{2dG^2} \\ \sum_{j=0}^{G-1} v_j^\top \widetilde{Z}_K \sum_{s=0}^{dG-1} \left( e_{j+s+1}^{(2dG^2)} + e_{j+s+dG^2+1}^{(2dG^2)} \right)^\top \end{bmatrix} \qquad \text{(by (E.26))}
$$

$$
= \begin{bmatrix} I_{dG^2} & 0_{dG^2 \times dG^2} \\ 0_{dG^2 \times dG^2} & I_{dG^2} \\ \sum_{j=0}^{G-1} v_j^\top \widetilde{Z}_K \sum_{s=0}^{dG-1} \left( e_{j+s+1}^{(2dG^2)} \right)^\top & \sum_{j=0}^{1-1} v_j^\top \widetilde{Z}_K \sum_{s=0}^{dG-1} \left( e_{j+s+1}^{(2dG^2)} \right)^\top \end{bmatrix}
$$

$$
= \begin{bmatrix} I_{dG^2} & 0_{dG^2 \times dG^2} \\ 0_{dG^2 \times dG^2} & I_{dG^2} \\ X_K & X_K \end{bmatrix}, \tag{E.28}
$$

$$
\mathrm{Linear}_Q(Z_Q) = \sum_{j=0}^{G-1} v_j^\top \widetilde{Z}_Q e_{j+1}^{(n+G)} \begin{bmatrix} 1_{1 \times n} & 0_{1 \times (2dG^2-n)} \end{bmatrix} + \begin{bmatrix} 0_{G \times n} & 0_{G \times (2dG^2-n)} \\ I_n & 0_{n \times (2dG^2-n)} \end{bmatrix}
$$

$$= \begin{bmatrix} \sum_{j=0}^{G-1} v_j^\top \widetilde{Z}_Q e_{j+1}^{(2dG^2+G)} 1_{1\times n} & 0_{1\times(2dG^2-n)} \\ I_n & 0_{n\times(2dG^2-n)} \end{bmatrix} \qquad \text{(by (E.27))}$$

$$= \begin{bmatrix} X_Q & 0_{G\times(2dG^2-n)} \\ I_n & 0_{n\times(2dG^2-n)} \end{bmatrix}, \qquad\qquad\qquad\text{(E.29)}$$

in which $X_K$ and $X_Q$ are defined as

$$X_K := \underbrace{\begin{bmatrix} v_0^\top \widetilde{Z}_K 1_{1\times dG} & v_1^\top \widetilde{Z}_K 1_{1\times dG} & v_2^\top \widetilde{Z}_K 1_{1\times dG} & \cdots & v_{G-1}^\top \widetilde{Z}_K 1_{1\times dG} \end{bmatrix}}_{1\times dG^2},$$

$$X_Q := \underbrace{\begin{bmatrix} v_0^\top \widetilde{Z}_Q 1_{1\times n} \\ v_1^\top \widetilde{Z}_Q 1_{1\times n} \\ v_2^\top \widetilde{Z}_Q 1_{1\times n} \\ \cdots \\ v_{G-1}^\top \widetilde{Z}_Q 1_{1\times n} \end{bmatrix}}_{G\times n}.$$

We now construct the $W_k$ and $W_Q$ matrices in the self-attention block and calculate the output of $\text{Softmax}\left(K^\top Q\right)$.

In the following, we define $W_K$ in parts. First, we present it as a block matrix

$$W_K := \begin{bmatrix} 0_{1\times dG^2} & 0_{1\times dG^2} & 1 \\ W_0 & W_0 & 0 \\ W_1 & W_1 & 0 \\ W_T & W_E & 0 \end{bmatrix}. \qquad\qquad\text{(E.30)}$$

We then define the submatrices in (E.30) as follows

$$W_0 := \begin{bmatrix} -\frac{\|v_0\|_2^2}{2} 1_{1\times dG} + \overline{W}_0 & -\frac{\|v_1\|_2^2}{2} 1_{1\times dG} + \overline{W}_0 & -\frac{\|v_2\|_2^2}{2} 1_{1\times dG} + \overline{W}_0 & \cdots & -\frac{\|v_{G-1}\|_2^2}{2} 1_{1\times dG} + \overline{W}_0 \end{bmatrix},$$

$$W_T := \begin{bmatrix} W_T^{(0)} & W_T^{(1)} & \cdots & W_T^{(G-1)} \end{bmatrix},$$

$$W_E := \begin{bmatrix} W_E^{(0)} & W_E^{(1)} & \cdots & W_E^{(G-1)} \end{bmatrix},$$

$$W_1 := \begin{bmatrix} \overline{W}_1 & \overline{W}_1 & \overline{W}_1 & \cdots & \overline{W}_1 \end{bmatrix}_{G\times dG^2},$$

in which

$$\overline{W}_0 := \begin{bmatrix} -\frac{\|v_0\|_2^2}{2} 1_{1\times d} & -\frac{\|v_1\|_2^2}{2} 1_{1\times d} & -\frac{\|v_2\|_2^2}{2} 1_{1\times d} & \cdots & -\frac{\|v_{G-1}\|_2^2}{2} 1_{1\times d} \end{bmatrix}$$

$$W_T^{(j)} := \underbrace{\begin{bmatrix} \ln(T(\widetilde{v}_j, \widetilde{v}_0))^\top & \ln(T(\widetilde{v}_j, \widetilde{v}_1))^\top & \cdots & \ln(T(\widetilde{v}_j, \widetilde{v}_{G-1}))^\top \end{bmatrix}}_{d\times Gn}, \quad j\in\{0,1,2,\cdots,G-1\},$$

$$W_E^{(j)} := \underbrace{\begin{bmatrix} \ln(E(\widetilde{v}_j, \widetilde{v}_0))^\top & \ln(E(\widetilde{v}_j, \widetilde{v}_1))^\top & \cdots & \ln(E(\widetilde{v}_j, \widetilde{v}_{G-1}))^\top \end{bmatrix}}_{d\times Gn}, \quad j\in\{0,1,2,\cdots,G-1\},$$

$$\overline{W}_1 := \underbrace{\begin{bmatrix} Re_1^{(G)} 1_{1\times d} & Re_2^{(G)} 1_{1\times d} & \cdots & Re_G^{(G)} 1_{1\times d} \end{bmatrix}}_{G\times d}.$$

The definition of $W_K$ yields that

$$K := W_K \text{Linear}_K(Z_K)$$

$$= \begin{bmatrix} 0_{1\times dG^2} & 0_{1\times dG^2} & 1 \\ W_0 & W_0 & 0 \\ W_1 & W_1 & 0 \\ W_T & W_E & 0 \end{bmatrix} \cdot \begin{bmatrix} I_{dG^2} & 0_{dG^2\times dG^2} \\ 0_{dG^2\times dG^2} & I_{dG^2} \\ X_K & X_K \end{bmatrix} \qquad \text{(By (E.28))}$$

$$= \begin{bmatrix} X_K & X_K \\ W_0 & W_0 \\ W_1 & W_1 \\ W_T & W_E \end{bmatrix}.$$

Next, we construct the $W_Q$ matrix as

$$W_Q := \begin{bmatrix} 0_{1\times G} & R1_{1\times n} \\ 0_{1\times G} & R1_{1\times n} \\ I_G & 0_{1\times n} \\ 0_{1\times G} & I_n \end{bmatrix}.$$

In this definition, the $Q$ matrix in attention can be calculated as follows

$$
\begin{aligned}
Q &:= W_Q \text{Linear}_Q(Z_Q) \\
&= \begin{bmatrix} 0_{1\times G} & R1_{1\times n} \\ 0_{1\times G} & R1_{1\times n} \\ I_G & 0_{1\times n} \\ 0_{1\times G} & I_n \end{bmatrix} \cdot \begin{bmatrix} X_Q & 0_{G\times(2dG^2-n)} \\ I_n & 0_{n\times(2dG^2-n)} \end{bmatrix} \qquad (\text{By (E.29)}) \\
&= \begin{bmatrix} R1_{1\times n} & 0_{1\times(2dG^2-n)} \\ R1_{1\times n} & 0_{1\times(2dG^2-n)} \\ X_Q & 0_{G\times(2dG^2-n)} \\ I_n & 0_{n\times(2dG^2-n)} \end{bmatrix}.
\end{aligned}
$$

Now we calculate the attention matrix $\text{Softmax}\left(K^\top Q\right)$.

$K^\top Q$ can be calculated as follows

$$
\begin{aligned}
K^\top Q &= \begin{bmatrix} X_K & X_K \\ W_0 & W_0 \\ W_1 & W_1 \\ W_T & W_E \end{bmatrix}^\top \begin{bmatrix} R1_{1\times n} & 0_{1\times(2dG^2-n)} \\ R1_{1\times n} & 0_{1\times(2dG^2-n)} \\ X_Q & 0_{G\times(2dG^2-n)} \\ I_n & 0_{n\times(2dG^2-n)} \end{bmatrix} \\
&= \begin{bmatrix} (RX_K^\top + RW_0^\top)1_{1\times n} + W_1^\top X_Q + W_T^\top & 0_{dG^2\times(2dG^2-n)} \\ (RX_K^\top + RW_0^\top)1_{1\times n} + W_1^\top X_Q + W_E^\top & 0_{dG^2\times(2dG^2-n)} \end{bmatrix}.
\end{aligned}
$$

The $W_1^\top X_Q$ in the expression of $K^\top Q$ matrix is further calculated as

$$
\begin{aligned}
W_1^\top X_Q &= \begin{bmatrix} \overline{W}_1 & \overline{W}_1 & \overline{W}_1 & \cdots & \overline{W}_1 \end{bmatrix}_{G\times dG^2}^\top X_Q \\
&= \begin{bmatrix} \overline{W}_1^\top X_Q \\ \overline{W}_1^\top X_Q \\ \vdots \\ \overline{W}_1^\top X_Q \end{bmatrix}_{dG^2\times G}.
\end{aligned}
$$

We define $Q_1 := \overline{W}_1^\top X_Q$, then $W_1^\top X_Q$ can be denoted as stacking this block vertically for $G$ times.

In this definition, $Q_1$ matrix can be expressed as

$$
\begin{aligned}
Q_1 &:= \overline{W}_1^\top X_Q \\
&= \begin{bmatrix} Re_1^{(G)}1_{1\times d} & Re_2^{(G)}1_{1\times d} & \cdots & Re_G^{(G)}1_{1\times d} \end{bmatrix}^\top \begin{bmatrix} v_0^\top \widetilde{Z}_Q 1_{1\times n} \\ v_1^\top \widetilde{Z}_Q 1_{1\times n} \\ v_2^\top \widetilde{Z}_Q 1_{1\times n} \\ \vdots \\ v_{G-1}^\top \widetilde{Z}_Q 1_{1\times n} \end{bmatrix} \\
&= \begin{bmatrix} Re_1^{(G)}1_d \\ Re_2^{(G)}1_d \\ \vdots \\ Re_G^{(G)}1_d \end{bmatrix} \cdot \begin{bmatrix} v_0^\top \widetilde{Z}_Q 1_{1\times n} \\ v_1^\top \widetilde{Z}_Q 1_{1\times n} \\ v_2^\top \widetilde{Z}_Q 1_{1\times n} \\ \vdots \\ v_{G-1}^\top \widetilde{Z}_Q 1_{1\times n} \end{bmatrix}.
\end{aligned}
$$

$$= \begin{bmatrix} Rv_0^\top \widetilde{Z}_Q 1_{d \times n} \\ Rv_1^\top \widetilde{Z}_Q 1_{d \times n} \\ Rv_2^\top \widetilde{Z}_Q 1_{d \times n} \\ \vdots \\ Rv_{G-1}^\top \widetilde{Z}_Q 1_{d \times n} \end{bmatrix}. \tag{E.31}$$

The calculation of $\text{Softmax}(K^\top Q)$ can be disassembled into two parts, the numerator $\exp(\text{Softmax}(K^\top Q))$ in the expression of Softmax and the denominator of every column of $\text{Softmax}(K^\top Q)$, as in the expression of Softmax, explicitly written out as $\sum_{j=1}^{2dG} \exp(K^\top Q)_{ij}$ for each $i \in [2dG]$.

We calculate $\exp(K^\top Q)$ as follows

$$\exp(K^\top Q) = \begin{bmatrix} \exp\big((RX_K^\top + RW_0^\top)1_{1 \times n} + RW_1^\top X_Q\big) \odot \exp(W_T^\top) & 1_{dG^2 \times (2dG^2 - n)} \\ \exp\big((RX_K^\top + RW_0^\top)1_{1 \times n} + RW_1^\top X_Q\big) \odot \exp(W_E^\top) & 1_{dG^2 \times (2dG^2 - n)} \end{bmatrix}. \tag{E.32}$$

For the denominator, we calculate it in columns. Let $i$ denote the column which we calculate the denominator in Softmax. When $i \in \{n+1, n+2, \cdots, 2dG^2\}$, the $i$-th column has $1$ in every entry. Thus the sum of all entries in this column equals to $1 \cdot 2dG = 2dG$.

And when $i \in [n]$, we have

$$\sum_{j=1}^{2dG^2} \exp(K^\top Q)_{i,j}$$

$$= \sum_{j_1=1}^{G} \sum_{j_2=1}^{G} \left[ \big(1_{1 \times d} T(\widetilde{v}_{j_1-1}, \widetilde{v}_{j_2-1})_{:,i} + 1_{1 \times d} E(\widetilde{v}_{j_1-1}, \widetilde{v}_{j_2-1})_{:,i}\big) \right.$$

$$\left. \cdot \exp\left( R\left( v_{j_1-1}^\top \widetilde{Z}_K - \frac{\|v_{j_1-1}\|_2^2}{2} + v_{j_2-1}^\top \widetilde{Z}_Q - \frac{\|v_{j_2-1}\|_2^2}{2} \right) \right) \right]$$

$$= \sum_{j_1=1}^{G} \sum_{j_2=1}^{G} \left[ 1_{1 \times d}(E + T)(v_{j_1-1}, v_{j_2-1})_{:,i} \cdot \exp\left( R\left( v_{j_1-1}^\top \widetilde{Z}_K - \frac{\|v_{j_1-1}\|_2^2}{2} + v_{j_2-1}^\top \widetilde{Z}_Q - \frac{\|v_{j_2-1}\|_2^2}{2} \right) \right) \right]$$

$$= \sum_{j_1=1}^{G} \sum_{j_2=1}^{G} \left[ (1_{1 \times d}(2_{d \times n})_{:,i}) \cdot \exp\left( R\left( v_{j_1-1}^\top \widetilde{Z}_K - \frac{\|v_{j_1-1}\|_2^2}{2} + v_{j_2-1}^\top \widetilde{Z}_Q - \frac{\|v_{j_2-1}\|_2^2}{2} \right) \right) \right]$$

$$= \sum_{j_1=1}^{G} \sum_{j_2=1}^{G} 2d \cdot \exp\left( R\left( v_{j_1-1}^\top \widetilde{Z}_K - \frac{\|v_{j_1-1}\|_2^2}{2} + v_{j_2-1}^\top \widetilde{Z}_Q - \frac{\|v_{j_2-1}\|_2^2}{2} \right) \right), \quad i \in [n]. \tag{E.33}$$

We observe from (E.33), that $\sum_{j=1}^{2dG^2} \exp(K^\top Q)_{ij}$ is **invariant** of $i$ for $i \in [n]$. In this case, we define

$$\alpha(Z_K, Z_Q) := \frac{1}{2d} \sum_{j=1}^{2dG^2} \exp(K^\top Q)_{i,j}$$

$$= \sum_{j_1=1}^{G} \sum_{j_2=1}^{G} \exp\left( R\left( v_{j_1-1}^\top \widetilde{Z}_K - \frac{\|v_{j_1-1}\|_2^2}{2} + v_{j_2-1}^\top \widetilde{Z}_Q - \frac{\|v_{j_2-1}\|_2^2}{2} \right) \right)$$

to denote the $1/2d$ of this value invariant of $i$ for simplicity.

Because

$$\alpha(Z_K, Z_Q) = \frac{1}{2d} \sum_{j=1}^{2dG^2} \exp(K^\top Q)_{i,j},$$

from (E.32) and (E.33) we have

$$\text{Softmax}\left(K^\top Q\right)$$

$$= \underbrace{\exp\left(K^\top Q\right)}_{\text{nominator of Softmax}} \odot \underbrace{\left[\frac{1}{\sum_{j=1}^{2dG^2}\exp(K^\top Q)_{1j}}1_{2dG\times n} \quad \frac{1}{2dG^2}1_{2dG\times(2dG-n)}\right]}_{\text{denominator of Softmax}}$$

$$\left(\text{By } \frac{1}{\sum_{j=1}^{2dG}\exp\left(K^\top Q\right)_{ij}} \text{ is invariant of } i \text{ for } i\in[n]\right)$$

$$= \exp\left(K^\top Q\right) \odot \left[\frac{1}{2d\alpha(Z_K,Z_Q)}1_{2dG\times n} \quad \frac{1}{2dG^2}1_{2dG\times(2dG-n)}\right]$$

$$= \begin{bmatrix} \exp\left((RX_K^\top + RW_0^\top)1_{1\times n} + RW_1^\top X_Q\right)\odot\exp\left(W_T^\top\right) & 1_{dG^2\times(2dG^2-n)} \\ \exp\left((RX_K^\top + RW_0^\top)1_{1\times n} + RW_1^\top X_Q\right)\odot\exp\left(W_E^\top\right) & 1_{dG^2\times(2dG^2-n)} \end{bmatrix}$$

$$\odot \underbrace{\left[\frac{1}{2d\alpha(Z_K,Z_Q)}1_{2dG\times n} \quad \frac{1}{2dG^2}1_{2dG\times(2dG-n)}\right]}_{\text{denominator in Softmax}} \qquad \left(\text{By (E.32)}\right)$$

$$= \begin{bmatrix} \frac{1}{2d\alpha(Z_K,Z_Q)}\exp\left((RX_K^\top + RW_0^\top)1_{1\times n} + RW_1^\top X_Q\right)\odot\exp\left(W_T^\top\right) & \frac{1}{2dG^2}1_{dG^2\times(2dG^2-n)} \\ \frac{1}{2d\alpha(Z_K,Z_Q)}\exp\left((RX_K^\top + RW_0^\top)1_{1\times n} + RW_1^\top X_Q\right)\odot\exp\left(W_E^\top\right) & \frac{1}{2dG^2}1_{dG^2\times(2dG^2-n)} \end{bmatrix}.$$

Now we've defined and calculated the attention score matrix $\text{Softmax}(K^\top Q)$, we go on to construct the $W_V$ matrix and calculate the result of multiplying $V = W_V\text{Linear}(Z_K)$ to the attention score matrix.

We define $W_V$ as

$$W_V := \underbrace{\begin{bmatrix} X_2 & -X_2 & 0_d \end{bmatrix}}_{d\times(2dG^2+1)},$$

where

$$X_2 = \underbrace{\begin{bmatrix} I_d & I_d & \cdots & I_d \end{bmatrix}}_{d\times dG^2}.$$

This yields the $V$ matrix to be

$$V = W_V\text{Linear}(Z_K)$$

$$= \underbrace{\begin{bmatrix} X_2 & -X_2 & 0_d \end{bmatrix}}_{d\times(2dG^2+1)} \cdot \underbrace{\begin{bmatrix} I_{dG^2} & 0_{dG^2\times dG^2} \\ 0_{dG^2\times dG^2} & I_{dG^2} \\ X_K & X_K \end{bmatrix}}_{(2dG^2+1)\times 2dG^2}$$

$$= \underbrace{\begin{bmatrix} X_2 & -X_2 \end{bmatrix}}_{d\times 2dG^2} \cdot \underbrace{\begin{bmatrix} I_{dG^2} & 0_{dG^2\times dG^2} \\ 0_{dG^2\times dG^2} & I_{dG^2} \end{bmatrix}}_{2dG^2\times 2dG^2} \qquad \left(\text{since } X_K \text{ is multiplied by } 0\right)$$

$$= \begin{bmatrix} X_2 & -X_2 \end{bmatrix}.$$

With $V$, we compute the output of $V\,\text{Softmax}(K^\top Q)$ as follows

$$V\,\text{Softmax}(K^\top Q)$$

$$= \begin{bmatrix} X_2 & -X_2 \end{bmatrix} \cdot \begin{bmatrix} \frac{1}{2d\alpha(Z_K,Z_Q)}\exp\left((RX_K^\top + RW_0^\top)1_{1\times n} + RW_1^\top X_Q\right)\odot\exp\left(W_T^\top\right) & \frac{1}{2dG^2}1_{dG^2\times(2dG^2-n)} \\ \frac{1}{2d\alpha(Z_K,Z_Q)}\exp\left(RX_K^\top + RW_0^\top\right)1_{1\times n}\odot\exp\left(RW_1^\top X_Q\right)\odot\exp\left(W_E^\top\right) & \frac{1}{2dG^2}1_{dG^2\times(2dG^2-n)} \end{bmatrix}$$

$$= X_2\left[\frac{1}{2d\alpha(Z_K,Z_Q)}\exp\left((RX_K^\top + RW_0^\top)1_{1\times n} + RW_1^\top X_Q\right)\odot\exp\left(W_T^\top\right) \quad \frac{1}{2dG^2}1_{dG^2\times(2dG^2-n)}\right]$$

$$\underbrace{-X_2}_{-X_2 \text{ in } [X_2\ -X_2]}\left[\frac{1}{2d\alpha(Z_K,Z_Q)}\exp\left(RX_K^\top + RW_0^\top\right)1_{1\times n}\odot\exp\left(RW_1^\top X_Q\right)\odot\exp\left(W_E^\top\right) \quad \frac{1}{2dG^2}1_{dG^2\times(2dG^2-n)}\right]$$

$$= \frac{1}{2d\alpha(Z_K, Z_Q)} X_2 \left[ \exp\left((RX_K^\top + RW_0^\top)1_{1\times n} + RW_1^\top X_Q\right) \odot \left[\exp(W_T^\top) - \exp(W_E^\top)\right] \quad 0_{dG^2 \times (2dG^2 - n)} \right].$$

(E.34)

To further calculate $V \, \text{Softmax}(K^\top Q)$, we now calculate the result of its non-trivial part (the part beside $0_{dG^2 \times (2dG^2 - n)}$)

$$X_2 \left[ \exp\left((RX_K^\top + RW_0^\top)1_{1\times n} + RW_1^\top X_Q\right) \odot \left[\exp(W_T^\top) - \exp(W_E^\top)\right] \right].$$

(E.35)

We now calculate each part in (E.35)

$$\exp(W_T^\top) - \exp(W_E^\top)$$
$$= \left(\exp\left(\begin{bmatrix} W_T^{(0)} & W_T^{(1)} & \cdots & W_T^{(G-1)} \end{bmatrix}\right) - \exp\left(\begin{bmatrix} W_E^{(0)} & W_E^{(1)} & \cdots & W_E^{(G-1)} \end{bmatrix}\right)\right)^\top$$
$$= \begin{bmatrix} \exp\left(W_T^{(0)}\right)^\top - \exp\left(W_E^{(0)}\right)^\top \\ \exp\left(W_T^{(1)}\right)^\top - \exp\left(W_E^{(1)}\right)^\top \\ \vdots \\ \exp\left(W_T^{(G-1)}\right)^\top - \exp\left(W_E^{(G-1)}\right)^\top \end{bmatrix}.$$

(E.36)

In (E.36), we have

$$\exp\left(W_T^{(i)}\right)^\top - \exp\left(W_E^{(i)}\right)^\top = \begin{bmatrix} \exp(\ln(T(\widetilde{v}_i, v_0))) - \exp(\ln(E(\widetilde{v}_i, v_0))) \\ \exp(\ln(T(\widetilde{v}_i, v_1))) - \exp(\ln(E(\widetilde{v}_i, v_0))) \\ \vdots \\ \exp(\ln(T(\widetilde{v}_i, v_{G-1}))) - \exp(\ln(E(\widetilde{v}_i, v_0))) \end{bmatrix}$$
$$= \begin{bmatrix} T(\widetilde{v}_i, v_0) - E(\widetilde{v}_i, v_0) \\ T(\widetilde{v}_i, v_1) - E(\widetilde{v}_i, v_0) \\ \vdots \\ T(\widetilde{v}_i, v_{G-1}) - E(\widetilde{v}_i, v_0) \end{bmatrix}$$
$$= \begin{bmatrix} \frac{2f(\widetilde{v}_i, v_0)}{B_0} \\ \frac{2f(\widetilde{v}_i, v_1)}{B_0} \\ \vdots \\ \frac{2f(\widetilde{v}_i, v_{G-1})}{B_0} \end{bmatrix}.$$

Thus (E.36) is equal to

$$\left(\exp\left(W_T^{(i)}\right)^\top - \exp\left(W_E^{(i)}\right)^\top\right)_{(i-1)G+1:iG,:} = \begin{bmatrix} \frac{2f(\widetilde{v}_{i-1}, v_0)}{B_0} \\ \frac{2f(\widetilde{v}_{i-1}, v_1)}{B_0} \\ \vdots \\ \frac{2f(\widetilde{v}_{i-1}, v_{G-1})}{B_0} \end{bmatrix}, \quad i \in [G].$$

(E.37)

We also calculate the other part $\exp\left((RX_K^\top + RW_0^\top)1_{1\times n} + RW_1^\top X_Q\right)$ in separate parts

$$\exp\left((RX_K^\top + RW_0^\top)1_{1\times n}\right)_{idG+jd+1:idG+(j+1)d,:} = \begin{bmatrix} \exp\left(v_0^\top \widetilde{Z}_K - \frac{\|v_0\|_2^2}{2}\right)1_{dG\times n} \\ \exp\left(v_1^\top \widetilde{Z}_K - \frac{\|v_1\|_2^2}{2}\right)1_{dG\times n} \\ \cdots \\ \exp\left(v_{G-1}^\top \widetilde{Z}_K - \frac{\|v_{G-1}\|_2^2}{2}\right)1_{dG\times n} \end{bmatrix}_{idG+jd+1:idG+(j+1)d,:}$$
$$= \exp\left(v_i^\top \widetilde{Z}_K - \frac{\|v_i\|_2^2}{2}\right)1_{d\times n},$$

and

$$\exp\big((RW_1^\top X_Q)1\big)_{idG+jd+1:idG+(j+1)d,:} = \begin{bmatrix} Q_1 \\ Q_1 \\ \vdots \\ Q_1 \end{bmatrix}_{idG+jd+1:idG+(j+1)d,:}$$

$$\left(\text{This is a stack of } G\ Q_1 \text{ in (E.31)}\right)$$

$$= (Q_1)_{jd+1:(j+1)d,:}$$

$$= v_j^\top \widetilde{Z}_Q 1_{d\times n}.$$

Thus

$$\exp\big((RX_K^\top + RW_0^\top)1_{1\times n} + RW_1^\top X_Q\big)_{idG+jd+1:idG+(j+1)d,:}$$

$$= \exp\left(R(v_i^\top \widetilde{Z}_K - \frac{\|v_i\|_2^2}{2} - \frac{\|v_j\|_2^2}{2} + v_j^\top \widetilde{Z}_Q)\right)1_{d\times n}, \quad i,j \in \{0,1,\cdots,G-1\}. \qquad \text{(E.38)}$$

Combing (E.37) and (E.38), we have

$$\big[\exp\big((RX_K^\top + RW_0^\top)1_{1\times n} + RW_1^\top X_Q\big) \odot [\exp(W_T^\top) - \exp(W_E^\top)]\big]_{idG+(j-1)d+1:idG+jd,:}$$

$$= \exp\left(R(v_i^\top \widetilde{Z}_K - \frac{\|v_i\|_2^2}{2} - \frac{\|v_j\|_2^2}{2} + v_j^\top \widetilde{Z}_Q)\right)1_{d\times n} \odot \frac{2f(\widetilde{v}_{i-1}, v_{j-1})}{B_0}, \quad i,j \in \{0,1,\cdots,G-1\}.$$

Thus we compute (E.35) as

$$\left(X_2 \big[\exp\big((RX_K^\top + RW_0^\top)1_{1\times n} + RW_1^\top X_Q\big) \odot [\exp(W_T^\top) - \exp(W_E^\top)]\big]\right)$$

$$= \sum_{i=0}^{G-1}\sum_{j=0}^{G-1} (X_2)_{:,idG+jd+1:idG+(j+1)d}$$

$$\cdot R\big[\exp\big((RX_K^\top + RW_0^\top)1_{1\times n} + RW_1^\top X_Q\big) \odot [\exp(W_T^\top) - \exp(W_E^\top)]\big]_{idG+(j-1)d+1:idG+jd,:}$$

$$= \sum_{i=0}^{G-1}\sum_{j=0}^{G-1} I_d \cdot \exp\left(R(v_i^\top \widetilde{Z}_K - \frac{\|v_i\|_2^2}{2} - \frac{\|v_j\|_2^2}{2} + v_j^\top \widetilde{Z}_Q)\right)1_{d\times n} \odot \frac{2f(\widetilde{v}_{i-1}, v_{j-1})}{B_0}$$

$$\left(\text{Because } X_2 \text{ is a horizontal stack of } I_d\right)$$

$$= \sum_{i=0}^{G-1}\sum_{j=0}^{G-1} \exp\left(R(v_i^\top \widetilde{Z}_K - \frac{\|v_i\|_2^2}{2} - \frac{\|v_j\|_2^2}{2} + v_j^\top \widetilde{Z}_Q)\right)1_{d\times n} \odot \frac{2f(\widetilde{v}_{i-1}, v_{j-1})}{B_0}.$$

We now put back the $1/2d\alpha(Z_K, Z_Q)$ in (E.34) and calculate the final output as

$$V \operatorname{Softmax} K^\top Q$$

$$= \frac{1}{2d\alpha(Z_K, Z_Q)} X_2 \left[\exp\big((RX_K^\top + RW_0^\top)1_{1\times n} + RW_1^\top X_Q\big) \odot [\exp(W_T^\top) - \exp(W_E^\top)] \quad 0_{dG^2\times(2dG^2-n)}\right]$$

$$= \frac{1}{2d\alpha(Z_K, Z_Q)} \sum_{i=0}^{G-1}\sum_{j=0}^{G-1}\left[\exp\left(R(v_i^\top \widetilde{Z}_K - \frac{\|v_i\|_2^2}{2} - \frac{\|v_j\|_2^2}{2} + v_j^\top \widetilde{Z}_Q)\right)1_{d\times n} \odot \underbrace{\frac{2f(\widetilde{v}_{i-1}, v_{j-1})}{B_0}}_{\exp(W_T^\top)-\exp(W_E^\top)} \quad 0_{d\times(2dG^2-n)}\right]$$

$$= \left[\frac{1}{dB_0}\sum_{i=0}^{G-1}\sum_{j=0}^{G-1} \frac{\exp\left(R(v_i^\top \widetilde{Z}_K - \frac{\|v_i\|_2^2}{2} - \frac{\|v_j\|_2^2}{2} + v_j^\top \widetilde{Z}_Q)\right)1_{d\times n}\odot\frac{f(\widetilde{v}_{i-1}, v_{j-1})}{B_0}}{\alpha(Z_K, Z_Q)} \quad 0_{d\times(2dG^2-n)}\right].$$

Next, we construct $W_O$ to be

$$W_O := \begin{bmatrix} dB_0 I_n \\ 0_{(2dG^2-n)\times n} \end{bmatrix}.$$

This yields the final output of $\text{Attn} \circ \text{Linear}$ to be

$$\text{Attn} \circ \text{Linear}(Z)$$

$$= V \, \text{Softmax} \, K^\top Q W_O$$

$$= \left[ \frac{1}{dB_0} \sum_{i=0}^{G-1} \sum_{j=0}^{G-1} \frac{\exp\left( R(v_i^\top \widetilde{Z}_K - \frac{\|v_i\|_2^2}{2} - \frac{\|v_j\|_2^2}{2} + v_j^\top \widetilde{Z}_Q) \right) 1_{d \times n} \odot \frac{f(\widetilde{v_{i-1}, v_{j-1}})}{B_0}}{\alpha(Z_K, Z_Q)} \quad 0_{d \times (2dG^2 - n)} \right] \cdot \underbrace{\begin{bmatrix} dB_0 I_n \\ 0_{(2dG^2 - n) \times n} \end{bmatrix}}_{W_O}$$

$$= \sum_{i=0}^{G-1} \sum_{j=0}^{G-1} \frac{\exp\left( R(v_i^\top \widetilde{Z}_K - \frac{\|v_i\|_2^2}{2} - \frac{\|v_j\|_2^2}{2} + v_j^\top \widetilde{Z}_Q) \right) 1_{d \times n} \odot \frac{f(\widetilde{v_{i-1}, v_{j-1}})}{B_0}}{\alpha(Z_K, Z_Q)}. \tag{E.39}$$

**Estimation of Error between** $\text{Attn} \circ \text{Linear}$ **and** $f$    We now calculate the loss between the result in (E.39) and the target function $f$. For simplicity, we first define $\widetilde{Z} := [[\widetilde{Z}_K^\top, \widetilde{Z}_Q^\top]^\top]$ to accommodate to the expression of affine functions.

**Definition E.3** (Max-Affine Function on $\widetilde{Z}$.).    Let $\text{Aff}_{i,j} \in \mathbb{R}^{dn} \to \mathbb{R}$, $j \in \{0.1.2.\cdots, G-1\}$ denote a group of affine functions defined as

$$\text{Aff}_{i,j}(\widetilde{Z}) = v_i^\top \widetilde{Z}_K + v_j^\top \widetilde{Z}_Q - \frac{1}{2}\|v_i\|_2^2 - \frac{1}{2}\|v_j\|_2^2, \quad i,j \in \{0,1,2,\cdots, G-1\}.$$

Then let $\text{MaxAff} \in \mathbb{R}^{dn} \to \mathbb{R}$ denote a max affine function whose affine components are $\{\text{Aff}_{i,j} | i,j \in \{0,1,2,\cdots, G-1\}\}$. Explicitly defined as:

$$\text{MaxAff}(\widetilde{Z}) = \max_{i,j \in \{0,1,2,\cdots,G-1\}} \left\{ \text{Aff}_{i,j}(\widetilde{Z}) \right\}.$$

In the following discussion, we use $\eta \in \{0,1,\cdots,G-1\}^2$ to refer to a pair of coefficients $(i,j)$, and denote $A_{i,j}$ as $A_\eta$ for the corresponding $\eta$. Furthermore, we denote the two labels encapsulated in $\eta$ as $i_\eta$ and $j_\eta$

Because the target function $f$ is a continuous function on a closed domain, the function $f$ is uniformly continuous. Thus for $\epsilon$, there exists a $\delta > 0$ such that for any $Z^{(1)} = [Z_K^{(1)}, Z_Q^{(1)}]$, $Z^{(2)} = [Z_K^{(2)}, Z_Q^{(2)}]$, as long as $\|Z^{(1)} - Z^{(2)}\|_\infty \le \delta$, we have $\|f(Z^{(1)}) - f(Z^{(1)})\|_\infty \le \epsilon/3$.

According to this $\delta$, we divide the affine components of $\text{MaxAff}$ into three parts, the maximal component (and also with the smallest label on both entry), whose label is denoted as $\eta_m$, the group of affine components equal to the maximal component or smaller than it by no more than $\delta$, and finally, the other $\text{Aff}_\eta$. We write out the labels of these groups of components as follows

$$\eta_m := \min_{\eta \in \{0,1,2,\cdots,G-1\}^2} \{\text{Aff}_\eta(\widetilde{Z}) = \text{MaxAff}(\widetilde{Z})\},$$

$$E_0 := \{\eta \mid \text{MaxAff}(\widetilde{Z}) - \text{Aff}_\eta(\widetilde{Z}) \le \delta\},$$

$$E_1 := \{\eta \mid \text{MaxAff}(\widetilde{Z}) - \text{Aff}_\eta(\widetilde{Z}) > \delta\}.$$

For any pair of $\eta_1, \eta_2 \in \{0,1,\cdots,G-1\}^2$, we denote that

$$\text{Aff}_{\eta_1}(\widetilde{Z}) - \text{Aff}_{\eta_2}(\widetilde{Z})$$

$$= v_{i_{\eta_1}}^\top \widetilde{Z}_K - \frac{\|v_{i_{\eta_1}}\|_2^2}{2} + v_{j_{\eta_1}}^\top \widetilde{Z}_Q - \frac{\|v_{j_{\eta_1}}\|_2^2}{2} - \left( v_{i_{\eta_2}}^\top \widetilde{Z}_K - \frac{\|v_{i_{\eta_2}}\|_2^2}{2} + v_{j_{\eta_2}}^\top \widetilde{Z}_Q - \frac{\|v_{j_{\eta_2}}\|_2^2}{2} \right)$$

$$= -\frac{\|\widetilde{Z}_K\|_2^2}{2} + v_{i_{\eta_1}}^\top \widetilde{Z}_K - \frac{\|v_{i_{\eta_1}}\|_2^2}{2} - \frac{\|\widetilde{Z}_Q\|_2^2}{2} + v_{j_{\eta_1}}^\top \widetilde{Z}_Q - \frac{\|v_{j_{\eta_1}}\|_2^2}{2} \tag{E.40}$$

$$\quad - \left( -\frac{\|\widetilde{Z}_K\|_2^2}{2} + v_{i_{\eta_2}}^\top \widetilde{Z}_K - \frac{\|v_{i_{\eta_2}}\|_2^2}{2} - \frac{\|\widetilde{Z}_Q\|_2^2}{2} + v_{j_{\eta_2}}^\top \widetilde{Z}_Q - \frac{\|v_{j_{\eta_2}}\|_2^2}{2} \right)$$

$$= -\frac{1}{2}\|\widetilde{Z}_K - v_{i_{\eta_1}}\|_2^2 - \frac{1}{2}\|\widetilde{Z}_Q - v_{j_{\eta_1}}\|_2^2 + \frac{1}{2}\|\widetilde{Z}_K - v_{i_{\eta_2}}\|_2^2 + \frac{1}{2}\|\widetilde{Z}_Q - v_{j_{\eta_2}}\|_2^2$$

$$= \frac{1}{2} \|\widetilde{Z} - \begin{bmatrix} v_{i_{\eta_2}} \\ v_{j_{\eta_2}} \end{bmatrix} \|_2^2 - \frac{1}{2} \|\widetilde{Z} - \begin{bmatrix} v_{i_{\eta_1}} \\ v_{j_{\eta_1}} \end{bmatrix} \|_2^2. \tag{E.41}$$

Let $v_\eta := [v_{i_\eta}^\top, v_{j_\eta}^\top]^\top$, denote a flatten stack of $v_{i_\eta}$ and $v_{j_\eta}$. Same as $v_i$, define $\widetilde{v}_\eta := [\widetilde{v}_{i_\eta}, \widetilde{v}_{j_\eta}]$. Then the above expression denotes $\eta_m$ is also the label of the $v_\eta$ closest to $\widetilde{Z}$ among all $v_\eta$, $\eta \in \{0, 1, \cdots, G-1\}^2$. Thus we have

$$\|v_{\eta_m} - \widetilde{Z}\|_2 = \min_{\eta \in \{0,1,\cdots,G-1\}^2} \{\|v_\eta - \widetilde{Z}\|_2\}. \tag{E.42}$$

This means that $v_{\eta_m}$ is the grid center closest to $\widetilde{Z}$ in 2-norm.

We now prove this closest grid center has a distance to $\widetilde{Z}$ smaller than half of the grid width $(D/g)$ in infinite norm.

Let $\mathcal{D} := 2D/g \times \{-1, 0, 1\}^{dn}$ denote a set of differences to $v_{\eta_m}$ of all the $v_i$ ($i \in \{0, 1, \cdots, G-1\}$) neighboring $v_{\eta_m}$. For any $\Delta$ in $\mathcal{D}$, from (E.42) we have

$$\|v_{\eta_m} - \widetilde{Z}\|_2^2 \le \|v_{\eta_m} + \Delta - \widetilde{Z}\|_2^2.$$

This yields

$$2\Delta^\top (\widetilde{Z} - v_{j_m}) \le \|\Delta\|_2^2,$$

which means for any $k \in [dn]$, by selecting $\Delta$ to be $\pm \frac{2D}{g} \cdot e_k^{(dn)}$, we have

$$\pm 2 \times \frac{2D}{g} (\widetilde{Z} - v_{\eta_m})_k = 2\Delta^\top (\widetilde{Z} - v_{\eta_m}) \le \|\Delta\|_2^2 = \frac{4D^2}{g^2}.$$

Thus we have

$$(|\widetilde{Z} - v_{\eta_m}|)_k \le \frac{D}{g}, \ k \in [dn].$$

This is equivalent to

$$\|\widetilde{Z} - v_{\eta_m}\|_\infty \le \frac{D}{g}, \ k \in [dn].$$

Set $g$ to be larger than $2D/\delta$, we have

$$\|\widetilde{Z} - v_{\eta_m}\|_\infty \le \frac{\delta}{2},$$

thus

$$\|f(Z) - f(\widetilde{v}_{\eta_m})\|_\infty \le \frac{\epsilon}{3}. \qquad \left(\text{because } \delta/2 < \delta\right)$$

**Calculation of** $\|\text{Attn} \circ \text{Linear} - f\|_{L_\infty}$**.** We now calculate the difference between the output in (E.39) and target function $f$

$$\|\text{Attn} \circ \text{Linear}(Z) - f(Z)\|_\infty = \|\sum_{\eta=0}^{G-1} \frac{\exp\left(R(v_\eta^\top \widetilde{Z} - \frac{\|v_\eta\|_2^2}{2})\right)}{\alpha(Z)} f(\widetilde{v}_\eta) - f(Z)\|_\infty$$

$$= \|\sum_{\eta=0}^{G-1} \frac{\exp\left(R(v_\eta^\top \widetilde{Z} - \frac{\|v_\eta\|_2^2}{2})\right)}{\alpha(Z)} (f(\widetilde{v}_\eta) - f(Z))\|$$

$$\left(\text{By } \sum_{\eta=0}^{G-1} \frac{\exp\left(R(v_\eta^\top \widetilde{Z} - \frac{\|v_\eta\|_2^2}{2})\right)}{\alpha(Z)} = 1\right)$$

$$\leq \sum_{\eta=0}^{G-1} \frac{\exp\Big(R(v_\eta^\top \widetilde{Z} - \frac{\|v_\eta\|_2^2}{2})\Big)}{\alpha(Z)} \|f(\widetilde{v}_\eta) - f(Z)\|_\infty$$

(By property of infinite norm)

$$= \frac{\exp\Big(R(v_{\eta_m}^\top \widetilde{Z} - \frac{\|v_{\eta_m}\|_2^2}{2})\Big)}{\alpha(Z)} \|f(\widetilde{v}_{\eta_m}) - f(Z)\|_\infty$$

$$+ \sum_{\eta \in \eta_0} \frac{\exp\Big(R(v_\eta^\top \widetilde{Z} - \frac{\|v_\eta\|_2^2}{2})\Big)}{\alpha(Z)} \|f(\widetilde{v}_\eta) - f(Z)\|_\infty$$

$$+ \sum_{\eta \in \eta_1} \frac{\exp\Big(R(v_\eta^\top \widetilde{Z} - \frac{\|v_\eta\|_2^2}{2})\Big)}{\alpha(Z)} \|f(\widetilde{v}_\eta) - f(Z)\|_\infty. \quad \text{(E.43)}$$

The last row is simply a separation of the summation in the row above.

We now calculate each part in (E.43).

As previously stated, for any $Z_1, Z_2$, as long as $\|\widetilde{Z}_1 - \widetilde{Z}_2\|_\infty \leq \delta$, we have $\|f(Z_1) - f(Z_2)\|_\infty \leq \epsilon/3$. Thus when we designate $Z_1 = v_\eta$ for any $\eta \in \eta_0$ and $Z_2 = v_{\eta_m}$, along with (E.40) we have

$$\sum_{\eta \in \eta_0} \frac{\exp\Big(R(v_\eta^\top \widetilde{Z} - \frac{\|v_\eta\|_2^2}{2})\Big)}{\alpha(Z)} \|f(\widetilde{v}_\eta) - f(Z)\|_\infty$$

$$\leq \sum_{\eta \in \eta_0} \frac{\exp\Big(R(v_\eta^\top \widetilde{Z} - \frac{\|v_\eta\|_2^2}{2})\Big)}{\alpha(Z)} (\|f(\widetilde{v}_\eta) - f(\widetilde{v}_{\eta_m})\|_\infty + \|f(\widetilde{v}_{\eta_m}) - f(Z)\|_\infty)$$

$$\leq \sum_{\eta \in \eta_0} \frac{\exp\Big(R(v_\eta^\top \widetilde{Z} - \frac{\|v_\eta\|_2^2}{2})\Big)}{\alpha(Z)} \cdot (\frac{\epsilon}{3} + \frac{\epsilon}{3})$$

$$= \sum_{\eta \in \eta_0} \frac{\exp\Big(R(v_\eta^\top \widetilde{Z} - \frac{\|\widetilde{v}_\eta\|_2^2}{2})\Big)}{\alpha(Z)} \cdot \frac{2\epsilon}{3}. \quad \text{(E.44)}$$

For any $\eta_m$, we have

$$\frac{\exp\Big(R(v_{\eta_m}^\top \widetilde{Z} - \frac{\|v_{\eta_m}\|_2^2}{2})\Big)}{\alpha(Z)} \|f(\widetilde{v}_{\eta_m}) - f(Z)\|_\infty \leq \frac{\exp\Big(R(v_{\eta_m}^\top \widetilde{Z} - \frac{\|v_{\eta_m}\|_2^2}{2})\Big)}{\alpha(Z)} \cdot \frac{\epsilon}{3}. \quad \text{(E.45)}$$

When $R$ is larger than $8\ln(3/2 \cdot B_0 G\epsilon)/(3\delta^2)$, we have

$$\sum_{\eta \in \eta_1} \frac{\exp\Big(R(v_\eta^\top \widetilde{Z} - \frac{\|v_\eta\|_2^2}{2})\Big)}{\alpha(Z)} \|f(\widetilde{v}_\eta) - f(Z)\|_\infty \leq \sum_{\eta \in \eta_1} \frac{\exp\Big(R(v_\eta^\top \widetilde{Z} - \frac{\|v_\eta\|_2^2}{2})\Big)}{\alpha(Z)} \cdot 2B_0$$

(By that $f$ is bounded)

$$\leq 2B_0 \frac{\sum_{\eta \in \eta_1} \exp\Big(R(v_\eta^\top \widetilde{Z} - \frac{\|v_\eta\|_2^2}{2})\Big)}{\alpha(Z)}$$

$$< 2B_0 \frac{\sum_{\eta \in \eta_1} \exp\Big(R(v_\eta^\top \widetilde{Z} - \frac{\|v_\eta\|_2^2}{2})\Big)}{\exp\Big(R(v_{\eta_m}^\top \widetilde{Z} - \frac{\|v_{\eta_m}\|_2^2}{2})\Big)}$$

$$\Big(\alpha(Z) \text{ is the sum of all } \exp\Big(R(v_\eta^\top \widetilde{Z} - \frac{\|v_\eta\|_2^2}{2})\Big), \text{ larger than any element in the sum}\Big)$$

$$= 2B_0 \sum_{\eta \in \eta_1} \exp\Big(\frac{R}{2}(\|v_{\eta_m} - Z\|_2^2 - \|v_\eta - Z\|_2^2)\Big)$$

$$\leq 2B_0 \|\eta_1\| \exp\left(\frac{R}{2}\left[(\frac{\delta}{2})^2 - \delta^2\right]\right)$$

$$< 2B_0 G \exp\left(\frac{-3R\delta^2}{8}\right)$$

$$= 2B_0 G \exp\left(\frac{-3\delta^2 \cdot \frac{8\ln\left(\frac{2}{3}B_0 G\epsilon\right)}{3\delta^2}}{8}\right)$$

$$= \frac{\epsilon}{3}. \tag{E.46}$$

Combing (E.44) and (E.45) yields

$$\sum_{\eta \in \eta_0 \cup \{\eta_m\}} \frac{\exp\left(R(v_\eta^\top \widetilde{Z} - \frac{\|v_\eta\|_2^2}{2})\right)}{\alpha(Z)} \|f(\widetilde{v}_\eta) - f(Z)\|_\infty$$

$$\leq \sum_{\eta \in \eta_0} \frac{\exp\left(R(v_\eta^\top \widetilde{Z} - \frac{\|v_\eta\|_2^2}{2})\right)}{\alpha(Z)} \cdot \frac{2\epsilon}{3} + \frac{\exp\left(R(v_{\eta_m}^\top \widetilde{Z} - \frac{\|v_{\eta_m}\|_2^2}{2})\right)}{\alpha(Z)} \cdot \frac{\epsilon}{3} \quad \left(\text{By (E.44) and (E.45)}\right)$$

$$\leq \sum_{\eta \in \eta_0 \cup \{\eta_m\}} \frac{\exp\left(R(v_\eta^\top \widetilde{Z} - \frac{\|v_\eta\|_2^2}{2})\right)}{\alpha(Z)} \cdot \frac{2\epsilon}{3}$$

$$\leq \frac{2\epsilon}{3}, \tag{E.47}$$

where the last line is by $\sum_{\eta \in E_0 \cup \{\eta_m\}} \frac{\exp\left(R(v_\eta^\top \widetilde{Z} - \frac{\|v_\eta\|_2^2}{2})\right)}{\alpha(Z)} \leq 1$.

By (E.47) and (E.46), we have

$$\|\text{Attn} \circ \text{Linear}(Z) - f(Z)\|_\infty \leq \frac{\exp\left(R(v_{\eta_m}^\top \widetilde{Z} - \frac{\|v_{\eta_m}\|_2^2}{2})\right)}{\alpha(Z)} \|f(\widetilde{v}_{\eta_m}) - f(Z)\|_\infty$$

$$+ \sum_{\eta \in E_0} \frac{\exp\left(R(v_\eta^\top \widetilde{Z} - \frac{\|v_\eta\|_2^2}{2})\right)}{\alpha(Z)} \|f(\widetilde{v}_\eta) - f(Z)\|_\infty$$

$$+ \sum_{\eta \in E_1} \frac{\exp\left(R(v_\eta^\top \widetilde{Z} - \frac{\|v_\eta\|_2^2}{2})\right)}{\alpha(Z)} \|f(\widetilde{v}_\eta) - f(Z)\|_\infty$$

$$\leq \frac{2\epsilon}{3} + \frac{\epsilon}{3}$$

$$= \epsilon.$$

This completes the proof. $\qquad\square$

Theorem 4.2 can be easily extended to Lebesgue integrable functions in $L_p$ norm in the following result.

**Corollary E.2.1** ($L_p$-Norm Universal Approximation). Let $f : U_K \times U_Q \to \mathbb{R}^{d \times n}$ denote any Lebesgue integrable function on a compact domain $U_K \times U_Q$ and let $\epsilon$ be any positive real number. Here $U_K, U_Q \in \mathbb{R}^{d \times n}$ stands for the compact domain of the two input sequences of cross-attention. Then, there exists a cross-attention $\text{Attn}$ prepended with a $\text{Linear}$ layer such that

$$\|f - \text{Attn} \circ \text{Linear}\|_{L_p} \leq \epsilon.$$

*Proof.* Without loss of generality, assume $U_K = U_Q = [-D, D]^{d \times n}$ for a $D \in R_+$.

Since $f$ is Lebesgue integrable on a compact set, $f$ is bounded almost every where. Let $B_p$ denote the bound of $\|f\|_p$.

By Lusin's theorem, for $f$ on a compact domain $U$, there exists a continuous function $g$ which is equal to $f$ in $U$ except for a region $D_\delta$ such that $\mu(D_\delta) \leq \Delta$. This can be written as

$$D_\delta = \{Z|f(Z) \neq g(Z)\}, \tag{E.48}$$
$$\mu(D_\delta) \leq \Delta, \tag{E.49}$$

where $\mu$ stands for the Lebesgue measure of a set.

By Theorem 4.2, there exists a network $\mathrm{Attn} \circ \mathrm{Linear}$, consists of a cross-attention $\mathrm{Attn}$ and a layer of sum of linear transformation $\mathrm{Linear}$ such that

$$\|\mathrm{Attn} \circ \mathrm{Linear} - g\|_{L_\infty} \leq \epsilon_0,$$

for any $\epsilon_0 > 0$.

This denote that for any $Z \in U \times U$

$$\|\mathrm{Attn} \circ \mathrm{Linear}(Z) - g(Z)\|_p \leq (dn \cdot \epsilon^p)^{\frac{1}{p}} = \epsilon_0 (dn)^{\frac{1}{p}}.$$

Combing this with (E.48) and (E.49), we get

$$\mu(\{Z | \|\mathrm{Attn} \circ \mathrm{Linear}(Z) - g(Z)\|_\infty > \epsilon_0\}) \leq \mu(\{f(Z) \neq g(Z)\}) \leq \Delta, \tag{E.50}$$

since if $f(Z) = g(Z)$, $\|\mathrm{Attn} \circ \mathrm{Linear}(Z) - g(Z)\| = \|\mathrm{Attn} \circ \mathrm{Linear}(Z) - f(Z)\| \leq \epsilon_0$

This yields

$$
\begin{aligned}
\|f - \mathrm{Attn} \circ \mathrm{Linear}\|_{L_p} &= \left(\int_{Z \in U \times U} \|f - \mathrm{Attn} \circ \mathrm{Linear}\|_p^p \, \mathrm{d}x\right)^{\frac{1}{p}} \\
&\leq \left(\int_{Z \in U \times U \setminus D_\delta} \|f - \mathrm{Attn} \circ \mathrm{Linear}\|_p^p \, \mathrm{d}x + \int_{Z \in D_\delta} \|f - \mathrm{Attn} \circ \mathrm{Linear}\|_p^p \, \mathrm{d}x\right)^{\frac{1}{p}} \\
&= \left(\int_{Z \in U \times U \setminus D_\delta} \|g - \mathrm{Attn} \circ \mathrm{Linear}\|_p^p \, \mathrm{d}x + \int_{Z \in D_\delta} \|f - \mathrm{Attn} \circ \mathrm{Linear}\|_p^p \, \mathrm{d}x\right)^{\frac{1}{p}} \\
&\leq \left(\mu(U \times U \setminus D_\delta)(\epsilon_0 (dn)^{\frac{1}{p}})^p + \Delta \cdot B_p^p\right)^{\frac{1}{p}} \\
&\leq \epsilon_0 (dn\mu(U \times U))^{\frac{1}{p}} + \Delta^{\frac{1}{p}} B_p.
\end{aligned}
$$

Set

$$\epsilon_0 \leq \frac{\epsilon}{2(dn\mu(U \times U))^{\frac{1}{p}}}$$

$$\Delta \leq \frac{\epsilon^p}{B_p \cdot 2^p}.$$

We have

$$
\begin{aligned}
\|f - \mathrm{Attn} \circ \mathrm{Linear}\|_{L_p} &\leq \epsilon_0 (dn\mu(U \times U))^{\frac{1}{p}} + \Delta^{\frac{1}{p}} B_p \\
&\leq (dn\mu(U \times U))^{\frac{1}{p}} \cdot \frac{\epsilon}{2(dn\mu(U \times U))^{\frac{1}{p}}} + \left(\frac{\epsilon^p}{B_p \cdot 2^p}\right)^{\frac{1}{p}} B_p \\
&= \frac{\epsilon}{2} + \frac{\epsilon}{2} \\
&= \epsilon.
\end{aligned}
$$

This completes the proof. $\qquad\square$

# F Proof of Results in Appendix A

## F.1 Proof of Theorem A.1

**Theorem F.1** (Theorem A.1 Restated). Let $f : \mathbb{R}^{d \times n} \to \mathbb{R}^{d \times n}$ denote an $L$-Lipschitz function (in terms of 2-norm) whose input domain is $\mathcal{X}$. For any $\epsilon > 0$, assume $\mathcal{X}$ is contained in $N_x$ sphere by the radius of $\epsilon/(3L)$ in 2-norm. Then, there exists a Linear layer and a Attn layer such that:

$$\|\text{Attn} \circ \text{Linear} - f\|_\infty \le \epsilon.$$

Furthermore, Attn and Linear have a total number of $\mathcal{O}(dnN_x)$ trainable parameters.

*Proof sketch.* This proof is identical with Theorem 4.1, except for an alteration on the set of $v_i$. ☐

*Proof.* We follow the proof of Theorem 4.1.

**Notation of Sphere Centers.** Let $Z = [z_1, z_2, \cdots, z_n] \in \mathbb{R}^{d \times n}$ denote the input to Linear. Define $\widetilde{Z} := [z_1^\top, z_2^\top, \cdots, z_n^\top]^\top$. $P \in N_+$ is a parameter that controls the size of the attention block and the error of our approximation.

Let $v_i, i \in [N_x]$ denote the centers of the $N_x$ spheres that covers $\mathcal{X}$. Let $V := \{v_i | i \in [N_x]\}$ denote the set of all $v_i$.

For every $v \in V$, we define $\widetilde{v} := [v_{1:d}^\top, v_{d+1:2d}^\top, \cdots, v_{(n-1)d+1:nd}^\top]^\top$.

**Construction of $f$ Related Functions.** Because $f$ is continuous within a closed region, its output value is bounded in $\infty$-norm. Let $B_0$ denote this bound, we now construct two functions that. For any $a \in \mathbb{R}^{d \times n}$, we define $E(a) := 1_{d \times n} - f(a)/B_0$ and $T(a) = 1_{d \times n} + f(a)/B_0$. We define $(E + T)(a) = E(a) + T(a)$. By the definition of $E$ and $T$, $(E + T)(a) \equiv 2_{d \times n}$ for any $a \in \mathbb{R}^{d \times n}$.

**Construction of the Layer of Sum of Linear Transformations.** We now construct the Linear layer to be

$$\text{Linear}(Z) := \sum_{j=0}^{N_x-1} \left( \sum_{k=0}^{(n-1)} (Ze_{k+1}^{(n)})^\top (v_j)_{kd+1:kd+d} \right) e_1^{(2dN_x+1)} \sum_{s=0}^{d-1} \left( e_{j+s+1}^{(2dN_x)} + e_{j+s+dN_x+1}^{(2dN_x)} \right)^\top + \begin{bmatrix} 0_{1 \times 2dN_x} \\ I_{2dN_x} \end{bmatrix},$$

where $N_x = P^{dn}$.

We now express the output of Linear in a simpler form in the following discussion. First, we show that

$$\sum_{k=0}^{(n-1)} (Ze_{k+1}^{(n)})^\top (v_j)_{kd+1:kd+d} = \sum_{k=0}^{(n-1)} z_{k+1}^\top (v_j)_{kd+1:kd+d}$$

$$= [z_1^\top, z_2^\top, \cdots, z_n^\top] v_j$$

$$= v_j^\top \widetilde{Z} \in \mathbb{R}, \ j \in \{0, 1, 2, \cdots, N_x - 1\}.$$

This yields

$$\text{Linear}(Z) = \sum_{j=0}^{N_x-1} v_j^\top \widetilde{Z} \sum_{s=0}^{d-1} \left( e_{j+s+1}^{(2dN_x)} + e_{j+s+dN_x+1}^{(2dN_x)} \right)^\top e_1^{(2dN_x+1)} + \begin{bmatrix} 0_{1 \times 2dN_x} \\ I_{2dN_x} \end{bmatrix}$$

$$= \begin{bmatrix} X_0 & X_0 \\ I_{dN_x} & 0_{dN_x \times dN_x} \\ 0_{dN_x \times dN_x} & I_{dN_x} \end{bmatrix},$$

in which $X_0$ is defined as follows

$$X_0 := \begin{bmatrix} v_0^\top \widetilde{Z} 1_{1 \times d} & v_1^\top \widetilde{Z} 1_{1 \times d} & v_2^\top \widetilde{Z} 1_{1 \times d} & \cdots & v_{N_x-1}^\top \widetilde{Z} 1_{1 \times d} \end{bmatrix}.$$

**Construction of $K$ and $Q$ Matrices.** We now construct the $W_k$ and $W_Q$ matrices in the self-attention block and calculate the output of $\mathrm{Softmax}\left(K^\top Q\right)$.

We define $W_K$ as follows:

$$W_K := \begin{bmatrix} 1 & 0_{1\times d} & \cdots & 0_{1\times d} & 0_{1\times d} & \cdots & 0_{1\times d} \\ 0 & -\frac{\|v_0\|_2^2}{2}1_{1\times d} & \cdots & -\frac{\|v_{N_x-1}\|_2^2}{2}1_{1\times d} & -\frac{\|v_0\|_2^2}{2}1_{1\times d} & \cdots & -\frac{\|v_{N_x-1}\|_2^2}{2}1_{1\times d} \\ 0 & \ln(T(\widetilde{v}_0))^\top & \cdots & \ln(T(\widetilde{v}_{N_x-1}))^\top & \ln(E(\widetilde{v}_0))^\top & \cdots & \ln(E(\widetilde{v}_{N_x-1}))^\top \end{bmatrix}.$$

The definition of $W_K$ yields

$$K := W_K \mathrm{Linear}(Z)$$
$$= \begin{bmatrix} 1 & 0_{1\times d} & \cdots & 0_{1\times d} & 0_{1\times d} & \cdots & 0_{1\times d} \\ 0 & -\frac{\|v_0\|_2^2}{2}1_{1\times d} & \cdots & -\frac{\|v_{N_x-1}\|_2^2}{2}1_{1\times d} & -\frac{\|v_0\|_2^2}{2}1_{1\times d} & \cdots & -\frac{\|v_{N_x-1}\|_2^2}{2}1_{1\times d} \\ 0 & \ln(T(\widetilde{v}_0))^\top & \cdots & \ln(T(\widetilde{v}_{N_x-1}))^\top & \ln(E(\widetilde{v}_0))^\top & \cdots & \ln(E(\widetilde{v}_{N_x-1}))^\top \end{bmatrix} \cdot \begin{bmatrix} X_0 & X_0 \\ I_{dN_x} & 0_{dN_x \times dN_x} \\ 0_{dN_x \times dN_x} & I_{dN_x} \end{bmatrix}$$
$$= \begin{bmatrix} v_0^\top \widetilde{Z}1_{1\times d} & \cdots & v_{N_x-1}^\top \widetilde{Z}1_{1\times d} & v_0^\top \widetilde{Z}1_{1\times d} & \cdots & v_{N_x-1}^\top \widetilde{Z}1_{1\times d} \\ -\frac{\|v_0\|_2^2}{2}1_{1\times d} & \cdots & -\frac{\|v_{N_x-1}\|_2^2}{2}1_{1\times d} & -\frac{\|v_0\|_2^2}{2}1_{1\times d} & \cdots & -\frac{\|v_{N_x-1}\|_2^2}{2}1_{1\times d} \\ \ln(T(\widetilde{v}_0))^\top & \cdots & \ln(T(\widetilde{v}_{N_x-1}))^\top & \ln(E(\widetilde{v}_0))^\top & \cdots & \ln(E(\widetilde{v}_{N_x-1}))^\top \end{bmatrix}.$$

Next, we construct $W_Q$ to be

$$W_Q := \begin{bmatrix} 0 & R1_{1\times n} & 0_{1\times(2dN_x-n)} \\ 0 & R1_{1\times n} & 0_{1\times(2dN_x-n)} \\ 0_n & I_n & 0_{n\times(2dN_x-n)} \end{bmatrix}.$$

This yields that

$$Q = W_Q \mathrm{Linear}(Z)$$
$$= \begin{bmatrix} 0 & R1_{1\times n} & 0_{1\times(2dN_x-n)} \\ 0 & R1_{1\times n} & 0_{1\times(2dN_x-n)} \\ 0_n & I_n & 0_{n\times(2dN_x-n)} \end{bmatrix} \cdot \begin{bmatrix} X_0 & X_0 \\ I_{dN_x} & 0_{dN_x \times dN_x} \\ 0_{dN_x \times dN_x} & I_{dN_x} \end{bmatrix}$$
$$= \begin{bmatrix} R1_{1\times n} & 0_{1\times(2dN_x-n)} \\ R1_{1\times n} & 0_{1\times(2dN_x-n)} \\ I_n & 0_{n\times(2dN_x-n)} \end{bmatrix}.$$

We now calculate the attention matrix $\mathrm{Softmax}\left(K^\top Q\right)$.

**Calculation of $\mathrm{Softmax}(K^\top Q)$.** First, $K^\top Q$ can be expressed as follows

$$K^\top Q = \begin{bmatrix} v_0^\top \widetilde{Z}1_d & \frac{\|v_0\|_2^2}{2}1_d & \ln(T(\widetilde{v}_0)) \\ v_1^\top \widetilde{Z}1_d & \frac{\|v_1\|_2^2}{2}1_d & \ln(T(\widetilde{v}_1)) \\ & \vdots & \\ v_{N_x-1}^\top \widetilde{Z}1_d & \frac{\|v_1\|_2^2}{2}1_d & \ln(T(\widetilde{v}_{N_x-1})) \\ v_0^\top \widetilde{Z}1_d & \frac{\|v_0\|_2^2}{2}1_d & \ln(E(\widetilde{v}_0)) \\ v_1^\top \widetilde{Z}1_d & \frac{\|v_1\|_2^2}{2}1_d & \ln(E(\widetilde{v}_1)) \\ & \vdots & \\ v_{N_x-1}^\top \widetilde{Z}1_d & \frac{\|v_1\|_2^2}{2}1_d & \ln(E(\widetilde{v}_{N_x-1})) \end{bmatrix} \cdot \begin{bmatrix} R1_{1\times n} & 0_{1\times(2dN_x-n)} \\ R1_{1\times n} & 0_{1\times(2dN_x-n)} \\ I_n & 0_{n\times(2dN_x-n)} \end{bmatrix}$$

$$
= \begin{bmatrix}
R(v_0^\top \widetilde{Z} - \frac{\|v_0\|_2^2}{2})1_{d\times n} + \ln(T(\widetilde{v}_0)) & 0_{d\times(2dN_x-n)} \\
R(v_1^\top \widetilde{Z} - \frac{\|v_1\|_2^2}{2})1_{d\times n} + \ln(T(\widetilde{v}_1)) & 0_{d\times(2dN_x-n)} \\
\vdots & \vdots \\
R(v_{N_x-1}^\top \widetilde{Z} - \frac{\|v_{N_x-1}\|_2^2}{2})1_{d\times n} + \ln(T(\widetilde{v}_{N_x-1})) & 0_{d\times(2dN_x-n)} \\
R(v_0^\top \widetilde{Z} - \frac{\|v_0\|_2^2}{2})1_{d\times n} + \ln(E(\widetilde{v}_0)) & 0_{d\times(2dN_x-n)} \\
R(v_1^\top \widetilde{Z} - \frac{\|v_1\|_2^2}{2})1_{d\times n} + \ln(E(\widetilde{v}_1)) & 0_{d\times(2dN_x-n)} \\
\vdots & \vdots \\
R(v_{N_x-1}^\top \widetilde{Z} - \frac{\|v_{N_x-1}\|_2^2}{2})1_{d\times n} + \ln(E(\widetilde{v}_{N_x-1})) & 0_{d\times(2dN_x-n)}
\end{bmatrix}.
$$

Now, we divide the calculation of $\mathrm{Softmax}\left(K^\top Q\right)$ into two counterparts, the calculation of $\exp\left(K^\top Q\right)$ and the calculation of the denominator of every column of $\mathrm{Softmax}\left(K^\top Q\right)$, as in the expression of Softmax, explicitly written out as $\sum_{j=1}^{2dN_x} \exp\left(K^\top Q\right)_{ij}$ for each $i \in [2dN_x]$.

For $\exp\left(K^\top Q\right)$, we have

$$
\exp(K^\top Q) = \begin{bmatrix}
\exp\left(R(v_0^\top \widetilde{Z} - \frac{\|v_0\|_2^2}{2})\right)T(\widetilde{v}_0) & 1_{d\times(2dN_x-n)} \\
\exp\left(R(v_1^\top \widetilde{Z} - \frac{\|v_1\|_2^2}{2})\right)T(\widetilde{v}_1) & 1_{d\times(2dN_x-n)} \\
\vdots & \\
\exp\left(R(v_{N_x-1}^\top \widetilde{Z} - \frac{\|v_{N_x-1}\|_2^2}{2})\right)T(\widetilde{v}_{N_x-1}) & 1_{d\times(2dN_x-n)} \\
\exp\left(R(v_0^\top \widetilde{Z} - \frac{\|v_0\|_2^2}{2})\right)E(\widetilde{v}_0) & 1_{d\times(2dN_x-n)} \\
\exp\left(R(v_1^\top \widetilde{Z} - \frac{\|v_1\|_2^2}{2})\right)E(\widetilde{v}_1) & 1_{d\times(2dN_x-n)} \\
\vdots & \\
\exp\left(R(v_{N_x-1}^\top \widetilde{Z} - \frac{\|v_{N_x-1}\|_2^2}{2})\right)E(\widetilde{v}_{N_x-1}) & 1_{d\times(2dN_x-n)}
\end{bmatrix} \tag{F.1}
$$

$$
= \begin{bmatrix}
\exp\left(R(v_0^\top \widetilde{Z} - \frac{\|v_0\|_2^2}{2})\right)T(\widetilde{v}_0) & 1_{d\times(2dN_x-n)} \\
\exp\left(R(v_1^\top \widetilde{Z} - \frac{\|v_1\|_2^2}{2})\right)T(\widetilde{v}_1) & 1_{d\times(2dN_x-n)} \\
\vdots & \\
\exp\left(R(v_{N_x-1}^\top \widetilde{Z} - \frac{\|v_{N_x-1}\|_2^2}{2})\right)T(\widetilde{v}_{N_x-1}) & 1_{d\times(2dN_x-n)} \\
\exp\left(R(v_0^\top \widetilde{Z} - \frac{\|v_0\|_2^2}{2})\right)E(\widetilde{v}_0) & 1_{d\times(2dN_x-n)} \\
\exp\left(R(v_1^\top \widetilde{Z} - \frac{\|v_1\|_2^2}{2})\right)E(\widetilde{v}_1) & 1_{d\times(2dN_x-n)} \\
\vdots & \\
\exp\left(R(v_{N_x-1}^\top \widetilde{Z} - \frac{\|v_{N_x-1}\|_2^2}{2})\right)E(\widetilde{v}_{N_x-1}) & 1_{d\times(2dN_x-n)}
\end{bmatrix}. \tag{F.2}
$$

For the denominator, we calculate it in columns. Let $i$ denote the column which we calculate the denominator in Softmax. When $i \in \{n+1, n+2, \cdots, 2dN_x\}$, it obviously equals to $1 \cdot 2dN_x = 2dN_x$. And when $i \in [n]$, we denote that

$$
\sum_{j=1}^{2dN_x} \exp\left(K^\top Q\right)_{ij} = \sum_{j=1}^{N_x}\left[\left(1_{1\times d}T(\widetilde{v}_{j-1})_{:,i} + 1_{1\times d}E(\widetilde{v}_{j-1})_{:,i}\right) \cdot \exp\left(R\left(v_{j-1}^\top \widetilde{Z} - \frac{\|v_{j-1}\|_2^2}{2}\right)\right)\right]
$$

$$
= \sum_{j=1}^{N_x}\left[\left(1_{1\times d}(E+T)(v_{j-1})_{:,i}\right) \cdot \exp\left(R\left(v_{j-1}^\top \widetilde{Z} - \frac{\|v_{j-1}\|_2^2}{2}\right)\right)\right]
$$

$$
= \sum_{j=1}^{N_x}\left[\left(1_{1\times d}(2_{d\times n})_{:,i}\right) \cdot \exp\left(R\left(v_{j-1}^\top \widetilde{Z} - \frac{\|v_{j-1}\|_2^2}{2}\right)\right)\right]
$$

$$= \sum_{j=1}^{N_x} 2d \cdot \exp\left( R\left( v_{j-1}^\top \widetilde{Z} - \frac{\|v_{j-1}\|_2^2}{2} \right) \right), \quad i \in [n]. \tag{F.3}$$

We observe from (F.3), that $\sum_{j=1}^{2dN_x} \exp\left(K^\top Q\right)_{ij}$ is invariant of $i$ for $i \in [n]$. In this case, we define

$$\alpha(Z) := \frac{1}{2d} \sum_{j=1}^{2dN_x} \exp\left(K^\top Q\right)_{ij} = \sum_{j=1}^{N_x} \exp\left( R\left( v_{j-1}^\top \widetilde{Z} - \frac{\|v_{j-1}\|_2^2}{2} \right) \right) \in \mathbb{R}, \quad i \in [n].$$

From (F.1) and (F.3) we have

$\text{Softmax}\left(K^\top Q\right)$

$= \exp\left(K^\top Q\right) \odot \left[ \frac{1}{\sum_{j=1}^{2dN_x} \exp(K^\top Q)_{1j}} 1_{2dN_x \times n} \quad \frac{1}{2dN_x} 1_{2dN_x \times (2dN_x - n)} \right]$

$$\left( \text{By } 1/\sum_{j=1}^{2dN_x} \exp\left(K^\top Q\right)_{i,j} \text{ is invariant of } i \text{ for } i \in [n] \right)$$

$$= \begin{bmatrix} \exp\left(R(v_0^\top \widetilde{Z} - \frac{\|v_0\|_2^2}{2})\right) T(\widetilde{v}_0) & 1_{d \times (2dN_x - n)} \\ \exp\left(R(v_1^\top \widetilde{Z} - \frac{\|v_1\|_2^2}{2})\right) T(\widetilde{v}_1) & 1_{d \times (2dN_x - n)} \\ \cdots & \\ \exp\left(R(v_{N_x-1}^\top \widetilde{Z} - \frac{\|v_{N_x-1}\|_2^2}{2})\right) T(\widetilde{v}_{N_x-1}) & 1_{d \times (2dN_x - n)} \\ \exp\left(R(v_0^\top \widetilde{Z} - \frac{\|v_0\|_2^2}{2})\right) E(\widetilde{v}_0) & 1_{d \times (2dN_x - n)} \\ \exp\left(R(v_1^\top \widetilde{Z} - \frac{\|v_1\|_2^2}{2})\right) E(\widetilde{v}_1) & 1_{d \times (2dN_x - n)} \\ \cdots & \\ \exp\left(R(v_{N_x-1}^\top \widetilde{Z} - \frac{\|v_{N_x-1}\|_2^2}{2})\right) E(\widetilde{v}_{N_x-1}) & 1_{d \times (2dN_x - n)} \end{bmatrix} \odot \left[ \frac{1}{2d\alpha(Z)} 1_{2dN_x \times n} \quad \frac{1}{2dN_x} 1_{2dN_x \times (2dN_x - n)} \right]$$

$$= \frac{1}{2d} \begin{bmatrix} \frac{\exp\left(R(v_0^\top \widetilde{Z} - \frac{\|v_0\|_2^2}{2})\right)}{\alpha(Z)} T(\widetilde{v}_0) & \frac{1}{N_x} 1_{d \times (2dN_x - n)} \\ \frac{\exp\left(R(v_1^\top \widetilde{Z} - \frac{\|v_1\|_2^2}{2})\right)}{\alpha(Z)} T(\widetilde{v}_1) & \frac{1}{N_x} 1_{d \times (2dN_x - n)} \\ \cdots & \\ \frac{\exp\left(R(v_{N_x-1}^\top \widetilde{Z} - \frac{\|v_{N_x-1}\|_2^2}{2})\right)}{\alpha(Z)} T(\widetilde{v}_{N_x-1}) & \frac{1}{N_x} 1_{d \times (2dN_x - n)} \\ \frac{\exp\left(R(v_0^\top \widetilde{Z} - \frac{\|v_0\|_2^2}{2})\right)}{\alpha(Z)} E(\widetilde{v}_0) & \frac{1}{N_x} 1_{d \times (2dN_x - n)} \\ \frac{\exp\left(R(v_1^\top \widetilde{Z} - \frac{\|v_1\|_2^2}{2})\right)}{\alpha(Z)} E(\widetilde{v}_1) & \frac{1}{N_x} 1_{d \times (2dN_x - n)} \\ \cdots & \\ \frac{\exp\left(R(v_{N_x-1}^\top \widetilde{Z} - \frac{\|v_{N_x-1}\|_2^2}{2})\right)}{\alpha(Z)} E(\widetilde{v}_{N_x-1}) & \frac{1}{N_x} 1_{d \times (2dN_x - n)} \end{bmatrix}.$$

**Construction of $W_V$ and $W_O$.** We now construct the $W_V$ matrix and calculate the $V$ matrix of the self-attention.

We define $W_V$ as

$$W_V := [0_d \quad X_1 \quad -X_1],$$

where

$$X_1 := [I_d \quad I_d \quad \cdots \quad I_d]_{d \times dN_x},$$

is a matrix formed by stacking $N_x$ $I_d$ matrices horizontally.

In this definition, $V$ matrix can be calculated as follows:

$$V := W_V \text{Linear}(Z)$$

$$= \begin{bmatrix} 0_d & X_1 & -X_1 \end{bmatrix} \begin{bmatrix} X_0 & X_0 \\ I_{dN_x} & 0_{dN_x \times dN_x} \\ 0_{dN_x \times dN_x} & I_{dN_x} \end{bmatrix}$$

$$= \begin{bmatrix} X_1 & -X_1 \end{bmatrix}.$$

After the construction and calculation of $V$, we go on to construct $W_O$ as

$$W_O = \begin{bmatrix} dB_0 I_n \\ 0_{(2dN_x - n) \times n} \end{bmatrix}.$$

The sole purpose of $W_O$ is to extract the non-zero entries of the final output.

**Calculation of the Output of** $\mathrm{Attn} \circ \mathrm{Linear}$**.** We now calculate the final output of the self-attention block.

$$\mathrm{Attn} \circ \mathrm{Linear}(Z) = \frac{1}{2d} \begin{bmatrix} X_1 & -X_1 \end{bmatrix} \begin{bmatrix} \frac{\exp\left(R(v_0^\top \widetilde{Z} - \frac{\|v_0\|_2^2}{2})\right)}{\alpha(Z)} T(\widetilde{v}_0) & \frac{1}{N_x} 1_{d \times (2dN_x - n)} \\ \frac{\exp\left(R(v_1^\top \widetilde{Z} - \frac{\|v_1\|_2^2}{2})\right)}{\alpha(Z)} T(\widetilde{v}_1) & \frac{1}{N_x} 1_{d \times (2dN_x - n)} \\ \cdots & \\ \frac{\exp\left(R(v_{N_x-1}^\top \widetilde{Z} - \frac{\|v_{N_x-1}\|_2^2}{2})\right)}{\alpha(Z)} T(\widetilde{v}_{N_x-1}) & \frac{1}{N_x} 1_{d \times (2dN_x - n)} \\ \frac{\exp\left(R(v_0^\top \widetilde{Z} - \frac{\|v_0\|_2^2}{2})\right)}{\alpha(Z)} E(\widetilde{v}_0) & \frac{1}{N_x} 1_{d \times (2dN_x - n)} \\ \frac{\exp\left(R(v_1^\top \widetilde{Z} - \frac{\|v_1\|_2^2}{2})\right)}{\alpha(Z)} E(\widetilde{v}_1) & \frac{1}{N_x} 1_{d \times (2dN_x - n)} \\ \cdots & \\ \frac{\exp\left(R(v_{N_x-1}^\top \widetilde{Z} - \frac{\|v_{N_x-1}\|_2^2}{2})\right)}{\alpha(Z)} E(\widetilde{v}_{N_x-1}) & \frac{1}{N_x} 1_{d \times (2dN_x - n)} \end{bmatrix} W_O$$

$$= \frac{1}{2d} X_1 \begin{bmatrix} \frac{\exp\left(R(v_0^\top \widetilde{Z} - \frac{\|v_0\|_2^2}{2})\right)}{\alpha(Z)} (T(\widetilde{v}_0) - E(\widetilde{v}_0)) & 0_{d \times (2dN_x - n)} \\ \frac{\exp\left(R(v_1^\top \widetilde{Z} - \frac{\|v_1\|_2^2}{2})\right)}{\alpha(Z)} (T(\widetilde{v}_1) - E(\widetilde{v}_1)) & 0_{d \times (2dN_x - n)} \\ \cdots & \\ \frac{\exp\left(R(v_{N_x-1}^\top \widetilde{Z} - \frac{\|v_{N_x-1}\|_2^2}{2})\right)}{\alpha(Z)} (T(\widetilde{v}_{N_x-1}) - E(\widetilde{v}_{N_x-1})) & 0_{d \times (2dN_x - n)} \end{bmatrix} W_O$$

$$= \frac{1}{2d} X_1 \begin{bmatrix} \frac{\exp\left(R(v_0^\top \widetilde{Z} - \frac{\|v_0\|_2^2}{2})\right)}{\alpha(Z)} \frac{2f(\widetilde{v}_0)}{B_0} & 0_{d \times (2dN_x - n)} \\ \frac{\exp\left(R(v_1^\top \widetilde{Z} - \frac{\|v_1\|_2^2}{2})\right)}{\alpha(Z)} \frac{2f(\widetilde{v}_1)}{B_0} & 0_{d \times (2dN_x - n)} \\ \cdots & \\ \frac{\exp\left(R(v_{N_x-1}^\top \widetilde{Z} - \frac{\|v_{N_x-1}\|_2^2}{2})\right)}{\alpha(Z)} \frac{2f(\widetilde{v}_{N_x-1})}{B_0} & 0_{d \times (2dN_x - n)} \end{bmatrix} W_O.$$

We have

$$X_1 \begin{bmatrix} \frac{\exp\left(R(v_0^\top \widetilde{Z} - \frac{\|v_0\|_2^2}{2})\right)}{\alpha(Z)} \frac{2f(\widetilde{v}_0)}{B_0} \\ \frac{\exp\left(R(v_1^\top \widetilde{Z} - \frac{\|v_1\|_2^2}{2})\right)}{\alpha(Z)} \frac{2f(\widetilde{v}_1)}{B_0} \\ \cdots \\ \frac{\exp\left(R(v_{N_x-1}^\top \widetilde{Z} - \frac{\|v_{N_x-1}\|_2^2}{2})\right)}{\alpha(Z)} \frac{2f(\widetilde{v}_{N_x-1})}{B_0} \end{bmatrix} = \begin{bmatrix} I_d & I_d & \cdots & I_d \end{bmatrix}_{d \times dN_x} \cdot \begin{bmatrix} \frac{\exp\left(R(v_0^\top \widetilde{Z} - \frac{\|v_0\|_2^2}{2})\right)}{\alpha(Z)} \frac{2f(\widetilde{v}_0)}{B_0} \\ \frac{\exp\left(R(v_1^\top \widetilde{Z} - \frac{\|v_1\|_2^2}{2})\right)}{\alpha(Z)} \frac{2f(\widetilde{v}_1)}{B_0} \\ \cdots \\ \frac{\exp\left(R(v_{N_x-1}^\top \widetilde{Z} - \frac{\|v_{N_x-1}\|_2^2}{2})\right)}{\alpha(Z)} \frac{2f(\widetilde{v}_{N_x-1})}{B_0} \end{bmatrix}$$

$$= \sum_{j=0}^{N_x-1} I_d \cdot \frac{\exp\left(R(v_j^\top \widetilde{Z} - \frac{\|v_{j-1}\|_2^2}{2})\right)}{\alpha(Z)} \frac{2f(\widetilde{v}_j)}{B_0}$$

$$= \sum_{j=0}^{N_x-1} \frac{\exp\left(R(v_j^\top \widetilde{Z} - \frac{\|v_j\|_2^2}{2})\right)}{\alpha(Z)} \frac{2f(\widetilde{v}_j)}{B_0}.$$

This yields

$$\text{Attn} \circ \text{Linear}(Z) = \left[\sum_{j=0}^{N_x-1} \frac{\exp\left(R(v_j^\top \widetilde{Z} - \frac{\|v_j\|_2^2}{2})\right)}{\alpha(Z)} \frac{2f(\widetilde{v}_j)}{B_0} \quad 0_{d\times(2dN_x-n)}\right] W_O$$

$$= \left[\sum_{j=0}^{N_x-1} \frac{\exp\left(R(v_j^\top \widetilde{Z} - \frac{\|v_j\|_2^2}{2})\right)}{\alpha(Z)} \frac{2f(\widetilde{v}_j)}{B_0} \quad 0_{d\times(2dN_x-n)}\right] \begin{bmatrix} dB_0 I_n \\ 0_{(2dN_x-n)\times n} \end{bmatrix}$$

$$= \sum_{j=0}^{N_x-1} \frac{\exp\left(R(v_j^\top \widetilde{Z} - \frac{\|v_j\|_2^2}{2})\right)}{\alpha(Z)} f(\widetilde{v}_j). \tag{F.4}$$

**Estimation of the Error between** $\text{Attn} \circ \text{Linear}(Z)$ **and** $f(Z)$**.** After the above calculations of the output of the network, we can now demonstrate how this output approximates our target function.

---

**Definition F.1** (Max-Affine Function on $\widetilde{Z}$). Let $\text{Aff}_j \in \mathbb{R}^{dn} \to \mathbb{R}$, $j \in \{0, 1, 2, \cdots, N_x - 1\}$ denote a group of affine functions defined as

$$\text{Aff}_j(\widetilde{Z}) = v_j^\top \widetilde{Z} - \frac{\|v_j\|_2^2}{2}, \ j \in \{0, 1, 2, \cdots, N_x - 1\}.$$

Then let $\text{MaxAff} \in \mathbb{R}^{dn} \to \mathbb{R}$ denote a max affine function whose affine components are $\{\text{Aff}_j \mid j \in \{0, 1, 2, \cdots, N_x - 1\}\}$. Explicitly defined as

$$\text{MaxAff}(\widetilde{Z}) = \max_{j\in\{0,1,2,\cdots,N_x-1\}} \{\text{Aff}_j(\widetilde{Z})\}.$$

---

Because the target function $f$ is a continuous function on a closed domain, the function $f$ is uniformly continuous. Thus for $\epsilon$, there exists a $\delta > 0$ such that for any $Z_1, Z_2$, as long as $\|\widetilde{Z}_1 - \widetilde{Z}_2\|_\infty \le \delta$, we have $\|f(Z_1) - f(Z_2)\|_\infty \le \epsilon/3$.

According to this $\delta$, we divide the affine components of $\text{MaxAff}$ into three parts, the maximal component(and also with the smallest label), whose label is denoted as $j_m$, the group of affine components equal to the maximal component or smaller than it by no more than $\delta$, and finally, the other $\text{Aff}_j$, $j \in \{0, 1, 2, \cdots, N_x - 1\}$. We write out the labels of these groups of components as follows

$$j_m := \min_{j\in\{0,1,2,\cdots,N_x-1\}} \{\text{Aff}_j(\widetilde{Z}) = \text{MaxAff}(\widetilde{Z})\},$$

$$J_0 := \{j \mid \text{MaxAff}(\widetilde{Z}) - \text{Aff}_j(\widetilde{Z}) \le \delta\},$$

$$J_1 := \{j \mid \text{MaxAff}(\widetilde{Z}) - \text{Aff}_j(\widetilde{Z}) > \delta\}.$$

For any pair of $i, j \in \{0, 1, \cdots, N_x - 1\}$, we have

$$\text{Aff}_i(\widetilde{Z}) - \text{Aff}_j(\widetilde{Z}) = v_i^\top \widetilde{Z} - \frac{\|v_i\|_2^2}{2} - \left(v_j^\top \widetilde{Z} - \frac{\|v_j\|_2^2}{2}\right)$$

$$= -\frac{\|\widetilde{Z}\|_2^2}{2} + v_i^\top \widetilde{Z} - \frac{\|v_i\|_2^2}{2} - \left(-\frac{\|\widetilde{Z}\|_2^2}{2} + v_j^\top \widetilde{Z} - \frac{\|v_j\|_2^2}{2}\right)$$

$$= -\frac{1}{2}\|\widetilde{Z} - v_i\|_2^2 + \frac{1}{2}\|\widetilde{Z} - v_j\|_2^2.$$

This denotes $j_m$ is also the label of the closest $v_i$ to $\widetilde{Z}$ among all $v_i$, $i \in \{0, 1, \cdots, N_x - 1\}$. Thus we have

$$\|v_{j_m} - \widetilde{Z}\|_2 = \min_{i\in\{0,1,\cdots,N_x-1\}} \{\|v_i - \widetilde{Z}\|_2\}. \tag{F.5}$$

Thus, when considering the $Z$ in the input domain of $f$, which by definition is contained in $N_x$ spheres, the closest center to $Z$ is the sphere containing $Z$. This gives

$$\|Z - \widetilde{v}_{j_m}\|_2 \le \frac{\epsilon}{3L}. \tag{F.6}$$

Then, with the $L$ Lipschitzness of $L$ we have

$$\|f(Z) - f(\widetilde{v}_{j_m})\|_\infty \le \frac{\epsilon}{3L} \cdot L = \frac{\epsilon}{3}. \tag{F.7}$$

**Difference between** $\mathrm{Attn} \circ \mathrm{Linear}$ **and** $f$. We now calculate the difference between the output in (F.4) and target function $f$

$$\|\mathrm{Attn} \circ \mathrm{Linear}(Z) - f(Z)\|_\infty$$

$$= \| \sum_{j=0}^{N_x-1} \frac{\exp\left(R(v_j^\top \widetilde{Z} - \frac{\|v_j\|_2^2}{2})\right)}{\alpha(Z)} f(\widetilde{v}_j) - f(Z)\|_\infty$$

$$= \| \sum_{j=0}^{N_x-1} \frac{\exp\left(R(v_j^\top \widetilde{Z} - \frac{\|v_j\|_2^2}{2})\right)}{\alpha(Z)} (f(\widetilde{v}_j) - f(Z))\| \qquad \left(\sum_{j=0}^{N_x-1} \frac{\exp\left(R(v_j^\top \widetilde{Z} - \frac{\|v_j\|_2^2}{2})\right)}{\alpha(Z)} = 1\right)$$

$$\le \sum_{j=0}^{N_x-1} \frac{\exp\left(R(v_j^\top \widetilde{Z} - \frac{\|v_j\|_2^2}{2})\right)}{\alpha(Z)} \|f(\widetilde{v}_j) - f(Z)\|_\infty \qquad \left(\text{property of infinite norm}\right)$$

$$= \frac{\exp\left(R(v_{j_m}^\top \widetilde{Z} - \frac{\|v_{j_m}\|_2^2}{2})\right)}{\alpha(Z)} \|f(\widetilde{v}_{j_m}) - f(Z)\|_\infty$$

$$+ \sum_{j \in J_0} \frac{\exp\left(R(v_j^\top \widetilde{Z} - \frac{\|v_j\|_2^2}{2})\right)}{\alpha(Z)} \|f(\widetilde{v}_j) - f(Z)\|_\infty$$

$$+ \sum_{j \in J_1} \frac{\exp\left(R(v_j^\top \widetilde{Z} - \frac{\|v_j\|_2^2}{2})\right)}{\alpha(Z)} \|f(\widetilde{v}_j) - f(Z)\|_\infty. \tag{F.8}$$

We now calculate each part in (F.8).

For the $L$-Lipschitzness of $f$, for any $Z_1, Z_2$, as long as $\|\widetilde{Z}_1 - \widetilde{Z}_2\|_\infty \le \frac{\epsilon}{3L}$, we have $\|f(Z_1) - f(Z_2)\|_\infty \le \epsilon/3$. Thus when we designate $Z_1 = v_j$ for any $j \in J_0$ and $Z_2 = v_{j_m}$, along with (F.7) we have:

$$\sum_{j \in J_0} \frac{\exp\left(R(v_j^\top \widetilde{Z} - \frac{\|v_j\|_2^2}{2})\right)}{\alpha(Z)} \|f(\widetilde{v}_j) - f(Z)\|_\infty \tag{F.9}$$

$$\le \sum_{j \in J_0} \frac{\exp\left(R(v_j^\top \widetilde{Z} - \frac{\|v_j\|_2^2}{2})\right)}{\alpha(Z)} (\|f(\widetilde{v}_j) - f(\widetilde{v}_{j_m})\|_\infty + \|f(\widetilde{v}_{j_m}) - f(Z)\|_\infty)$$

$$\le \sum_{j \in J_0} \frac{\exp\left(R(v_j^\top \widetilde{Z} - \frac{\|v_j\|_2^2}{2})\right)}{\alpha(Z)} \cdot (\frac{\epsilon}{3} + \frac{\epsilon}{3})$$

$$= \sum_{j \in J_0} \frac{\exp\left(R(v_j^\top \widetilde{Z} - \frac{\|v_j\|_2^2}{2})\right)}{\alpha(Z)} \cdot \frac{2\epsilon}{3}. \tag{F.10}$$

For $j_m$, we have

$$\frac{\exp\left(R(v_{j_m}^\top \widetilde{Z} - \frac{\|v_{j_m}\|_2^2}{2})\right)}{\alpha(Z)} \|f(\widetilde{v}_{j_m}) - f(Z)\|_\infty \le \frac{\exp\left(R(v_{j_m}^\top \widetilde{Z} - \frac{\|v_{j_m}\|_2^2}{2})\right)}{\alpha(Z)} \cdot \frac{\epsilon}{3}. \tag{F.11}$$

When $R$ is larger than $\frac{8}{3\delta^2}\ln\left(\frac{3}{2}\cdot B_0 N_x \epsilon\right)$, we have:

$$\sum_{j\in J_1}\frac{\exp\left(R(v_j^\top \widetilde{Z} - \frac{\|v_j\|_2^2}{2})\right)}{\alpha(Z)}\|f(\widetilde{v}_j) - f(Z)\|_\infty$$

$$\leq \sum_{j\in J_1}\frac{\exp\left(R(v_j^\top \widetilde{Z} - \frac{\|v_j\|_2^2}{2})\right)}{\alpha(Z)}\cdot 2B_0 \qquad\qquad \text{(by the bounded nature of } f)$$

$$\leq 2B_0\frac{\sum_{j\in J_1}\exp\left(R(v_j^\top \widetilde{Z} - \frac{\|v_j\|_2^2}{2})\right)}{\alpha(Z)}$$

$$< 2B_0\frac{\sum_{j\in J_1}\exp\left(R(v_j^\top \widetilde{Z} - \frac{\|v_j\|_2^2}{2})\right)}{\exp\left(R(v_{j_m}^\top \widetilde{Z} - \frac{\|v_{j_m}\|_2^2}{2})\right)}$$

$$\left(\alpha(Z) \text{ is the sum of all } \exp\left(R(v_j^\top \widetilde{Z} - \frac{\|v_j\|_2^2}{2})\right), \text{ thus larger than any element within the summation}\right)$$

$$= 2B_0\sum_{j\in J_1}\exp\left(\frac{R}{2}(\|v_{j_m} - Z\|_2^2 - \|v_j - Z\|_2^2)\right)$$

$$\leq 2B_0\|J_1\|\exp\left(\frac{R}{2}\left[(\frac{\delta}{2})^2 - \delta^2\right]\right)$$

$$< 2B_0 N_x \exp\left(\frac{-3R\delta^2}{8}\right)$$

$$= 2B_0 N_x \exp\left(\frac{-3\delta^2\cdot\frac{8\ln\left(\frac{2}{3}B_0 N_x \epsilon\right)}{3\delta^2}}{8}\right)$$

$$= \frac{\epsilon}{3}. \tag{F.12}$$

Combing (F.10) and (F.11) yields

$$\sum_{j\in J_0\cup\{j_m\}}\frac{\exp\left(R(v_j^\top \widetilde{Z} - \frac{\|v_j\|_2^2}{2})\right)}{\alpha(Z)}\|f(\widetilde{v}_j) - f(Z)\|_\infty$$

$$\leq \sum_{j\in J_0}\frac{\exp\left(R(v_j^\top \widetilde{Z} - \frac{\|v_j\|_2^2}{2})\right)}{\alpha(Z)}\cdot\frac{2\epsilon}{3} + \frac{\exp\left(R(v_{j_m}^\top \widetilde{Z} - \frac{\|v_{j_m}\|_2^2}{2})\right)}{\alpha(Z)}\cdot\frac{\epsilon}{3} \qquad \text{(By (F.10) and (F.11))}$$

$$\leq \sum_{j\in J_0\cup\{j_m\}}\frac{\exp\left(R(v_j^\top \widetilde{Z} - \frac{\|v_j\|_2^2}{2})\right)}{\alpha(Z)}\cdot\frac{2\epsilon}{3}$$

$$\leq \frac{2\epsilon}{3}, \tag{F.13}$$

where the last line is by $\sum_{j\in J_0\cup\{j_m\}}\frac{\exp\left(R(v_j^\top \widetilde{Z} - \frac{\|v_j\|_2^2}{2})\right)}{\alpha(Z)}\leq 1$.

We plug (F.12) and (F.13) to (F.8) and get

$$\|\text{Attn}\circ\text{Linear}(Z) - f(Z)\|_\infty \leq \frac{\exp\left(R(v_{j_m}^\top \widetilde{Z} - \frac{\|v_{j_m}\|_2^2}{2})\right)}{\alpha(Z)}\|f(\widetilde{v}_{j_m}) - f(Z)\|_\infty$$

$$+ \sum_{j\in J_0}\frac{\exp\left(R(v_j^\top \widetilde{Z} - \frac{\|v_j\|_2^2}{2})\right)}{\alpha(Z)}\|f(\widetilde{v}_j) - f(Z)\|_\infty$$

$$+ \sum_{j \in J_1} \frac{\exp\left(R(v_j^\top \widetilde{Z} - \frac{\|v_j\|_2^2}{2})\right)}{\alpha(Z)} \|f(\widetilde{v}_j) - f(Z)\|_\infty$$

$$\leq \frac{2\epsilon}{3} + \frac{\epsilon}{3}$$

$$= \epsilon.$$

This concludes our result on the approximation error.

**Estimation of the Number of Trainable Parameters.** We now estimate the number of trainable parameter in the network we constructed to verify our claim on number of trainable parameters in the main text of this theorem.

> **Remark F.1** (Meaning of Trainable Parameters). By trainable parameters we denote the parameters that differs according to $f$. This includes the parameters related to the input domain of $\mathcal{X}$, and excludes the constants (i.e., $0$ and $1$) in the network.

We estimate the number of trainable parameters by each layer in the network.

First, we do the estimation for the Linear layer. It consists of a sum over $N_x$ $v_j^\top \widetilde{Z}$, $j \in [N_x]$, and thus contain $dn \cdot N_x$ trainable parameters.

Then we do the estimation for $W_K$ and $W_Q$. We restate the construction of $W_K$ and $W_Q$:

$$W_K := \begin{bmatrix} R & 0_{1\times d} & \cdots & 0_{1\times d} & 0_{1\times d} & \cdots & 0_{1\times d} \\ 0 & -R\frac{\|v_0\|_2^2}{2}1_{1\times d} & \cdots & -R\frac{\|v_{N_x-1}\|_2^2}{2}1_{1\times d} & -R\frac{\|v_0\|_2^2}{2}1_{1\times d} & \cdots & -R\frac{\|v_{N_x-1}\|_2^2}{2}1_{1\times d} \\ 0 & \ln(T(\widetilde{v}_0))^\top & \cdots & \ln(T(\widetilde{v}_{N_x-1}))^\top & \ln(E(\widetilde{v}_0))^\top & \cdots & \ln(E(\widetilde{v}_{N_x-1}))^\top \end{bmatrix},$$

$$W_Q := \begin{bmatrix} 0 & R1_{1\times n} & 0_{1\times(2dN_x-n)} \\ 0 & R1_{1\times n} & 0_{1\times(2dN_x-n)} \\ 0_n & I_n & 0_{n\times(2dN_x-n)} \end{bmatrix}.$$

From this, we observe they combined together have $2d \cdot N_x + 2dn \cdot N_x$ trainable parameters.

Finally, For $W_V$ and $W_O$, we restate their definition:

$$W_V := \begin{bmatrix} 0_d & X_1 & -X_1 \end{bmatrix},$$

$$W_O := \begin{bmatrix} dB_0 I_n \\ 0_{(2dG-n)\times n} \end{bmatrix},$$

where

$$X_1 := \begin{bmatrix} I_d & I_d & \cdots & I_d \end{bmatrix}_{d\times dG}.$$

$W_O$ contains $n$ trainable parameters ($dB_0$).

In conclusion, the whole network contains a total of

$$dnN_x + 2dN_x + 2dnN_x + n = 4dnN_x + 2dN_x + n,$$

trainable parameters, which is of $\mathcal{O}(dnN_x)$ level.

This completes the proof. $\qquad\square$

# G Extended Related Work

Recent empirical studies also shed light on the practical behavior of attention mechanism. Olsson et al. [2022] show that induction heads help models learn patterns in context. Sanford et al. [2024a] prove that Transformers can do complex computations with few layers because they work in parallel. In contrast, Luo et al. [2022] find that some Transformer designs lose expressivity when using relative positional encodings. Our work builds on these ideas. We prove that a single-layer, single-head softmax attention with a simple linear layer can approximate any continuous function on a compact domain. This shows that attention alone can learn arbitrary sequence-to-sequence mappings.

# H Extension to Transformer Setting

In this section, we demonstrate how our proof technique in attention extends to the transformer setting. Specifically, we show it is possible to use a three-layer multi-head attention-only network to achieve universal approximation in $L_p$ norm.

**Lemma H.1** (Attention Simulate Column-Wise Transformation). For an input $X \in \mathbb{R}^{d \times n}$. Let $l(X) := AXB, A \in \mathbb{R}^{d_{\text{output}} \times d}, B \in \mathbb{R}^{n \times n}$ be the linear operation (both token-wise and sequence-wise included) we wish to get. Without loss of generality, suppose $B$ has positive entries (since subtraction of positive matrices yields all matrix). We use an all-zero padding token. Let $I_{n+1}$ be the positional encoding below $X$. Then for any $\epsilon > 0$, there exists an single-head attention $\text{Attn}$ such that

$$\left\| \text{Attn}\left( \begin{bmatrix} X & 0_d \\ I_n & 0_n \\ 0_{1 \times n} & 1 \end{bmatrix} \right) - \begin{bmatrix} l(X) & 0_{d_{\text{output}}} \end{bmatrix} \right\|_\infty \leq \epsilon.$$

*Proof.* Let $s_i$ be the sum of all entries of $B_{:,i}$ ($i$-th column) and $M := max_{i \in [n]} s_i$. Define $S = [s_1 \cdots s_n]$. Construct $W_V := 3M * [A \quad 0_{d_{\text{output}} \times (n+1)}], W_K := [0_{n \times d} \quad ln(B^\top) \quad ln(3M \cdot 1_n - S^\top)]$ ($ln$ is entry-wise), and $W_Q := [0_{n \times d} \quad I_n \quad T \cdot 1_n]$ (T will be turned to arbitrarily large), then the output of attention is $3MA[X \quad 0_d]Softmax([ln(B^\top) \quad ln(3M \cdot 1_n - S^\top)]^\top \quad [I_n \quad T \cdot 1_n])$.

This equals

$$3MA[X \quad 0_d]Softmax\left( \begin{bmatrix} ln(B) & T \cdot H_1 \\ ln(3M \cdot 1_{1 \times n} - S) & T \cdot H_2 \end{bmatrix} \right),$$

in which $H_1, H_2$ are the sum of all columns in $ln(B), ln(3M \cdot 1_{1 \times n} - S)$ respectively.

This further equals to

$$3MA[X \quad 0_d] \begin{bmatrix} \frac{B}{3M} & a_0 \\ 1_{1 \times n} - \frac{S}{3M} & a_1 \end{bmatrix} = [AXB \quad C], ([a_0 \quad a_1]^\top \text{ is the } T\text{-related column}),$$

in which $C = 3MAXa_0$, when $T \to +\infty$, $a_0 \to 0_{d_{\text{output}}}$ and the padding token is preserved as well.

$\square$

**Lemma H.2** (Preservation of Identity Matrix). For any $\epsilon > 0$ and $n \in \mathbb{N}^+$, there exists an attention head $\text{Attn}$ that satisfies

$$\|\text{Attn}(I_n) - I_n\|_\infty \leq \epsilon.$$

*Proof.* Construct the $W_K, W_Q, W_V$ matrices of this attention head as

$$W_K := RI_n$$
$$W_Q := I_n$$
$$W_V := I_n.$$

The output of this layer is

$$\text{Softmax}(RI_n)$$

Hence when $R$ is sufficiently large, the output is within an arbitrarily small error to $I_n$. $\square$

**Corollary H.0.1.** For any $\epsilon > 0$ and input of form

$$\begin{bmatrix} A \\ I_n \end{bmatrix},$$

where $A \in \mathbb{R}^{m \times n}$ denotes any matrix, there exists an attention head $\text{Attn}$ such that

$$\left\| \text{Attn}\left( \begin{bmatrix} A \\ I_n \end{bmatrix} \right) - \begin{bmatrix} 0_{m \times n} \\ I_n \end{bmatrix} \right\|_\infty \leq \epsilon.$$

The proof is obvious. We only have to zero out $A$ in $W_K, W_Q, W_V$ and do the rest as Lemma H.2.

We now prove the universal approximation result in the transformer setting.

**Theorem H.1.** Let $X \in \mathbb{R}^{d \times n}$ denote the input sequence. Let $f$ be a continuous function on a compact support. For any $\eta > 0$, there exists a three-layer multi-head attention $\text{Attn}^{(i)}, i \in [3]$ such that

$$\left\| \text{Attn}^{(3)} \circ \text{Attn}^{(2)} \circ \text{Attn}^{(1)} \left( \begin{bmatrix} X & 0_d \\ I_n & 0_n \\ 0_{1 \times n} & 1 \end{bmatrix} \right)_{:,1:n} - f(X) \right\|_{L_p} \leq \eta.$$

### The First Layer

Let $v_i, i \in [P^{dn}]$ be as defined in Remark E.1.

Let $\widetilde{X}$ represent the flattened input $X$ and let $\widetilde{v}$ represents the $\mathbb{R}^{d \times n}$ sequence form of $v \in \mathbb{R}^{dn}$.

Let $\delta_i, i \in [d]$ be defined as

$$\delta_i := \begin{bmatrix} 0_{i-1} \\ \frac{1}{P} \\ 0_{d-i} \end{bmatrix} \in \mathbb{R}^d.$$

Let $\text{Attn}_h^{(1)}, h \in [n]$ denote the $h$-th head of $\text{Attn}^{(1)}$. According to Lemma H.1, for any $\epsilon_0 > 0$, there exists such set of $\text{Attn}_h$ such that

$$\left\| \text{Attn}_h^{(1)} \left( \begin{bmatrix} X & 0_d \\ I_n & 0_n \\ 0_{1 \times n} & 1 \end{bmatrix} \right) - \begin{bmatrix} l_h(X) & 0_{(d+1)n} \end{bmatrix} \right\|_\infty \leq \epsilon_0,$$

in which

$$l_h(X) := \begin{bmatrix} 0_{d(h-1) \times n} \\ x_h \cdot 1_{1 \times n} \\ 0_{[d(n-h)+n] \times n} \end{bmatrix}, h \in [n]$$

And construct the $n+1$-th head to preserve the identity matrix using Corollary H.0.1 ($A$ having the dimension of $dn \times (n+1)$ and the identity matrix preserved being $I_{n+1}$ in this case). So the output approximates

$$\begin{bmatrix} \widetilde{X} \cdot 1_{1 \times n} & 0_{dn} \\ I_n & 0_n \\ 0_{1 \times n} & 1 \end{bmatrix} \tag{H.1}$$

to an arbitrarily small error in the infinite norm.

The first attention layer is then defined as

$$\text{Attn}^{(1)} := \sum_{h=1}^{n+1} \text{Attn}_h^{(1)}$$

### The Second Layer

Label the heads in the second layer with $(v, g, s)$ pair, $v \in [P^{dn}], g \in [n], s \in \{\pm 1\}$. Construct parameters in $\text{Attn}_{(v,g,s)}^{(2)}$ to be

$$W_Q := \begin{bmatrix} 0_{n \times (g-1)d} & D & 0_{n \times (n-g)d} & -D \cdot (\widetilde{v})_{:,g} \cdot 1_{1 \times n} & 0 \\ 0_{1 \times (g-1)d} & 0_{1 \times d} & 0_{1 \times (n-g)d} & s \cdot \frac{1}{2P^2} \cdot 1_{1 \times n} & 0 \end{bmatrix}$$

$$W_K := R \begin{bmatrix} 0_{n \times dn} & I_n & 0_n \\ 0_{1 \times dn} & 1_{1 \times n} & 0 \end{bmatrix},$$

in which $D$ is

$$D := \begin{bmatrix} \delta_1^\top \\ \vdots \\ \delta_n^\top \end{bmatrix}$$

and $R \in \mathbb{R}^+$ is a parameter that will be set in later process.

Then the attention score matrix of $\mathrm{Attn}_{(v,g,s)}^{(2)}$ is

$$\mathrm{Softmax}\left( R \begin{bmatrix} I_n & 0_n \\ 1_{1\times n} & 0 \end{bmatrix}^\top \begin{bmatrix} D(x_g - (\widetilde{v}_v)_{:,g}) \cdot 1_{1\times n} & 0_n \\ s \cdot \frac{1}{2P^2} \cdot 1_{1\times n} & 0 \end{bmatrix} \right)$$

$$= \mathrm{Softmax}\left( R \begin{bmatrix} D(x_g - (\widetilde{v}_v)_{:,g}) \cdot 1_{1\times n} + s \cdot \frac{1}{2P^2} \cdot 1_{n\times n} & 0_n \\ 0_{1\times n} & 0 \end{bmatrix} \right)$$

$$= \mathrm{Softmax}\left( R \begin{bmatrix} D(x_g - (\widetilde{v}_v)_{:,g}) \cdot 1_{1\times n} & 0_n \\ -s \cdot \frac{1}{2P^2} \cdot 1_{1\times n} & 0 \end{bmatrix} \right)$$

When $R$ is taken sufficiently large, for any given $\epsilon_1 > 0$ and an arbitrary $\Delta_s > 0$ invariant of $\epsilon_1$ and $R$, if

$$\delta_i^\top (x_g - (\widetilde{v}_v)_{:,g}) > \frac{1}{2P^2} + \Delta_s$$

for a specific $i$, then in the attention score matrix of $\mathrm{Attn}_{(v,g,1)}^{(2)}$, the $i$-th row should be larger than the last row by $\Delta_s$, and this difference after amplified by $R$ makes the last row to be lesser than the given $\epsilon_1$.

Similarly, when

$$\delta_i^\top (x_g - (\widetilde{v}_v)_{:,g}) < -\frac{1}{2P^2} - \Delta_s$$

for a specific $i$, the $(n+1, j)$ entries in $\mathrm{Attn}_{(v,g,-1)}^{(2)}$ $(j = 1, \cdots, n)$ are smaller than $\epsilon_1$ when setting $R$ to be sufficiently large. And the sum the rest of the $j - th$ column is larger than $1 - \epsilon_1$.

Finally, when $|\delta_i^\top (x_g - (\widetilde{v}_v)_{:,g})| < \frac{1}{2P^2} - \Delta_s$, for all $i \in [n]$, it makes the last row largest in every entry (except the last column). Thus there exists a sufficiently large $R$, such that the last row of the attention score matrix is larger than $1 - \epsilon_1$ in every entry. This also makes the sum of the rest of the rows less than or equal to $\epsilon_1$ in every column.

We show that the last situation is equivalent to the input token $x_g$ falling into the grid centered at $(\widetilde{v}_v)_{:,g}$ in $\mathbb{R}^d$. (notice $v_v$ is a grid point in $\mathbb{R}^{dn}$ and its column is a grid point in $\mathbb{R}^d$)

$$\delta_i^\top (x_g - (\widetilde{v}_v)_{:,g}) - \frac{1}{2P^2} = \delta_i^\top (x_g - (\widetilde{v}_v)_{:,g}) - \frac{1}{2}\|\delta_i\|_2^2$$

$$= \frac{1}{2}(\|x_g - (\widetilde{v}_v)_{:,g}\|_2^2 - \|x_g - ((\widetilde{v}_v)_{:,g} + \delta_i)\|_2^2)$$

$$-\delta_i^\top (x_g - (\widetilde{v}_v)_{:,g}) - \frac{1}{2P^2} = -\delta_i^\top (x_g - (\widetilde{v}_v)_{:,g}) - \frac{1}{2}\|\delta_i\|_2^2$$

$$= \frac{1}{2}(\|x_g - (\widetilde{v}_v)_{:,g}\|_2^2 - \|x_g - ((\widetilde{v}_v)_{:,g} - \delta_i)\|_2^2)$$

Therefore

$$|\delta_i^\top (x_g - (\widetilde{v}_v)_{:,g})| < \frac{1}{2P^2} - \Delta_s$$

is equivalent to $x_g$ being closer to $(\widetilde{v}_v)_{:,g}$ than its neighboring grid points in $\mathbb{R}^d$ (which are $\{(\widetilde{v}_v)_{:,g} \pm \delta_i | i \in [d]\}$), this concludes it to be within the grid centered at $(\widetilde{v}_v)_{:,g}$.

This means that when $x_g$ is within the grid centered at $\widetilde{v}_v$, both heads in $\mathrm{Attn}^{(2)}_{(v,g,s)}, s \in [\pm 1]$ has their last row in the attention score matrix larger than $1 - \epsilon_1$ (which makes their sum larger than $2 - 2\epsilon_1$) and if $x_g$ is not in the grid, their sum would be smaller than $1 + \epsilon_1$ (because at least one is smaller $\epsilon$ and all entries should be smaller than 1)

Let $W_V$ of head labeled $(v, g, s)$ be

$$\begin{bmatrix} 0_{s_v \times dn} & 0_{s_v \times n} & 0_{s_v} \\ 0_{1 \times dn} & 1_{1 \times n} & 0 \\ 0_{t_v \times dn} & 0_{t_v \times n} & 0_{t_v} \end{bmatrix}$$

in which $s_v := vd + (g-1), t_v = P^{dn}n - s_v + n - 1$. We note that here head $(v, g, 1)$ and $(v, g, -1)$ have the same head.

Then $V$ is

$$\begin{aligned} V &:= W_V \begin{bmatrix} \widetilde{X} \cdot 1_{1 \times n} & 0_{dn} \\ I_n & 0_n \\ 0_{1 \times n} & 1 \end{bmatrix} \\ &= \begin{bmatrix} 0_{s_v \times dn} & 0_{s_v \times n} \\ 0_{n \times dn} & 1_{1 \times n} \\ 0_{t_v \times dn} & 0_{t_v \times n} \end{bmatrix} \begin{bmatrix} \widetilde{X} \cdot 1_{1 \times n} & 0_{dn} \\ I_n & 0_n \end{bmatrix} \\ &= \begin{bmatrix} 0_{s_v \times n} & 0 \\ 1_{1 \times n} & 0 \\ 0_{t_v \times n} & 0 \end{bmatrix} \in \mathbb{R}^{(P^{dn}n+n) \times (n+1)} \end{aligned}$$

The output of the $(v, g, s)$-th head in the second attention layers is

$$V \cdot S_a^{(v,g,s)} = \begin{bmatrix} 0_{s_v \times n} & 0_{s_v} \\ 1_{1 \times n} - (S_a^{(v,g,s)})_{n+1,1:n} & \frac{n}{n+1} \\ 0_{t_v \times (n+1)} & 0_{t_v} \end{bmatrix},$$

in which

$$S_a^{(v,g,s)} := \mathrm{Softmax}\left( R \begin{bmatrix} D(x_g - (\widetilde{v}_v)_{:,g}) \cdot 1_{1 \times n} & 0_n \\ -s \cdot \frac{1}{2P^2} \cdot 1_{1 \times n} & 0 \end{bmatrix} \right)$$

is the previously constructed attention score of the $(v, g, s)$ head.

Finally, add a head $\mathrm{Attn}^{(2)}_I$ to preserve the identity matrix as that done in the first layer using Corollary H.0.1.

This head outputs (with an arbitrarily small error)

$$\begin{bmatrix} 0_{P^{dn}n \times n} & 0_{P^{dn}n \times 1} \\ I_n & 0_n \\ 0_{1 \times n} & 1 \end{bmatrix}.$$

Define $\mathrm{Attn}^{(2)}$ as the sum of previously defined heads

$$\mathrm{Attn}^{(2)} := \sum_{v,g,s} \mathrm{Attn}^{(2)}_{(v,g,s)} + \mathrm{Attn}^{(2)}_I$$

Then we have that when the input is

$$\begin{bmatrix} \widetilde{X} \cdot 1_{1 \times n} & 0_{dn} \\ I_n & 0_n \\ 0_{1 \times n} & 1 \end{bmatrix},$$

the output of this layer is within an arbitrary small error to

$$
\begin{bmatrix}
2_{1\times n} - S(1,1) & \frac{n}{n+1} \\
\vdots & \vdots \\
2_{1\times n} - S(1,n) & \frac{n}{n+1} \\
\vdots & \vdots \\
2_{1\times n} - S(P^{dn},n) & \frac{n}{n+1} \\
I_n & 0_n \\
0_{1\times n} & 1
\end{bmatrix},
\tag{H.2}
$$

in which

$$
S(v,g) := (S_a^{(v,g,-1)})_{n+1,1:n} + (S_a^{(v,g,1)})_{n+1,1:n}.
$$

We note that only when input is within the grid centered at $v$, $2_{1\times n} - S(v,g)$ is smaller than $2 - (2 - 2\epsilon_1) = 2\epsilon_1$ for every $g$ in all entries.

**The Third Layer**

The third layer consists of $P^{dn}$ blocks, the $v$-th block identifies if the input falls inside the $v$-th grid.

For the $v$-th block, construct its $W_K, W_Q$ as

$$
W_K := \begin{bmatrix}
0_{n\times(v-1)n} & 0_{n\times n} & 0_{n\times(P^{dn}-v)n} & \frac{1}{2}I_n & 0_n \\
0_{1\times(v-1)n} & 1_{1\times n} & 0_{1\times(P^{dn}-v)n} & 0_{1\times n} & -\frac{P^{dn}n^2}{n+1}
\end{bmatrix}
$$

$$
W_Q := R_1 \begin{bmatrix} 0_{(n+1)\times P^{dn}n} & I_{n+1} \end{bmatrix},
$$

in which $R_1$ is a scaler whose size we will determine in later process.

When the input has the form of that in (H.2). The $K$ and $Q$ matrices are

$$
K := W_K \begin{bmatrix}
2_{1\times n} - S(1,1) & \frac{n}{n+1} \\
\vdots & \vdots \\
2_{1\times n} - S(1,n) & \frac{n}{n+1} \\
\vdots & \vdots \\
2_{1\times n} - S(P^{dn},n) & \frac{n}{n+1} \\
I_n & 0_n \\
0_{1\times n} & 1
\end{bmatrix}
$$

$$
= \begin{bmatrix}
n \cdot 2_{1\times n} - \sum_{g\in[n]} S(v,g) & \frac{1}{2}I_n & 0_n \\
& & 0
\end{bmatrix} \in \mathbb{R}^{(n+1)\times(n+1)}
$$

$$
Q := R_1 \cdot I_{n+1}.
$$

This yields the attention score matrix to be

$$
\mathrm{Softmax}(K^\top Q) = \mathrm{Softmax}(R_1 \begin{bmatrix} n \cdot 2_{1\times n} - \sum_{g\in[n]} S(v,g) & \frac{1}{2}I_n & 0_n \\ & & 0 \end{bmatrix})
$$

By $n \cdot 2_{1\times n} - \sum_{g\in[n]} S(v,g)$ is smaller than $2n\epsilon_1$ if input is in the grid centered at $v$ and larger than $1 - \epsilon_1$ if not. Set $\epsilon_1 = 1/4n$, we get that for any $\epsilon_2 > 0$, when $R_1$ is sufficiently large,

$$
\| \mathrm{Softmax}(K^\top Q) - \begin{bmatrix} I_n & \frac{1}{n+1} \cdot 1_n \\ 0_{1\times n} & \frac{1}{n+1} \end{bmatrix} \|_\infty \le \epsilon_2,
$$

when input is within the grid centered at $v$.

And when the input is not in the grid centered at $v$, we have

$$
\| \mathrm{Softmax}(K^\top Q) - \begin{bmatrix} 0_{n\times n} & \frac{1}{n+1} \cdot 1_n \\ 1_{1\times n} & \frac{1}{n+1} \end{bmatrix} \|_\infty \le \epsilon_2
$$

Construct $W_V$ as

$$\begin{bmatrix} 0_{d \times P^{dn}n} & \frac{1}{2}f(\widetilde{v_v}) & 0_d \end{bmatrix}$$

This makes the $V$ matrix

$$V = \begin{bmatrix} f(\widetilde{v_v}) & 0_d \end{bmatrix},$$

when input has the form in (H.2).

Thus the third layer's $v$-th block outputs

$$V \operatorname{Softmax}(K^\top Q) = \begin{bmatrix} f(\widetilde{v_v}) & 0_n \end{bmatrix} \cdot \operatorname{Softmax}(R_1 \begin{bmatrix} \frac{1}{2}I_n & 0_n \\ n \cdot 2_{1 \times n} - \sum_{g \in [n]} S(v,g) & 0 \end{bmatrix}).$$

Then for any $\epsilon_3 > 0$, there exists an $R_1$ such that the output of the third layer satisfies

$$\| V \operatorname{Softmax}(K^\top Q) - \begin{bmatrix} f(\widetilde{v_v}) & * \end{bmatrix} \|_\infty \le \epsilon_3,$$

when the input falls inside the grid centered at $v_v$.

And

$$\| V \operatorname{Softmax}(K^\top Q) - \begin{bmatrix} 0_{d \times n} & * \end{bmatrix} \|_\infty \le \epsilon_3,\text{'}$$

when the input does not fall inside the grid centered at $v_v$. Here $*$ denotes the padded column that is not considered in the final output and is hence omitted.

Now we sum all these heads and get the final output of $\operatorname{Attn}^{(3)}$. The final output is within an error of $P^{dn}\epsilon_3$ to

$$\begin{bmatrix} f(\widetilde{v_v}) & * \end{bmatrix} + \begin{bmatrix} (P^{dn} - 1) \cdot 0_{d \times n} & * \end{bmatrix}$$

with $v_v$ being the center of the grid that has the input sequence in it.

**Analysis of Total Error**

Let $\Omega$ be the compact support of $f$ with finite measure $\mu(\Omega) = V$.

Let $F(x) = (\operatorname{Attn}^{(3)} \circ \operatorname{Attn}^{(2)} \circ \operatorname{Attn}^{(1)})(x)$ denote the output of the network given ideal inputs $Z_i$, and let $F^*(x)$ denote the output given perturbed inputs $Z_i^*$ (which are the actual inputs).

We have formerly proven the following conditions for any $\epsilon > 0$:

- **Ideal Approximation:** $\|F(x) - f(x)\|_\infty < \epsilon$.
- **Perturbed Inputs:** The input to the third layer, $Z_3^*(x)$, satisfies $\|Z_3^*(x) - Z_3(x)\|_\infty \le \epsilon$ on a set $\Omega \setminus B_\delta$, where $B_\delta$ is a region of arbitrarily small measure $\mu(B_\delta) < \delta$. On $B_\delta$, the error exceeds $\epsilon$ but is still bounded.

Given the conditions above, for any precision $\eta > 0$ and any $p \in [1, \infty)$, the actual output $F^*$ approximates the target function $f$ in the $L_p$ norm such that $\|F^* - f\|_p < \eta$.

We aim to bound the $L_p$ error $\|F^* - f\|_p$. By the Minkowski inequality (triangle inequality for $L_p$ norms), we can separate the error into an approximation component and a stability component:

$$\|F^* - f\|_p \le \|F^* - F\|_p + \|F - f\|_p. \tag{H.3}$$

We analyze these two terms separately.

**1. Bounding the Approximation Error**

From the problem statement, the ideal network approximates $f$ uniformly with error $\epsilon$. Using the definition of the $L_p$ norm on the domain $\Omega$:

$$\|F - f\|_p = \left( \int_\Omega \|F(x) - f(x)\|^p \, d\mu(x) \right)^{1/p} \le \left( \int_\Omega \epsilon^p \, d\mu(x) \right)^{1/p} = \epsilon V^{1/p}. \tag{H.4}$$

Since $\epsilon$ can be chosen arbitrarily small, this term converges to 0 as $\epsilon \to 0$.

## 2. Bounding the Stability Error

Consider the term $||F^* - F||_p$, which represents the deviation caused by the perturbed intermediate inputs. Recall that $F(x) = \text{Attn}^{(3)}(Z_3(x))$ and $F^*(x) = \text{Attn}^{(3)}(Z_3^*(x))$. To evaluate the integral over $\Omega$, we partition the domain into the "good" set $S = \Omega \setminus B_\delta$ and the "bad" set $B_\delta$:

$$||F^* - F||_p^p = \int_S ||\text{Attn}^{(3)}(Z_3^*) - \text{Attn}^{(3)}(Z_3)||^p \, d\mu + \int_{B_\delta} ||\text{Attn}^{(3)}(Z_3^*) - \text{Attn}^{(3)}(Z_3)||^p \, d\mu.$$

$$\text{(H.5)}$$

**Analysis on $S$ (Lipschitz Continuity):** The Multi-Head Attention layer $\text{Attn}^{(3)}$ consists of linear projections and the Softmax function, which are smooth and differentiable operations. Consequently, $\text{Attn}^{(3)}$ is locally Lipschitz continuous on compact domains. Let $K$ be the Lipschitz constant for $\text{Attn}^{(3)}$. On the set $S$, we are given that $||Z_3^* - Z_3|| \leq \epsilon$. Therefore:

$$||\text{Attn}^{(3)}(Z_3^*) - \text{Attn}^{(3)}(Z_3)|| \leq K||Z_3^* - Z_3|| \leq K\epsilon. \tag{H.6}$$

The integral over $S$ is bounded by:

$$\int_S ||\text{Attn}^{(3)}(Z_3^*) - \text{Attn}^{(3)}(Z_3)||^p \, d\mu \leq \mu(S)(K\epsilon)^p \leq V(K\epsilon)^p. \tag{H.7}$$

**Analysis on $B_\delta$ (Uniform Boundedness):** On the region $B_\delta$, the Lipschitz bound fails. However, since $\Omega$ is compact and the functions comprising the network are continuous, the image of the network is bounded. There exists a constant $M > 0$ such that $||F(x)|| \leq M$ and $||F^*(x)|| \leq M$ for all $x \in \Omega$. By the triangle inequality, the pairwise difference is bounded by $2M$. Thus:

$$\int_{B_\delta} ||\text{Attn}^{(3)}(Z_3^*) - \text{Attn}^{(3)}(Z_3)||^p \, d\mu \leq \int_{B_\delta} (2M)^p \, d\mu = \delta(2M)^p. \tag{H.8}$$

Combining these results, the stability error satisfies:

$$||F^* - F||_p \leq \left(V(K\epsilon)^p + \delta(2M)^p\right)^{1/p}. \tag{H.9}$$

## 3. Conclusion

Substituting the bounds back into the original inequality, we obtain:

$$||F^* - f||_p \leq \epsilon V^{1/p} + \left(V K^p \epsilon^p + \delta(2M)^p\right)^{1/p}. \tag{H.10}$$

To ensure $||F^* - f||_p < \eta$, we select parameters as follows:

- Choose $\epsilon$ sufficiently small such that both $\epsilon V^{1/p}$ and $V^{1/p}K\epsilon$ are smaller than $\eta/3$.
- Choose the region of perturbation $B_\delta$ sufficiently small such that $\delta < (\eta/3)^p/(2M)^p$.

Since $\epsilon$ and $\delta$ can be made arbitrarily small, for any $\eta > 0$, the condition holds, and we get the global $L_p$ approximation error converges to zero.

