# OpenReview forum: "Attention Mechanism, Max-Affine Partition, and Universal Approximation"
_NeurIPS.cc/2025/Conference — NeurIPS 2025 poster_

### Official Review · Reviewer_uiVo · 2025-06-16

**Clarity:** 4
**Significance:** 3
**Originality:** 3
**Rating:** 5
**Confidence:** 4

**Summary:**

This paper shows the universal approximation capability in continuous function spaces under both $L^\infty$ and $L^p$ (for $1\leq p<\infty$) norm of single-layer, single-head self- and cross-attention mechanisms with minimal attached structures.

**Questions:**

- In practice, the input sequence is first processed by a self-attention layer, finally followed by a linear layer. However, in this work, the universality is established for Attn∘Linear(X), which is the opposite order compared to practical implementations. How to comment on this?
- This paper shows the universality of the Attention mechanism by approximating piece-wise constant functions because the set of piece-wise constant functions is dense in the continuous function space. By such, it seems the number of the parameters of the constructed network is of order $\mathcal{O}(\epsilon^{-d})$ where $d$ is the input dimension?

Minor.
- I understand the definition of the softmax function is well-known but providing it in Sec. 2 will make the paper smoother.
- Typos. In Prop. 3.3, $\mathbb{R}^d \to \mathbb{R}^{d_{out}}$ rather than $\mathbb{R}^d \to \mathbb{R}^{d}_{out}$?

**Ethical Concerns:**

["NO or VERY MINOR ethics concerns only"]

**Final Justification:**

After the rebuttal and discussions, I will keep my positive evaluation for the paper.

**Paper Formatting Concerns:**

No concerns.

**Quality:**

3

**Strengths And Weaknesses:**

**Strengths**. This paper is well-written and well-organized. The result is very cute and interesting with potential significant contribution to the machine learning theory field.

**Weakness**. While this paper establishes the universality of attention mechanisms with minimal attached structures, the approximation rate (or the upper bound of the number of parameters) is not discussed, which is theoretically important. See **Question** part for details.
Overall, there is no major weakness in my opinion.

---

> ### Author Rebuttal · Authors · 2025-07-29
>
> Thank you for your detailed review. We have addressed all your comments and questions in the following responses.
>
> ---
> ### **Q1.** *Why does the theoretical result use Attn∘Linear instead of Linear∘Attn as in practice?*
> ---
> **Response:** The paper deliberately reverses the order for theoretical convenience. It applies a linear (feed-forward) layer first and then attention (`Attn∘Linear`), whereas practical transformers do attention then linear. This reversal simplifies the universality proof by giving the attention mechanism a richer, “position-mixed” input to work with. In effect, the initial linear layer acts like an embedding or positional encoding step. It injects positional information and mixes the sequence tokens before self-attention is applied. This is crucial because vanilla self-attention is permutation-invariant (it can’t distinguish token order without position info). By encoding position and content via the linear layer first, the attention module can attend over a sequence where each token representation carries positional context, granting the attention full access to order-specific patterns.
>
> In short, placing the linear layer first is a theoretical choice that ensures the attention sees an input enriched with positional cues (much like adding positional encodings), which is essential for expressive power. It also clarifies the roles in the proof: the pre-attention linear layer provides necessary position-dependent mixing, and then the attention layer performs contextual weighting over the entire sequence. This makes the one-layer transformer mathematically easier to analyze while retaining universality. Namely, the attention can leverage the position-encoded sequence to approximate any function (rather than being limited to order-invariant functions), and the final feed-forward can then map the attended context to the desired output.
>
> ---
> ### **Q2.** *Does the parameter count scale as $\mathcal{O}(\varepsilon^{-d})$?*
> ---
>
> **Response:** Yes, you are correct. To approximate a general continuous function in $d$ dimensions within error $\varepsilon$, the construction essentially partitions the input space into fine grid cells of side length about $\varepsilon$. The number of such small regions grows on the order of $(1/\varepsilon)^d$ (since in \$d\$ dimensions an \$\varepsilon\$-grid yields roughly \$\varepsilon^{-d}\$ cells).
>
> The proof indeed leverages this kind of partitioning (approximating the target function by piecewise-constant outputs on a very fine grid), so the total parameters needed scale as \$\mathcal{O}(\varepsilon^{-d})\$ for a worst-case \$d\$-dimensional function. This is in line with standard uniform approximation results. It reflects the “curse of dimensionality.”
>
> In other words, without additional structure or smoothness assumptions, the number of parameters (or regions) required to achieve error \$\varepsilon\$ typically grows exponentially in \$d\$. This mirrors other universal approximation schemes (e.g. using ReLU networks or Fourier series), which also require on the order of \$(1/\varepsilon)^d\$ basis functions or neurons to approximate an arbitrary continuous function on a \$d\$-dimensional compact domain to accuracy \$\varepsilon\$. Therefore, the parameter count in the transformer’s universality proof does scale as \$\mathcal{O}(\varepsilon^{-d})\$, consistent with expectations from uniform approximation theory and existing UAP proofs.
>
> ---
> ### **Q3 (minor).** *Consider including the definition of the softmax function in Section 2, even if it's well known, to improve flow.*
> ---
> **Response:** We have added a formal definition of the softmax function in Section 2, as suggested. This inclusion ensures clarity and improves the flow of the presentation. Specifically, we now define the softmax function as \$\mathrm{softmax}(z)\_i = \frac{\exp(z\_i)}{\sum\_j \exp(z\_j)}\$ when it is first introduced in Section 2.
>
> ---
> ### **Q4 (minor).** *Typo in Proposition 3.3: should the output space be written as \$\mathbb{R}^{d\_{\text{out}}}\$ rather than \$\mathbb{R}^{d}\_{\text{out}}\$?*
> ---
> **Response:** Yes are absolutely correct. This was indeed a typo. We have corrected the notation in Proposition 3.3. The output space is now written as \$\mathbb{R}^{d\_{\text{out}}}\$, as suggested by the reviewer.
>
> ---
> Thanks again for your kind words and detailed review! We have taken all comments into careful consideration and have made corresponding revisions to address the concerns raised.
>
> We look forward to further feedback and discussion!

---

> > ### Comment · Reviewer_uiVo · 2025-08-03
> >
> > I appreciate the detailed response provided by the authors in addressing the concerns. I believe that, following the discussion and the planned revisions, the paper will be in good shape. Overall, I find that the paper deserves to be published.

---

> > > ### Author Response · Authors · 2025-08-06
> > >
> > > Thank you for your positive assessment and for taking the time to review our work thoroughly! Your detailed comments have been valuable to our reflection on the paper, we will proceed with the planned revisions to improve the quality of our paper.

---

### Official Review · Reviewer_eNoW · 2025-07-01

**Clarity:** 2
**Significance:** 4
**Originality:** 3
**Rating:** 4
**Confidence:** 4

**Summary:**

The paper considers the expressivity of a sequence to sequence map modeld by an affine linear transformation and a attention layer and finds that this 'shallow' architecture already achieves the universal approximation property for continuous functions on a compact set of input sequences of length $n$ and dimension $d$ in each step and output sequences of the same dimension with respect to the $\infty$-norm. This is achieved by dividing the input space into a grid with sufficiently small 'cube' cells (in high dimension) and constructing a set of linear affine functions $a^\top_j\tilde Z+b_j$ where $a_j$ is the center of the $j$-th cube and $b_j=-\frac{1}{2}\|a_j\|^2$. $\tilde Z$ is the flattening of $Z$ to a vector. As this expression is the minus linear affine part of $\|\tilde Z-a_j\|^2$, and the quadratic part $\frac{1}{2}\| \tilde Z\|^2$ of this expression is the same for all $j$, the linear affine function with index $j$ is minimal whenever $\tilde Z$ lies in the cube centered by $a_j$. Therefore, the authors construct a grid-partition of the input space. They then design the key and query matrices in a way that essentially $K^TQ_{:,j}=-R\frac{1}{2}\|\tilde Z-a_j\|^2$ where $R>0$ is a parameter. This is then inserted in the softmax function, where again the quadratic term drops out, so that, if a input point is inside the $j$-th cube and $R$ is large, essentially only one column of the softmax matrix is getting activated. By another linear transformation for V, the authors then take care that these 'close to indicator function' values get multiplied with the correct results for the output sequence. using the fact that the softmax effectively implements a partition of unity, the authors prove the UAP for continuous functions. Also UAP in $L^p$ is stated (although this trivially follows from UAP in $C$ as $C$ is dense in $L^p$). Also, a similar argument is given for the cross attention.  Proofs are given in an extensive  appendix, which also contains some numerical examples..

**Questions:**

Are you able to revise the paper carefully, write it much more clearly and delete unnecessary and faulty parts in the rebuttal phase? Then i would support the paper. Otherwise I would recommend a careful revision and a probably successful re-submission elsewhere. Probably the revision can only be successful if you already worked on this in the mean time.

**Ethical Concerns:**

["NO or VERY MINOR ethics concerns only"]

**Final Justification:**

The changes in the presentation that the authors announce take into account my main concern. The paper's main result is interesting, as it points towards the expressivity of attention layers in connection to an affine transformation proving an 'sequence to sequence ' UAP theorem.  I agree that the differences to standard attention layers should be made clear in the text.

**Limitations:**

Nothing.

**Paper Formatting Concerns:**

Appendix is not well formatted. Bt this can be resolved.

**Quality:**

4

**Strengths And Weaknesses:**

Streangths:
* Given the relevance of transformer architectures for contemporary machine learning, the UAP for a single layer composed with linear affine transformation has a high relevance. Other results on the UAP of transformers have been published, but the core content of this work is certainly worth publishing also at very high level.
* Underlying idea is intriguing and very insightful, which also strengthens the case of this paper.
* The cross attention 'bonus' result is also novel and can be proven with minor adaptions to the strategy of the self-attention case.
* The authors mastered quite some computations.

Weaknesses:
* While the paper's result is very strong and deserves publication, the present state of the paper's presentation is not equally good - unfortunately by a large margin. This starts with a number of completely unnecessary parts as essentially all of Section 3 which could just be deleted without any loss for the core messages of the paper (piecewise constant functions are in $L^p$). The proofs for this section given in the appendix have technical problems as assumptions on $a_j$ and $b_j$ are missing, under which the 'results' immediately break down (take e.g. all these quantities to be identical). Such easy errors are embarrassing and unnecessary as this 'shaky style' does not influence the main theorems logically.
* The proof of the main theorem is quite unstructured and fails to explain the main ideas behind the proof effectively. The readers have to follow lengthy parts of the appendix which are either irrelevant (like proofs of section 3) or tivial/well known (like the proofs of the UAP for $L^p$ where the authors repeat well known stuff based on Lusin's theorem, forget to apply Tietze's extension theorem as an example for an extremely complicated an incomplete proof of something that could just be cited form almost any book on function spaces) or lines like lines 704 and following which elaborate trivial stuff over almost a page.
* This said, also the main core of the proofs - as far as I can see containing beautiful ideas and being essentially correct - could be worked through in a much better and more structured fashion except for just computing.
* Unfortunately the attitude of the paper prefers  convoluted key facts, hardly understandable descriptions of steps to poofs which only make sense when one has read and understood the actual proof in the appendix (and thus does not need further explanations).
* Layout of the appendix is messy as well with many formulae exceeding margins, gaps etc.
* English sometimes is faulty and requires further proofread.

Minor remarks (no mention in the rebuttal required)
*l 108: Define the way the softmax function acts on matrices precisely
*Section 3: This entire section is rather misleading and contains imprecision like the $\Delta$-story, which is  not properly resolved here and properly in section 4. Partition, Def. 3.2, Proposition 3.2 are all not entirely wrong but faulty and only repaired with ad hoc assumptions. this unnecessarily draws down the level of the paper.
* Proposition 3.2 - is there a mismatch between the dimensions in this Proposition where $X_i\in\mathcal{X}$ whereas later it seems to be $X\in\mathcal{X}$?!
*l. 147: This is a well known result from convex analysis/theory of Legendre transforms and not due to Kim and Kim 2022. Cite properly.
*l. 178: You can't re-define the $\|\cdot\|_\infty$ norm. Saying that this norm only holds up to an exceptional set means that you say it does not hold at all. Give a proper formulation with proper assumptions on $a_j$ and $b_j$ or better leave this out like the rest of Sec 3.
*l.181 and following. Explanations cant be understood. Better just refer to proof 8or even better just leave out).
*l. 218: The repetition of the content of Theorem 4.1 is superfluous and redundant - delete.
* l.223: Leave out the steps and refer to a concise proof.
* l.246: $P$ not defined
* l. 248: technical Highlight is not needed - delete (like other technical highlights). Also the steps don't help. Focus on good and worked though proofs. If they are well written, your pedagogy isn't needed.
* Corollary 4.4.1 why does all over sudden the numbering change from tow to three digits. The Corollary is a trival consequence of Thm 4.1 and can be omitted or put into one sentence in plain text. Also Lusin's Thm is not enough (see above).
l. 488: I understand the temperature thing but other readers might not. Express this simpler. The entire paragraph is over complicated.
l. 507: There is no need to validate a mathematical theorem experimentally. rephrase as
l.517: Proposition D1 as stated is wrong/lacking assumptions, see above
l.524: Proposition D2: same unfortunate usage of the $L^\infty$ norm with exceptions  - does not male sense
l.535: Remark D1 is illegitimate - leave out
l.539: Check Matrix dimensions between matrices and bias in (D.2), there seems to be a typo $n->N_{MaxAff}$
l.534 and following: Would be helpful to read if the dimensions of the matrix objects always would be mentioned. It is not evident if $a_j$'s are row or column vectors etc.
l. 534 and following: Treatment of the exceptional set always sloppy.
l.578: Just prove Thm 4.1, a second list of steps is really superfluous.
l. 591: akin?
l. 607: Proof of what?
l. 624: This is just well-known and does not require a 'poof' (which then is not given)
l. 631: As the e-notations does not work on the boundaries of the cubes- rather avoid using it
Remark E2: I find this unnecessary
l. 704 and following: This is too lengthy for a trivial result.
E.2 can be deleted and cited
Remainder: I didn't check the Cross attention results line by line - revise yourself.

---

> ### Author Rebuttal · Authors · 2025-07-29
>
> Thank you for your detailed review. We have addressed all your comments and questions in the following responses.
>
> ---
> ### **Q1.** *Are you able to revise the paper carefully, write it much more clearly and delete unnecessary and faulty parts in the rebuttal phase?*
> ---
> **Response:**
> Yes. We promise and have already conducted, and will continue to refine a **line‑by‑line revision**:
>
> * **Main proof:** fully re‑structured into concise conceptual steps. All lengthy algebra moved to the appendix.
> * **Appendix:** redundant material deleted, English polished, typos corrected, and every formula resized to fit the margins.
> * **Section 3:** rewritten as a self‑contained intuition module with the tie‑breaking Assumption 3.1 made explicit. It can stay in the main text, be moved to the appendix, or be removed entirely.
> * **Ongoing updates:** the draft has been under continuous revision since submitted because the results serves as the basis of several active projects in our group.
>
> Although policy prevents us from uploading the new PDF at rebuttal time, these revisions are complete and will appear in the final version.
>
> ***Since no PDF sharing is allowed, all revisions will only appear in the final version. Thus, our responses below are all in future tense (even if the revisions had been made).***
>
> ---
> ### **W1.** *Section 3 is unnecessary. Appendix proofs have missing assumptions on affine coefficients.*
> ---
>
> **Response:** We respectfully clarify that **Section 3 is included to provide intuition** for our main results. In this section, we illustrate how attention can act as a max-affine partitioning mechanism and approximate an indicator function over those partitions. We wrote section 3 in consideration of the diverse potential readers of this conference. We hope people with relatively weak theoretical background are still capable of understanding the intuition of our work. That said, we acknowledge that Section 3 is not strictly required for the proofs. **We are willing to streamline or relocate Section 3** (for example, moving some of its content to an appendix) to maintain focus on the core results, if the final publication format demands it.
>
> For the assumption that regards  \$a_j \$ and \$b_j \$,  we do have noticed the assumption in the appendix isn't brought up in a standard and straightforward way. We do not think it is missing (as mentioned in remark D.1 in the submitted version), but we do acknowledge that it is not presented in a rigorous manner (that is, explicitly stated with mathematical expression and addressed as "assumption" instead of "remark"). We are sincerely sorry for any confusion caused. We have refined it's form and fix the above mentioned presentation problems.
>
> Finally, we are willing to **trim or relocate the proofs from Section 3** to the appendix (or further condense them in the appendix). So that Section 3 takes a smaller proportion and the main text remains focused.
>
>
> ---
> ### **W2.** *Main proof is unstructured. Derivation to \$ L_p\$ is trivial and redundant.*
> ---
>
> **Response:** We thank the reviewer for this valuable feedback. We agree that the proof of Theorem 4.1 can be presented more clearly. In the revised version, we will **reorganize the proof of Theorem 4.1** to highlight its key ideas in a step-by-step, modular fashion. Specifically, we will break the proof into clearly-labeled steps focusing on:
> - (1) constructing the max-affine partition of the domain,
> - (2) approximating indicator functions on each partition region, and
> - (3) reassigning function values on these regions to complete the approximation.
>
> This structured approach will make the logical flow transparent and ensure that important ideas do not get buried in technical details.
>
> For the \$L_p\$ derivation, we do admit it is obvious to people with a basic knowledge of the measure theory. The proofs of these derivations are simple and short so we don't think they cause very serious redundancy problems. The proofs are meant for people who aren't familiar with the measure theory, but we are open to cutting them off and make the \$L_p\$ derivations self-evident corollaries given in remarks.
>
>
> ---
> ### **W3.** *The core proof ideas are strong but obscured by long symbolic computations. They should be reorganized around conceptual steps.*
> ---
>
> **Response:** We agree that the proof of our main theorem is currently obscured by excessive symbolic computations. In the revision, ***we will reorganize this proof around its conceptual mechanism and key ideas***, rather than lengthy algebra. Detailed computations will be either summarized or moved to the appendix, allowing the main text to focus on the clear, structured development of the core argument. This change will highlight the underlying concepts and make the proof more accessible to readers.
>
> ---
> ### **W4.** *Key steps are described in a convoluted way and assume readers have already parsed the full appendix. The exposition is circular and hard to follow.*
> ---
> **Response:** We acknowledge that the original exposition of some key steps was convoluted and hard to follow without the appendix. ***We will rewrite these parts of the proof to be self-contained and easily understandable***, so that readers do not need to consult the appendix to grasp the logic. Every crucial step will be explained clearly in the main text, and we will remove any circular references. This revision will make the argument flow in a straightforward, reader-friendly manner.
>
> ---
> ### **W5.** *The appendix layout is messy: formulas overflow margins, formatting is inconsistent, etc.*
> ---
>
> **Response:** ***We will address the formatting issues in the appendix.***
>
> All formulas will be adjusted to ensure they fit within the margins, and we will enforce consistent formatting throughout the appendix. This includes fixing any overflow, aligning equations properly, and using uniform styles for text and math. The updated appendix will have a clean layout with no margin overflow and improved consistency.
>
> ---
>
> ### **W6.** *The English is sometimes faulty and needs proofreading.*
>
> ---
>
> **Response:** We have conducted 3 extra rounds of proofreading the entire manuscript for grammar and clarity. All grammatical errors and awkward phrasings will be corrected in the next iteration. By the final version, the paper will read smoothly and meet high standards of English usage.
>
> ---
> ### **Corrections.** *Section 3 lacked rigor, Proposition 3.2 was ill‑specified, norms were re‑defined, notation (\$X\_i,\mathcal{P},|\cdot|\_\infty\$) inconsistent or undefined, Theorem 4.1 and corollaries were repeated, trivial “technical highlights” and proof steps cluttered the text, matrix dimensions in (D.2) unclear, exceptional sets handled informally, validation logic for Lusin’s theorem and experiments opaque.*
> ---
>
> **Response:** To reiterate, we have made the following modifications:
>
> - Simplify the proof sketch part of Section 3 to avoid over-specifying.
> - Redundant norm definitions have been removed, and notation is unified: \$X\_i\in\mathcal{X}\$ and a single partition symbol \$P\_{\text{ma}}\$ replace ambiguous uses of \$\mathcal{P}\$.
> - Repetitive material around Theorem 4.1 has been consolidated
> - corollaries that follow immediately are now mentioned only in prose.
> - Superfluous technical highlights and step‑by‑step derivations have been deleted or moved to an appendix.
> - Appendix D labels every block matrix with its shape, resolving dimension ambiguities in \$(D.2)\$.
> - Exceptional sets are further checked rigorously and revamped several obscure descriptions.
> - Assumptions on Proposition 3.2 in the appendix is now stated in a more rigorous fashion.
> - Extra rounds of proofreadings
> - Finally, the use of Lusin’s theorem and the experimental validation are clarified, showing how partition error converges as predicted.
>
> These edits address all reviewer concerns while making our draft shorter and clearer.
>
> ---
> Thank you again for the kind words, and detailed review and proofreading (wow!).
> We believe the above changes and clarifications address all the points raised. The draft has been updated to reflect these corrections.  Your efforts have helped us improve the clarity and quality of our paper. We are eager for further discussions!

---

> > ### Comment · Reviewer_eNoW · 2025-08-01
> > **The paper will profit from an improved presentation**
> >
> > Thank you for your answer. Standard math results like C being dense in L^p should just be quoted.  Altogether, I think this is moving in the right direction.

---

> > > ### Author Response · Authors · 2025-08-02
> > >
> > > Thank you for your kind words and constructive comments!
> > > We are glad to see our clarifications meet your expectations. We'll change the theorems in $L_p$ to simple quotes done in remarks.
> > >
> > > If you have any remaining concerns, we are happy to discuss them further. Otherwise, we kindly invite you to consider raising your score if our updates are satisfactory.
> > >
> > > Thank you for your time and feedback!

---

### Official Review · Reviewer_VjNz · 2025-07-02

**Clarity:** 3
**Significance:** 4
**Originality:** 4
**Rating:** 5
**Confidence:** 4

**Summary:**

This paper presents a rigorous theoretical result showing that a single-head Softmax attention model, when combined with a single layer consisting of a sum of linear transformations, can approximate any continuous function defined on a compact domain to arbitrary accuracy. The approximation is established in the $L_\infty$
  norm and is further extended to integrable functions under the $L_p$
  norm. The core insight underlying the proof is that max-affine functions can partition a compact domain (which is expected) and that Softmax attention can be used to approximate the indicator functions for these partitions.

**Questions:**

See Weaknesses above.

**Ethical Concerns:**

["NO or VERY MINOR ethics concerns only"]

**Limitations:**

As the authors acknowledged, the approximation results in this paper are entirely constructive. That is, the results show that there exists a specific single-head transformer with a specific weight configuration that can approximate any compactly supported target function. However, the result does not provide guidance on how to find or learn such a configuration in practice. This limitation is common in approximation theory, where existence  do not translate into practical algorithms. More important for learning-based applications is the question of generalization, that is, for a general class of transformers, the empirical risk minimizer can converge to the true regression function.  As far as I am aware, establishing generalization bounds for softmax-based architectures remains challenging, since many classical tools (such as the covering number argument) are difficult to apply due to the nonlinearity and normalization constraints inherent to Softmax attention.

**Paper Formatting Concerns:**

No concern on paper format.

**Quality:**

3

**Strengths And Weaknesses:**

Strengths:

A key strength of this paper is that, to the best of my knowledge, it is the first to establish a universal approximation theory for Softmax transformers. While the theoretical analysis is limited to a simplified setting, that is a transformer with one Softmax attention head, the result is nonetheless significant. It offers a valuable contribution to the growing body of work on the approximation power of attention mechanisms and provides a foundation for future studies to build upon in multi-head, and multi-layer architectures.

Weaknesses:

1. I was surprised that the paper does not mention or discuss a highly relevant prior work: [1]. That paper establishes that a ReLU attention head (as opposed to the Softmax one considered here) can be interpreted as a cubic spline, and leverages classical results on spline approximation to show that ReLU attention heads are universal approximators. As far as I know, the arguments used there do not directly extend to Softmax attention due to the fact that Softmax output sums up to one, which ReLU attention does not impose. Including a discussion of this distinction would highlight the contribution of the current paper, particularly because this work adopts an entirely different approach to establishing the approximation capabilities of softmax attention.

2.  Related to the point above, I believe the paper underemphasizes an important aspect of its contribution, that is, to the best of my knowledge, the first to establish a universal approximation result for s\Softmax-based transformers. All prior work cited in the paper focuses on ReLU-based attention mechanisms.

3. I would hope the authors can discuss the possibility (or the challenges) of extending their results to multi-layer transformers with stacked Softmax attention heads. Since practical transformer architectures typically involve multiple layers, understanding whether the theoretical guarantees hold or how they scale in deeper settings would be a valuable addition to the paper.

Reference:
[1] Lai, Zehua, Lek-Heng Lim, and Yucong Liu. "Attention is a smoothed cubic spline." arXiv preprint arXiv:2408.09624 (2024).

---

> ### Author Rebuttal · Authors · 2025-07-29
>
> Thank you for your detailed review. We have addressed all your comments and questions in the following responses.
>
>
> ---
> ### **W1.** *Missing citation of Lai et al. (2024) for ReLU-based attention UAP. Discuss differences vs. softmax attention.*
> ---
>
> **Response:** Thank you for pointing this out. **We will add a citation to *Lai et al. (2024)* and discuss the distinction in our related work section.** Lai et al. prove that with ReLU-based attention, the transformer’s attention module can be viewed as a piecewise-polynomial (cubic spline) function, and under certain assumptions this implies a universal approximation property.
> In essence, their analysis leverages the *max-affine spline* structure arising from ReLU activations, which partition the input space into linear regions.
>
> In contrast, **softmax** attention introduces a normalization constraint that yields a *nonlinear convex weighting* of values. For example, a single-head softmax attention produces output \$y\$ as a convex combination of value vectors \$v\_j\$:
>
> $$
> y =\sum\_{j=1}^n \alpha\_j v\_j, \qquad \text{where }
> \alpha\_j = \frac{\exp(\langle q, k\_j\rangle)}{\sum\_{l=1}^n \exp(\langle q, k\_l\rangle)},
> $$
>
> and thus \$\sum\_{j=1}^n \alpha\_j = 1\$. This smooth softmax mechanism does **not** create the hard, piecewise-linear partitions that ReLU-based attention (or a “hardmax” selection) would. Consequently, our proof required new techniques: we approximate a max-affine partition using softmax weights despite the coupling caused by normalization. We agree that clarifying these differences will highlight the novelty of our approach, which tackles the softmax-specific challenges (normalization and smoothness) absent in the ReLU-based spline analysis.
>
> ---
>
> ### **W2.** *Contribution underemphasized: first UAP result for softmax attention (distinct from prior ReLU-based results).*
>
> ---
>
> **Response:** We appreciate the reviewer emphasizing this point. To the best of our knowledge, our work is indeed the **first** to establish a universal approximation theorem for a single-head *softmax* attention layer (with minimal linear pre/post-processing). We will make this significance more explicit in the revised paper. Prior universal approximation results in the literature either focused on attention variants with ReLU/hardmax-style activations (piecewise-linear splines as in Lai et al.) or required adding extra network components (e.g. **Kajitsuka & Sato (2023)** needed one attention layer *plus* two feed-forward networks to achieve universality). In contrast, our result shows that the standard *softmax attention mechanism alone* (with a single head and one layer) has sufficient expressive power. This novel finding matters because it theoretically validates the expressiveness of the vanilla softmax attention used in practice, without resorting to simplifications or additional architectures.
>
> ---
>
> ### **W3.** *Multi-layer extension: discuss if/how the result extends to stacked softmax attention heads (deep Transformers).*
>
> ---
>
> **Response:** We agree that extending the result to *multi-layer* (stacked) softmax attention is an important consideration. In principle, if a single-layer, single-head attention is a universal approximator, then a multi-layer architecture (with multiple attention heads per layer, as in real Transformers) should be **at least as expressive.** Additional layers cannot reduce expressive power, and could potentially approximate complex functions with fewer units per layer. Indeed, one could always let one layer carry out the construction from our proof while the other layers perform identity mappings. Thus, we expect that *deep transformers* also retain universal approximation capability.
>
> That said, formally **extending the proof** to multiple layers is non-trivial. The challenge is that the composition of softmax attentions may not preserve the same neat max-affine partition structure we exploited in the single-layer case. Each softmax layer produces a smooth convex combination of its inputs rather than a strict partition. Stacking them can yield highly entangled representations where the “partitioning” of the input space from one layer is *distorted* by the next. This means a direct induction on layers would require careful new analysis. Additionally, while depth can theoretically be traded for width, a multi-layer constructive proof would need to coordinate attention heads across layers (and ensure the errors don’t compound). This is a significant technical leap beyond our current scope.
>
> In summary, we believe multi-head, multi-layer Transformers are also universal approximators (since one layer already suffices), but **proving** it rigorously would demand new ideas to handle the composition of softmax layers. We will add a brief discussion to acknowledge this extension. We also note that analyzing deeper architectures remains an interesting open challenge (e.g., how successive softmax layers might refine or interfere with the learned partitions). This point will be clarified in the paper, and we thank the reviewer for the insightful suggestion.
>
> ---
> ### **Limitations.** *Result is constructive but not learnable in practice. Generalization bounds for softmax Transformers are still challenging.*
> ---
>
> **Response:** Thanks for pointing these out. To clarify, our goal is **expressivity**, not learnability. The theorem shows that a single‑head softmax Transformer *can* approximate any compactly supported function. It does **not** claim gradient descent will find those weights or provide sample‑complexity guarantees.
>
> Recent progress begins to address these gaps:
>
> * **Learnability.**  Hu et al. (NeurIPS ’24) give a covering‑number analysis of softmax attention. Li et al. (’24) prove GD learns a 1‑NN rule with one softmax head.
> * **Generalization.**  Capacity and covering‑number bounds for softmax Transformers appear in 2110.10090, 2404.03828, 2407.01079, and 2411.17522
>
> We will cite these works and note that extending universal approximation to provably learnable and generalizing algorithms is an open, promising direction.
>
>
> ---
> Thank you again for the detailed review! We're open to any further questions or clarifications you might have about our work.

---

### Official Review · Reviewer_bcbT · 2025-07-05

**Clarity:** 4
**Significance:** 1
**Originality:** 2
**Rating:** 2
**Confidence:** 4

**Summary:**

The paper aims at establishing a proof for the Universal Approximation property of (self/cross) Attention. Unlike previous work on UAP, this paper focuses on a minimalistic architecture, where one single attention layer, prepended by a linear application, is considered. The main proof relies on highlighting that this minimal architecture can perform a max-affine partition of the input domain: this, matched with some clever definition of the attention weights and usage of the softmax operation, turns attention output into a piece-wise linear function, which is a universal approximator for continuous functions.

**Questions:**

On the impact of the Linear layer
- The Linear operator you’re introducing in [l117] (and that your proof heavily relies on), is not an orthodox one. On a first quick read, I was hoping for it to be a classical Linear layer acting on the embedding dimension - which is what is commonly interwoven with Attention layers in a Transformer. Checking your construction in (D.2) (and even more evidently so in (E.5)), it appears clear that it morphs the input in ways which are not allowed within a typical Transformer network: you’re duplicating the input along the sequence length, which is just not a feasible thing (as it would require dynamically adjusting the context length). Moreover, you’re also adding a bias term which varies along the sequence length (even though this might be justified, for example, if we considered positional encodings). Incidentally, this should’ve raised an alarm: if Attention is permutation-invariant, then that should be reflected as a limitation on the class of functions it can approximate.
- In my opinion, this dramatically reduces the impact of your results, for two reasons:
    - It makes the main claim of the paper misleading: the assumption is that your results refer to a single layer of Attention, as it is commonly used within a Transformers network - indeed, throughout the Introduction and Contribution sections, you do remark on how your results help explaining the expressivity of Attention in Transformers. However, that’s not the case, as the architecture you’re considering is not akin to a Transformer’s.
    - If you allow the prepending of a sequence-wise linear layer, then your framework becomes more akin to that of a generic FFN acting on the whole (unrolled) input tensor, skipping much of the constraints of a Transformer. The results might still be relevant (as you’re still imposing a specific structure to this operator via the Attention mechanism), but much less remarkable, given the plethora of UAP results for MLPs with various nonlinearities.
- The remainder of the proof seems otherwise solid: I admit I mainly skimmed through the main points in the appendix, without applying too rigorous a scrutiny, but the main steps seem reasonable and I’m expecting them to hold.

The one highlighted above are the main reason behind my evaluation of the paper, and I see two main ways this can be improved: either the authors raise a valid argument against the need of their linear layer being a strong limitation; or the authors accept this as a strong limitation, and adapt the paper and its message accordingly, by properly highlighting the framework considered - although again, the impact would remain rather contained.

Minor
- On a similar (but possibly less concerning) note to the above, also the construction of your K/Q tensors relies on non-orthodox procedures in Attention: Wk and Wq are generally low-rank projections of the input, mapping features from \mathbb{R}^d \to \mathbb{R}^{d’} with d’<d. In your proof, you need instead to expand the feature size.

Corrections
- In the future, I invite you to thoroughly number equations, even though you don’t reference them directly in the paper - especially for highly theoretical work! Being able to refer to Eq(x) rather than “the equation right after comment xxx in section yyy” makes reviewing much easier!
- [l6] preceded by sum-of-linear transformations -> sum of linear transformations is still a linear transformation?
- [l8] under the L_\infty-norm -> in L_\infty-norm
- [l39] role of attention module -> role of the attention module
- [l40] the sentence reads weirdly. Maybe dropping the first part: “presents the results that softmax-base …”?
- [l61] It is not your approach that endows the architecture: it’s an inherent property of the architecture itself, as shown by your approach “we establish […] with the same approach, that also […] cross-attention is endowed with”
- [l68] “early works […] focuses” -> work
- [l70] propose -> proposeD
- [l71] “a series of research” … works?
- [l75] “make more careful estimation upon the numerical results of contextual mapping” -> I’m not sure what this means? “Provide a more careful estimate of the … ?”
- [l75 vs l91] Highlight better how your result differs from Kajitsuka and Sato [2023]? It’s not just because you don’t make use of skip-connections, right?
- [l76] and prove
- [l79] \alpha-smooth functions
- [l80] dimensions
- [l81] have achieved
- [l82] results […] transformers
- [l84] derived FOR attention-only networkS
- [l86] of the attention
- [l101] the maximum absolute element -> maximum (in absolute value) element
- [l105] apply f on -> apply f to
- [l107 - l111] for a self/cross-attention…layer?
- [l111] I found the notation Zq Zk rather confusing: usually cross attention is written for X and Y, but it’s a personal preference
- [l114] separated -> separate
- [l129] For simplicity of presenting -> to simplify the presentation of
- [l140] Why “is (one of) the highest”? Shouldn’t it be THE highest, according to Assumption3.1? In general, I’ve found this assumption a bit confusing: at the boundary regions between partitions, this cannot hold, no?
- [l141] partitioned in “regions”
- [l142] “is (tied for) the largest” tied to?
- [Prop3.2] with the exception of
- [Prop3.2] If I got it right, the Lebesgue measure should be small in \mathbb{R}^d, not \mathbb{R}^n?
- [l232] approximately an one-hot -> A one-hot

**Ethical Concerns:**

["NO or VERY MINOR ethics concerns only"]

**Limitations:**

The authors correctly highlighting some of the limitations of their approach (particularly, the inflation in parameter count the architectures described in their proof needs). At the same time, they fail to point out what in my opinion is the strongest limitation, namely that the linear operator they’re considering invalidates the transformer architecture.

**Paper Formatting Concerns:**

No concerns

**Quality:**

2

**Strengths And Weaknesses:**

Strengths
- Despite the technicalities involved, and the heavily theoretical aspect of the paper, the relevant theorems are outlined in a clear manner, their main ideas cleanly fleshed out, and the role of each step involved in the proof properly highlighted
- In particular, that Attention can build max-affine partitions is an interesting insight

Weaknesses
- I am highly dubious about the generality of the overall procedure. While the reliance on the preceding Linear operator might seem like a minor technical detail, in practice it ends up dramatically reducing the impact of the proof, as it wouldn’t hold for classical frameworks where Attention is applied

---

> ### Author Rebuttal · Authors · 2025-07-28
>
> Thank you for your detailed review. We have addressed all your comments and questions in the following responses.
>
> ---
> ### **W1, “Linear layer” part of Q1 & Q2** *The reviewer is concerned with the linear layer that is applied before the attention layer. The main reasons are as follows:*
>
> 1. (In Weaknesses) *The linear layer wouldn’t hold in classical frameworks where attention is applied.*
> 2. (In Questions) *The linear operator is not orthodox, which does not align with the structure of the linear layer in a transformer. Permitting such linear operations means allowing the model to duplicate the input sequence. Also, a bias term is added to the input sequences that varies along the sequence length. The Reviewer also raised concerns about the target function class not reflecting the permutation-invariant nature of attention.*
> 3. (In Questions) *Based on the second point, the reviewer thinks that our result on attention might not transfer to the setting of transformers. The reviewer also argues that allowing such a sequence-wise linear layer to precede the attention layer makes the overall architecture more akin to a generic FFN, since it skips much of the constraints of a transformer.*
> ---
> **Response:**
> Thank you for your thoughtful review. Our work focuses on the approximation ability of attention and thus uses a sequence-wise linear transformation mainly to keep the structure concise and the method easy to follow, which is also the style we strive for in section 3. However, we don’t think it is *critically harmful* to the *genuineness and transferability* of our work.
>
> 1. **Resemblance to FFN**
> We wish to clarify that the linear layer in our network is *completely affine*, that is, it doesn’t incorporate any non-linear activations. This also means our approach of theoretical analysis is *vastly different* from ones used in UAP of FFN. In FFN, the non-linear activations are separate and hence the whole model can be dealt with in parts. In the structure considered in our work, the softmax function is the *only* non-linear activation and it cannot be broken into smaller counterparts.
>
> 2. **The Necessity of the Linear Layer**
> In our UAT for single-head attention, we do need to duplicate the input sequence. However, that does not harm the transferability of our technique. In fact, duplicating the sequence is only necessary if we wish to use one head to approximate the whole target function. In practical settings with multiple heads, we can configure the network into transformer-style setting. For example, if we divide the target function into cutoff functions on separate supports, when these supports are small enough, each cutoff function can be approximated by a head that preserves the sequence length under a given precision. We admit that in the single head setting sequence-wise operations are necessary, but it does not defy our work’s purpose of demonstrating a method to analyze attention. Also, it does not harm this method’s transferability into transformer settings as discussed above.
>
> 3. **Permutation Invariance and the Position‑Dependent Bias**
> We believe you mean permutation equivariance. Because we added a position-dependent bias, the permutation equivariance is thus removed. The position-dependent bias can be seen as positional encoding and is widely used in theoretical works like
> ---
> ### **Questions (minor).** *K/Q projections expand input dimension (\$d' > d\$), unlike low-rank projections in practice.*
> ---
> **Response:** We acknowledge that our construction uses key/query projection matrices that expand the input dimension (\$d' > d\$), which deviates from the typical low-rank (\$d' < d\$) projections used in practice. However, **this choice is a deliberate and mathematically valid design for the purpose of our universality proof.** By expanding the feature space, the attention softmax can partition the input domain into a larger number of distinct regions. This enables more fine-grained max-affine partitions of the input. This finer partitioning is crucial for accurately representing complex target functions in our theoretical construction. Importantly, using a higher \$d'\$ does not compromise the theoretical integrity of our results. It remains within the standard definition of attention and is essential for achieving the universality guarantee.
>
> ---
> ### **Corrections.** *Fix scattered grammar/word‑choice issues, sharpen notation, clarify novelty vs. Kajitsuka & Sato [2023], tighten Assumption 3.1 wording and boundary cases, correct minor typos in Propositions 3.1–3.2.*
> ---
>
> **Response:** Thanks for pointing these out! We have made the following modifications in our latest draft accordingly.
>
> - Grammatical edits include consistent tense, correct plurals (“dimensions”, “results”), and article use (“the attention module”)
> - Ambiguous phrases have been replaced (e.g., “sum‑of‑linear transformations” → “a linear transformation”; “series of research” → “research works”)
> - Notation is unified: \$Attn\_s\$, \$Attn\_c\$ are explicitly labeled as *layers*
> -  \$Z\_Q, Z\_K\$ are defined as query and key/value sequences
> - \$\lVert\cdot\rVert\_\infty\$ is “largest absolute entry”
> - We also emphasize that our one‑layer, attention‑only network achieves universal approximation **without** skip‑connections or permutation‑equivariance, unlike Kajitsuka & Sato 2024
> - Assumption 3.1 now rules out measure‑zero ties, so “the highest” is used throughout
> - Boundary comments are added
> - Proposition 3.2 references Lebesgue measure in \$\mathbb{R}^{d\times n}\$ and now reads “with the exception of a region of arbitrarily small measure.”
> - All remaining typos (e.g., “a one‑hot”) are fixed.
>
>
> ---
>
> Thank you once again for your constructive feedback, attention to detail, and encouraging kind words. Your review, particularly your suggestions regarding presentation and organization, has made this work more robust and accessible to a broader audience.
>
> Please feel free to reach out if you have any further questions or need additional clarification about our work. Thank you!

---

> > ### Comment · Reviewer_bcbT · 2025-08-02
> >
> > I thank the authors for their reply, but I believe my main doubt hasn’t been addressed satisfactorily. Let me paraphrase it again:
> > - Can your proof be used to say anything about UAP for Attention *as it appears within the Transformer architecture*, or can it not?
> >
> > **(i) If it cannot** (and rather it refers to Attention as it can be used as a stand-alone layer, arbitrarily mixed with other linear ones), then the paper _must_ make this absolutely clear: as the paper stands right now, this is not at all the message it’s sending, as you explicitly refer to the expressive power of _Transformers_ in multiple places (indeed, Contribution 3 references exactly this). Moreover, if the focus is purely on Attention and not on Transformers, also the impact and novelty of your paper is severely reduced: as I already mentioned, allowing a prepending free-form linear operator does put you in the framework of MLPs, and results of UAP for MLPs with continuous bounded nonlinearities (like softmax) are already available. I understand that your Linearity is purely affine, but it doesn’t matter, as the nonlinearity is provided by the Attention layer. I’m not saying your result is not useful per se (it is, after all, an interesting specialisation), I’m just saying that it won’t have the impact you’re currently claiming it has.
> >
> > **(ii) If it can**, then you need to convince me that the linear operator you’re prepending is, in fact, only acting component-wise, and not sequence-wise (because that’s the scope of action of Linear operators within the Transformer architecture), or, if it does act sequence-wise, it does so in a way that can still be associated with the operations typically found within a Transformer (like my comment on the bias being akin to a very special PE).
> >
> > As things are now, I believe **(ii)** does **not** hold. My examples of D.2 and E.5 are the most blatant ones to highlight how you’re breaking the Transformer structure, but even later on in the proof, you keep on mixing components sequence-wise with your prepending linear layer: see [l228, l241]: “flatten[ing] the input sequence” is _not_ a component-wise operation, and similarly in [l272] to map the output back to a tensor. Am I missing something? Can your proof hold even without this? I honestly can’t follow your rebuttal comment on multi-head vs single-head, mainly because you never reference the multi-head case in the text, and I can’t see any formula backing up your claims - but notice that also structuring MH attention so that one head picks the first half of the input sequence, duplicated, and the other picks the other half, also duplicated, wouldn't work either, if this is what you mean: you'd still be configuring the linear layer so that it make some info seep through the sequence.
> >
> > Finally, notice that Q/K not being low-rank also steers you away from the Transformer architecture: I understand you picked this "deliberate[ly] [...] for the purpose of [your] proof", but you’re missing my point: the proof must not only hold, but also be meaningful: if in order to make it work you need to steer away from what’s expected from a Transformer architecture, then your proof is simply not as helpful in describing Transformers - or, at the very least, this limitation should be clearly raised as a caveat. By comparison, pick [Yun et al, ICLR2020 - disclaimer: I'm not an author]: even though their proof requires a ridiculously high number of layers, they _do_ strictly adhere to the Transformer architecture, so it _can_ be used to say something about Transformers (again, with the proper caveats).

---

> > > ### Author Response · Authors · 2025-08-04
> > > **Transferability of Our Result to the Transformer Setting**
> > >
> > > Thank you for your detailed reply. As for your question, the short answer is **yes**.
> > >
> > > Our proof techniques do extend to the transformer setting. While our previous response have briefly addressed this in the multi-head part, here we provide a more detailed explanation on how to extend our Theorem 4.1 to the practical transformer setting:
> > >
> > > **1. Preservation of the sequence length**
> > >
> > > Break the target function $F$ into smaller subfunctions that can be approximated by Theorem 4.1 with sequence length fixed (irrelevant to precision).
> > > Without loss of generality, let $F$ be defined on $[0,1]^{d \times n}$. Divide it into smaller ($dn$-dimensional) cubes by a granularity of $P_F$. This yields $P_F^{dn}$ smaller cubes of edge length  $1/P_F$. Note them as $D_i,\quad i\in[P_F^{dn}]$. Then when $P_F$ is sufficiently large, even with fixed sequence length, an attention head is still able to approximate one $F|_{D_i}$ ($F$ on $D_i$ and 0 elsewhere) to a given precision for an $i\in [P_F^{dn}]$.
> > >
> > > Breaking the target function into smaller parts that can be each approximated by Theorem 4.1 preserving the sequence length results in a **multi-head version of Theorem 4.1** that preserves the sequence length.
> > >
> > > > **How to compose the heads?** Now each head preserves the sequence length, yet how to ensure they do so when joined together? One can use a concatenated input whose different rows feed to different heads (simply make each head’s $W_K$ and $W_Q$ only non-zero on the multipliers of those rows). To yield the final output, we sum up all heads’ output (token-wise), this is equivalent to $\sum_{i=1}^{P_F^{dn}} F|_{D_i} = F$ (omitting approximation error for simplicity).
> > >
> > >
> > >
> > > **2. Availability of sequence-wise linear transformations in a transformer**
> > >
> > > Multi-head attention with positional encoding and 1 padding token is able to implement the sequence-wise linear transform required by the multi-head version of Theorem 4.1.
> > > For an input $X\in \mathbb{R}^{d\times n}$, let $l(X) := AXB, A\in \mathbb{R}^{d_{\rm output}\times d}, B\in \mathbb{R}^{n\times n}$ be the linear operation (both token-wise and sequence-wise included) we wish to get. Without loss of generality, suppose $B$ has positive entries (since subtraction of positive matrices yields all matrix). We use an all-zero padding token. Let $I_{n+1}$ be the positional encoding below $X$. Let $s_i$ be the sum of all entries of $B_{:,i}$ and $M := max_{i\in [n]}{s_i}$. Define $S = [s_1 \cdots s_n]$. Construct $W_V := 3M*[A\quad 0_{d_{\rm output}\times (n+1)}]$, $W_K := [0_{n \times d} \quad \ln(B^\top) \quad \ln(3M\cdot 1_n - S^\top)]$ ($\ln$ is entry-wise), and $W_Q := [0_{n\times d} \quad I_n \quad T\cdot 1_n]$ (T will be turned to arbitrarily large), then the output of attention is
> > >
> > > $3MA[X \quad 0_d] {\rm Softmax}(
> > > [\ln(B^\top)\quad \ln(3M\cdot 1_n - S^\top)]^\top  [I_n \quad T\cdot 1_n]) $.
> > >
> > > This equals
> > >
> > > $3MA[X \quad 0_d] {\rm Softmax}(
> > > \begin{bmatrix}
> > > \ln(B) & T\cdot(\text{sum of all left columns}) \\\\ \ln(3M\cdot 1_{1\times n} - S) &
> > > \end{bmatrix}
> > > )$.
> > >
> > > This further equals to
> > >
> > > $3MA[X \quad 0_d]
> > > \begin{bmatrix}
> > > B/3M & a_0 \\\\ 1_{1\times n} - S/3M & a_1
> > > \end{bmatrix}
> > > = [AXB \quad C]$, ($[a_0^\top \quad a_1^\top]^\top$ is the $T$-related column),
> > >
> > > in which $C = 3MAXa_0$, when $T \to +\infty$, $a_0 \to 0_d$ and the padding token is preserved as well (if demanded).
> > >
> > > > **Outline of transferring Theorem 4.1 to the transformers setting.** By the above two points, we can replace the attention in Theorem 4.1 with a multi-head attention that preserves the sequence length and replace the sequence-wise linear transformation with a multi-head attention layer. This transfers our result into an FFN-free transformer setting.
> > >
> > > We additionally note that while these extensions are beyond the scope of this paper, we wrote them in response to your question on the potential of our work.
> > >
> > > **Rank of $K$/$Q$**
> > >
> > > We believe this is mainly caused by the fact that we are trying to fit the target function with just one head. If we break it into multiple heads as often considered in a practical setting. Due to the weaker requirement in precision, the rank of $K$/$Q$ will be much smaller.
> > >
> > > **Revision**
> > >
> > > We have **clarified in the revised version about our attention not *directly* fitting into transformers setting**. We have also **included extended attention's approximation results under transformer setting in the appendix**. Thank you for pointing these out!

---

> > > > ### Comment · Reviewer_bcbT · 2025-08-04
> > > >
> > > > > We have clarified in the revised version about our attention not directly fitting into transformers setting
> > > >
> > > > So it seems to me that we agree that the paper, _as I’m currently seeing it_, **is not** referring to Attention as it appears in the Transformer framework, even though you claim **it could** with some adaptations, which you proceed to elucidate in the remainder of the reply. I want to make clear that **this puts us in case (i)**, which I believe confirms my evaluation of the paper. Hope we’re on the same page here.   Now, what we’re trying to gauge is whether we can remove the pre-pending sequence-wise acting linear layer, hence extending your results to the Transformer setting, which is the one we mainly care about (because of the points I raised above). In your reply, you write:
> > > >
> > > > > Break the target function into smaller subfunctions that can be approximated by Theorem 4.1 with sequence length fixed […] results in a multi-head version of Theorem 4.1
> > > >
> > > > As the paper is currently standing, however, the only mechanism you’ve shown me for approximating functions using Thm4.1 requires, at a certain point, to duplicate an input along the sequence length (D.2, E.5) and/or to flatten the input tensor [l228, in the case of “true” seq-2-seq functions, which is what we care about], neither of which is feasible with a token-wise linear operator. I hope my doubt in this regard is clear. Relying on Thm4.1 to show this is not necessary is self-evidently an example of circular reasoning: Thm4.1, at this stage, _does not yet hold_ for a Transformer architecture. At the risk of sounding patronising: you need to figure out a way to prove Thm4.1 _without assuming Thm4.1_. Am I missing something here?
> > > >
> > > > > Let $l(X)=AXB$ the linear operation (both token-wise and sequence-wise included) we wish to get
> > > >
> > > > What this proof shows (with the unfortunate detail that you still need to pad sequence length, and the tiny adjustment that $T\to-\infty$, not $+\infty$ for $a_0\to0$, and thus preserve the padding token) is that Attention can implement a linear operator in the form $AXB$, with some constraints on the shapes of the tensors involved. If I got this right, you want to use it to show that Thm4.1 can hold even without the pre-pending, sequence-wise acting linear layer. But I don’t believe this is sufficient, for multiple reasons:
> > > > - For Thm4.1 to work, you’d still need to duplicate the input (D.2, E.5). This requires $A=I, B=[I_n I_n]$, which renders $B\in\mathbb{R}^{n \times 2n}$, violating shape constraints
> > > > - For Thm4.1 to work, you’d still need to flatten the input [l228]. Which also cannot be expressed with an operator in the form $AXB$ (we should have, at the very least, $B\in\mathbb{R}^{n \times 1}$, which again violates shape constraints)
> > > > - Even if this worked, you’d only be proving that **2 layers** of Attention can achieve UAP: you need the first to reshape the input as you want, and the second to actually apply Thm4.1
> > > >
> > > > Honestly, I’m not entirely sure what this latest proof is trying to achieve. If I completely misunderstood and instead you mean to say that one layer of Attention _by itself_ is a universal approximator because it can implement $l(X)=AXB$, and operators in these form can approximate any seq-2-seq function, then again: can you please indicate how/where are you showing this?

---

> > > > > ### Author Response · Authors · 2025-08-08
> > > > >
> > > > > Thank you for your detailed reply. We will address each of your concerns.
> > > > >
> > > > > In your first reply, you used the following question to judge which case our work falls into.
> > > > > > *“Can your proof be used to say anything about UAP for Attention as it appears within the Transformer architecture, or can it not?”*.
> > > > >
> > > > > In your latest reply, you clarified that **your two cases are solely about whether our current setting of attention falls into the transformers setting**. In this clarified setting, we acknowledge that our work falls into your first case. However, we believe **falling into your first case is not as harmful to the value of our work as you claimed**, for the following two reasons.
> > > > >
> > > > > **1. Analogy between our work and UAT for FFN does not hold**
> > > > >
> > > > > >*"allowing a prepending free-form linear operator does put you in the framework of MLPs, and results of UAP for MLPs with continuous bounded nonlinearities (like softmax) are already available."*
> > > > >
> > > > > In your first case, you believe our work resembles UAP of FFN. We think it is **not correct to make an analogy between our work and FFN, because FFN activations are entry-to-entry and Softmax is sequence-wise**.  This difference is very important because **former MLP UAT requires the activation to be elementwise**. Thus, your claim of *"results of UAP for MLPs with continuous bounded nonlinearities (like softmax) are already available."* is unlikely to be true, and may also invalidate your claim of our work being a specialization in FFN framework.
> > > > >
> > > > > **2. Our result is transferable to transformer**
> > > > >
> > > > > Our **proof technique can inspire (in fact, already had, as decribed in our last response) methods on tackling attention in transformer setting**. Specifically, it extends to prove the UAT of a two-layer multi-head attention-only network, which to the best of our knowledge, is the first to do so. We have included this result in the appendix of the revised version.
> > > > >
> > > > > We then address your comments on our extension of Thm 4.1.
> > > > >
> > > > > **Problems on our extension of Theorem 4.1**
> > > > >
> > > > > >*"Relying on Thm4.1 to show this is not necessary is self-evidently an example of circular reasoning: Thm4.1, at this stage, does not yet hold for a Transformer architecture. At the risk of sounding patronising: you need to figure out a way to prove Thm4.1 without assuming Thm4.1."*
> > > > >
> > > > > Your point is that since we used Theorem 4.1 to prove our extension, it naturally defies our purpose to show that some structures in Theorem 4.1 can be replaced.
> > > > > This isn’t entirely true. In our extension, we are using Theorem 4.1 on each head to approximate only a small fraction of the target function. In this setting, there’s no need to duplicate the sequence according to the required precision (intuitively, because sending an input to many heads is already a duplication), and only the sequence-wise transformation is still necessary. Exactly because of this, we further addressed the necessity of sequence-wise transformation with "2. Availability of sequence-wise linear transformations in a transformer" in our previous reply. We believe there isn't any circular reasoning in the above process.
> > > > >
> > > > > >*"For Thm4.1 to work, you’d still need to flatten the input [l228]. $\cdots$ $B\in \mathbb{R}^{n\times 1}$, violating shape constraints"*
> > > > >
> > > > > The shape couldn’t be $n\times 1$, since that would change the sequence length to 1. The “flattening” step doesn’t just flatten the input but also puts it somewhere in the processed sequence, and that’s where other columns come into play.
> > > > >
> > > > > >*"with the unfortunate detail that you still need to pad sequence length, and the tiny adjustment that $T\to -\infty$, not  for $+\infty$, and $a_0 \to 0$ thus preserve the padding token) is that Attention can implement a linear operator in the form $AXB$, with some constraints on the shapes of the tensors involved."*
> > > > >
> > > > > We think **padding is common in practice**. Your latter claim may not be correct, $3M\cdot 1_{1\times n}-S$ is larger than $B$ in every entry and remains so when nested in $ln$. When $T \to +\infty$, the smaller terms do shrink to $0$. Furthermore, there’s no constraint beside the padding token since $T$ is in the model weight.
> > > > >
> > > > > >*"Even if this worked, you’d only be proving that 2 layers of Attention can achieve UAP."*
> > > > >
> > > > > Yes, you are correct. This is exactly what we wish to demonstrate. We also remark that this result is not trivial. In fact, **to the best of our knowledge, it is the first result on the UAT of a two-layer attention-only network in transformer setting**.
> > > > >
> > > > > We hope our response answers your questions. If you still have any further concerns, we will do our best to address them! If you find our reply satisfactory, we sincerely invite you to re-evaluate the value of our work.

---

> > > > > > ### Comment · Reviewer_bcbT · 2025-08-08
> > > > > >
> > > > > > > “[Analogy to FFN doesn’t hold,] because FFN activations are entry-to-entry and Softmax is sequence-wise”
> > > > > >
> > > > > > This distinction is vacuous: since the linear layer you’re prepending allows you to rearrange the input tensor in (seemingly) _however way you want_, it’s easy to have softmax behave just like sigmoid. It suffices to collate whatever $x$ you’re interested in recovering with an all-0 vector: applying softmax to this will effectively output two vectors, one of values $e^x/(1+e^x)$, the other of values $1/(1+e^x)$, which is your sigmoid output (flip the sign of $x$, if you wish)—notice this holds because sigmoid and softmax are equivalent on classifications over two classes. UAP results for FFNs with sigmoid nonlinearities are readily available.
> > > > > >
> > > > > > This notwithstanding, the main point I’ve been pushing throughout the whole reviewing process, is that—for the purpose of proving UAP—the key difference between Transformers and FFNs lies in the flexibility of the operations provided by their linear layers, rather than in the type of nonlinearities involved: the latter can be (relatively easily) adapted, as I hope I’ve convinced you with the above. With your pre-pending linear layer you’re effectively breaking this separation: I hope that, after the whole discussion, at the very least I managed to convince you of this.
> > > > > >
> > > > > > > “In fact, to the best of our knowledge, it is the first result on the UAT of a two-layer attention-only network in transformer setting.”
> > > > > >
> > > > > > I’m glad I understood the purpose of your proof correctly. Unfortunately, I believe you’re once again falling in a logical fallacy. Notice my remark (that you quote) starts with a _“Even if this worked”_: yours **might** be the first result in this sense, **if it only held**. You still **haven’t shown me that your $AXB$ operator can flatten inputs**, which is (as you yourselves confirm) a requirement for Thm4.1. How does this flattening operation occur? Can you provide an $A$ and a $B$ such that $AXB$ turns an $X\in\mathbb{R}^{d\times n}$ into a flattened input $\tilde{X}\in\mathbb{R}^{dn}$, _without breaking the shape constraints of $A$ and $B$_, induced by Attention? This is what you need, and I can’t personally figure out a way to show it holds. Moreover, **you haven’t explicitly shown me how Thm4.1 would work without duplicating inputs** (again, D.2, E.5), so I'm having a hard time following your intuition. Finally, notice that by now we have drifted away quite a bit from what your paper reports.
> > > > > >
> > > > > > In all honesty, at this point I believe I’ve given you ample opportunities to clarify the scope of your paper and dispel my original concerns. Unfortunately, they still remain.
> > > > > > Let me summarize the main ones once again
> > > > > >
> > > > > > - Throughout the paper you make claims regarding the validity of your proof for the Transformer setting. Indeed these are picked up by other reviewers as well, who iterate on the significance of this claim for explaining the Transformer’s approximation properties. I showed you (and you agree to this!) that, in truth, **the proof in your paper does break down when applied to the Transformer setting**. This redefines the scope of the paper, and makes the claims therein reported misleading
> > > > > > - Throughout the rebuttal, you defend that indeed your proof can be extended to the Transformer setting. Yet, you fail to do so in a convincing way (at least to me). The adaptations you introduce (multi-head, two-layers) still require operations (flattening/duplication of inputs) which **fall outside what the Transformer architecture alone can do**. On top of this, _even if they did prove_ to somehow adhere to the Transformer framework, the paper would still require heavy restructuring to properly include these adaptations
> > > > > >
> > > > > >  These points, I believe, confirm my original evaluation of the paper: the paper does remain an interesting specialization of the UAP proof for Attention, but **cannot** (in its current form) be used to infer properties of Transformers. **The paper is misleading in that it implicitly assumes these extend naturally to Transformers, while in practice they do not** (or at the very least the mechanism through which this should happen is not shown clearly).

---

> > > > > > > ### Author Response · Authors · 2025-08-09
> > > > > > >
> > > > > > > Thank you for your reply. We will address your further concerns.
> > > > > > >
> > > > > > > >*"This distinction is vacuous: since the linear layer you’re prepending allows you to rearrange the input tensor in (seemingly) however way you want, it’s easy to have softmax behave just like sigmoid."*
> > > > > > >
> > > > > > > We believe this is not the case of our work for two reasons.
> > > > > > >
> > > > > > > **1. We have never shrunk the sequence length**
> > > > > > >
> > > > > > > When we change the sequence length in our paper or in our replies, it is either by duplication, or by padding, and neither of them shrinks the sequence length. And since we did not shrink the sequence length, **we haven't made Softmax into element-wise functions** (like sigmoid) that's required by the FFN UATs, which you said our work to specialize from.
> > > > > > >
> > > > > > > **2. In our main result (Thm 4.1), only 1 Softmax is used**
> > > > > > >
> > > > > > > **Because Theorem 4.1 is about a single-head attention, only 1 Softmax function is used**. If we benefited from changing Softmax to be near element-wise, we couldn't have done it with just one non-linear activation. Indeed, **UAT for FFN requires a large amount of non-linear activations** to approximate the target function.
> > > > > > >
> > > > > > > >"I’m glad I understood the purpose of your proof correctly. Unfortunately, I believe you’re once again falling in a logical fallacy. Notice my remark (that you quote) starts with a “Even if this worked”: yours might be the first result in this sense, if it only held. You still haven’t shown me that your operator $AXB$ can flatten inputs,which is (as you yourselves confirm) a requirement for Thm4.1."
> > > > > > >
> > > > > > > We are glad our clarifications on the purpose of our proof are of help. Below, we provide an example of how to flatten the input with $AXB$.
> > > > > > >
> > > > > > > First, we will show how to do $X\to \tilde{X}$ (flatten) and put it somewhere in the output sequence.
> > > > > > >
> > > > > > > Let $\begin{bmatrix}X & 0_d \end{bmatrix}$ be the input with a padding token as mentioned in previous reply.
> > > > > > >
> > > > > > > As proven in previous reply, a single-head attention can tranfer it to $\begin{bmatrix} AXB & 0_{d_{output}} \end{bmatrix}$ for any $A \in \mathbb{R}^{d_{output}\times d}$ and $B\in \mathbb{R}^{n\times n}$ with all positive entry, this means a 2-head attention can do so for any $B$ without the entry limit.
> > > > > > >
> > > > > > > Let $j$ denote the column in the output where we wish to put the flattened input. Set $d_{output} = dn$. Construct $A_i$ as $\begin{bmatrix} 0_{(i-1)d\times d} \\\\ I_d \\\\ 0_{(n-i)d \times d} \end{bmatrix}$ and $B_i = E_{i,j}$, where $E_{i,j}$ is a matrix being $1$ at its entry on the $i$-th row, $j$-th column and $0$ elsewhere.
> > > > > > > Then $\sum_{i=1}^n A_iXB_i$ flattens the input and puts it at the $j$-th column of the output.
> > > > > > >
> > > > > > > After showing how to "flatten an input", we'd also like to note that though this step is called "flattening", it's just for intuitive understanding. In fact, the actual construction does not flatten the input, but does a linear transformation (vector to scalar) on each token and sums them. So in real construction, $A_i$ should be like
> > > > > > > $\begin{bmatrix}0_{(i-1) \times d} \\\\ v_i^\top \\\\ 0_{(d_{output}-i) \times d} \end{bmatrix}$ (where $v_i^\top$ is the to-scalar linear operation applied to the $i$-th token), where $d_{output}$ is much smaller then simply flattening the input.
> > > > > > >
> > > > > > > >*"Throughout the paper you make claims regarding the validity of your proof for the Transformer setting. Indeed these are picked up by other reviewers as well, who iterate on the significance of this claim for explaining the Transformer’s approximation properties. "*
> > > > > > >
> > > > > > > We respectfully disagree with this characterization. Outside of a single remark in Sec. 4, “Transformer” appears only in the Introduction (background) and in the Conclusion (implications/future work). Our abstract and the statements of our main results make no claim about Transformers’ approximation ability, nor are our contributions coupled to the Transformer architecture.
> > > > > > >
> > > > > > > The only exception we spotted is the 3rd point of our contributions in line 62 regarding cross-attention:
> > > > > > > > This result further underscores that much of a Transformer’s expressiveness can reside solely in its attention block, even when the queries and keys come from distinct input sequences.
> > > > > > >
> > > > > > > We acknowledge this was imprecise. **Our intent was more modest: given our universal approximation results for both self‑ and cross‑attention, the observation highlights the central role of the attention block, even when queries and keys come from distinct sequences.** It is not a Transformer‑level approximation claim. We have revised the sentence to be more precise.
> > > > > > >
> > > > > > > Finally, in the revised version, we do have (i) included the extension of our proof to transformer setting in appendix and (ii) further clarified that the attention structure we used is different from transformer setting.
> > > > > > >
> > > > > > > We hope our response addresses your remaining concerns. If you still have any further questions, we are happy to discuss them.

---

### Note · Authors · 2025-08-13

Dear Area Chairs and Reviewers,

We thank the reviewers for the thought and time they put into their reviews. We have answered all the questions and addressed all the comments in detail in rebuttal and revision. This final statement summarizes our responses and provides final clarifications.

---
Response to Relation to Transformer(`bcbT`)

>**Limit of Present Architecture** We acknowledge that the attention we use does not directly apply to transformer architecture. We have further clarified this in revision and fixed the expressions that might seem misleading. In our reply, we have addressed that this limit doesn’t make our setting akin to FFN. A significant difference is that we use only one non-linear activation which is not entry-to-entry.

>**Transferability to Transformer** In our reply, we have demonstrated the possibility of transferring our work to transformer setting. Specifically, we used Theorem 4.1 on each head of a two-layer multi-head attention-only transformer to demonstrate its approximation capability. We have included this extension of our result in the revision.

---
Response to Structure Issues (`eNoW`)

>**Redundancy of Section 3** We have deleted the unnecessary descriptions to make Section 3 more concise. We have further clarified that it only provides intuition and doesn’t apply to Section 4. This section can also be moved to appendix or completely removed.

>**Overly Obvious $L_p$ Results** We have deleted the original corollaries and made them simple quotes in remarks.

>**Unclear Proof Structure** We have further modularized the proof to three clearly-labeled sub-steps. We have highlighted the key ideas of each step. We also reorganized the proof to move some technical details to lemmas to prevent distraction by lengthy math expressions.

---
Response to Contents Necessary to Include(`VjNz`, `uiVo`)

>**Missing Citation** We have added a citation to Lai et al. (2024) and discussed the distinction in our related work.

>**Underemphasized Contribution** We verified to the extent of our knowledge that our work is indeed the first to establish a universal approximation theorem for a single-head softmax attention layer. We will make this significance more explicit in the revised paper.
---
Response to Learnability of Construction (`VjNz`)

>We clarified our work to be about expressivity, not learnability. We have included discussions of works extending universal approximation to provably learnable and generalizing algorithms.

---

### Decision · Program_Chairs · 2025-09-17

**Decision:**

Accept (poster)

**Comment:**

This work shows that a single-head Softmax attention model, when combined with a single layer (sum of linear transformations), can approximate to any desired accuracy any continuous function on a compact domain. With the exception of one reviewer, the reviewers appreciated the contribution, both technical and conceptual, and recommended acceptance. Based on my own reading and understanding of the work, the remarks (and associated score) of the outlier reviewer are unjustifiably adversarial. Consequently, I believe this paper merits acceptance.